

# Classification of out-of-time-order correlators

**Felix M. Haehl**[1*], **R. Loganayagam**[2†], **Prithvi Narayan**[2‡] **and Mukund Rangamani**[3◦]

**1** Department of Physics and Astronomy, University of British Columbia,
6224 Agricultural Road, Vancouver, B.C. V6T 1Z1, Canada.
**2** International Centre for Theoretical Sciences (ICTS-TIFR),
Shivakote, Hesaraghatta Hobli, Bengaluru 560089, India.
**3** Center for Quantum Mathematics and Physics (QMAP),
Department of Physics, University of California, Davis, CA 95616 USA.

* f.m.haehl@gmail.com, † nayagam@gmail.com,
‡ prithvi.narayan@gmail.com, ◦ mukund@physics.ucdavis.edu

## Abstract

The space of $n$-point correlation functions, for all possible time-orderings of operators, can be computed by a non-trivial path integral contour, which depends on how many time-ordering violations are present in the correlator. These contours, which have come to be known as timefolds, or out-of-time-order (OTO) contours, are a natural generalization of the Schwinger-Keldysh contour (which computes singly out-of-time-ordered correlation functions). We provide a detailed discussion of such higher OTO functional integrals, explaining their general structure, and the myriad ways in which a particular correlation function may be encoded in such contours. Our discussion may be seen as a natural generalization of the Schwinger-Keldysh formalism to higher OTO correlation functions. We provide explicit illustration for low point correlators ($n \leq 4$) to exemplify the general statements.



# 1 Introduction

Euclidean quantum field theories are completely defined by their vacuum correlation functions, sometimes referred to as Schwinger functions [1]. These Schwinger functions, can be viewed as suitable analytic continuation of Wightman functions, which, in turn, describe the Lorentzian theory. The passage of going from the Euclidean theory to the Lorentzian one is captured by the theorems of Osterwalder-Schrader [2,3]. They assert that one can construct a Poincaré invariant relativistic quantum field theory whose observables are given by suitable analytic continuations of the Schwinger functions.

A-priori the Schwinger functions are bereft of any temporal ordering, owing to the absence of causal ordering in Euclidean signature. Given a particular Euclidean correlation function, the Lorentzian correlators can be recovered by suitably analytically continuing the arguments [4], through some $i\epsilon$ prescription. However, one can ask for an intrinsically Lorentzian formalism to algorithmically construct such correlators. The natural home for the

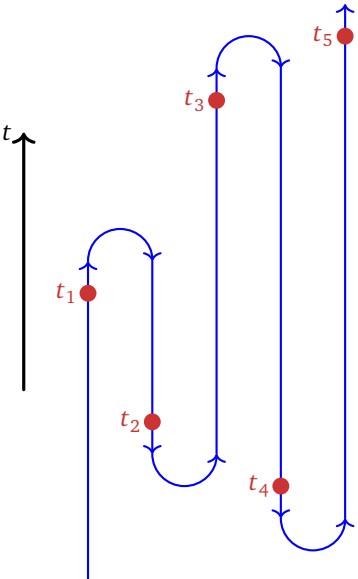

Figure 1: The timefolded contour necessary to compute the correlator with temporal ordering $t_1 > t_2$, $t_2 < t_3$, $t_3 > t_4$ and $t_4 < t_5$. In the above we have drawn the time running upwards, but soon we will switch to a notation where forward evolution runs left to right.

Lorentzian/relativistic analog of these Schwinger functions, happens to be the space of out-of-time-order, or timefolded correlation functions, which are the central focus of our analysis.[1]

To appreciate this point, consider a generic $n$-point function of Heisenberg operators, $\widehat{\mathbb{O}}_i(t_i)$, whose temporal locations are as indicated (we suppress the spatial positions because of their irrelevance to the present discussion). The Wightman correlation functions of interest are correlators, $\langle \widehat{\mathbb{O}}_1(t_1)\widehat{\mathbb{O}}_2(t_2)\cdots\widehat{\mathbb{O}}_n(t_n)\rangle$, with no prescribed temporal ordering. Writing out this correlation function in terms of Schrödinger operators, $\widehat{\mathbb{O}}_i(t_0) \equiv \mathbb{O}_i$, using $\widehat{\mathbb{O}}(t) = U(t_0,t)^\dagger\,\mathbb{O}\,U(t_0,t)$, we obtain

$$
\begin{aligned}
G(t_1,\cdots,t_n) &\equiv \langle \widehat{\mathbb{O}}_1(t_1)\widehat{\mathbb{O}}_2(t_2)\cdots\widehat{\mathbb{O}}_n(t_n)\rangle \\
&= \langle U^\dagger(t_0,t_1)\,\mathbb{O}_1\,U(t_0,t_1)\,U^\dagger(t_0,t_2)\,\mathbb{O}_2\,U(t_0,t_2)\cdots U^\dagger(t_0,t_n)\,\mathbb{O}_n\,U(t_0,t_n)\rangle\,.
\end{aligned}
\tag{1}
$$

One sees that the temporal evolution of the system between the operator insertions involves a series of forward and backward evolutions by $U(t_0,t_i)$ and $U^\dagger(t_0,t_j)$ respectively. This is an inevitable consequence of the lack of any temporal ordering. One can represent such an evolution by a path integral contour, called the timefolded contour [5] which involves a series of temporal switchbacks, see Fig. 1.

Such a timefold path integral contour is the primary object of interest of our current discussion. We wish to work out in some detail how various higher out-of-time-order (henceforth OTO) correlation functions can be encoded in such timefold contours, and the redundancies involved in such embeddings. One motivation is to view this construction as a suitable generalization of the Schwinger-Keldysh path integral construction [6, 7] computing singly out-of-time-order correlators, as recently described by some of us in [8] (an excellent review of the material from a more traditional viewpoint is [9]). A closely related analysis of such

---

[1] We have chosen to phrase the discussion in the context of relativistic QFT emphasizing the distinction between Lorentzian and Euclidean correlators. The analysis however is more broadly applicable, since it distinguishes time-ordered correlators versus unordered ones. The former rely only on the existence of a causal ordering and per se our analysis applies as stated for non-relativistic systems as well.

correlation functions defined on such contours appears in [10]. We will elaborate more on the connections in the course of our discussion.

Perhaps the main issue to explain is the physical reason to be interested in such higher OTO correlation functions. After all, given the forward/backward flow of time in the contour, it is clear that no physical experiment can access such observables (at least naively). The initial motivation for examining such contours was to understand the role of precursor operators in holography [11]. Roughly speaking, a precursor is an operator, which when inserted at a given instant of time, say $t = t_1$, acts so as to reproduce the effect of an operator inserted at an earlier time $t_0 < t_1$. Precursor operators prove useful in black hole gedanken-experiments (in the holographic context). For example, they help understand how the dual field theory probes the spacetime behind the horizon [5, 12].

For the action of an operator at $t_0$, to be effectively encoded in the action of its precursor at an later time $t_1 > t_0$, it must be that the two are related by the usual Heisenberg evolution, viz., $\mathbb{O}_p(t_1) = U(t_1, t_0)\, \mathbb{O}(t_0)\, U^\dagger(t_0, t_1)$. The main point is that generically both the operator and its precursor cannot simultaneously be local, since (non-integrable) quantum evolution tends to be ergodic and scramble the action. More prosaically, expanding out a Heisenberg operator using the Baker-Campbell-Hausdorff formula, we will note a series of nested commutators, which can be taken to be a proxy for ever increasing complexity of the precursor operator [13].

Motivated by this intuition, [14] studied the behaviour of precursors and higher out-of-time-order correlation functions, as a diagnostic of quantum chaos in the context of black hole physics and holography. Their primary goal was to understand how black holes scramble information.[2] Inspired, by these holographic analyses, [16] argued for a fundamental bound on quantum processing. This is phrased as an upper bound on the Lyapunov exponent $\lambda_L \leq \frac{2\pi}{\beta}$, when evaluated in an initial thermal state (inverse temperature $\beta$). The Lyapunov exponent itself, is encoded in a particular out-of-time order four-point function. As is well known, this bound is saturated by holographic field theories dual to classical gravity, and by an interesting quantum mechanical model of free fermions, the Sachdev-Ye-Kitaev (SYK) model [17, 18] (and generalization thereof).[3]

The main point we want the readers to note is the following: not only do the out-of-time-order correlation functions span out the full space of observables in the theory, but they also contain interesting physical information pertaining to how quantum dynamics is sensitive to initial conditions. For most part of our discussion, we will take it as given that the out-of-time-order correlation functions are useful objects to study, and delineate some of their features.[4] Our goal here is to build up a useful formalism for analyzing such objects. Consequently, we will explain how to compute particular OTO correlation function in terms a path integral. As we shall see there is a large degree of redundancy involved in the process; multiple different contours will lead to the same correlator. This has to do with unitarity of quantum evolution; it is trivial to add identical forward/backward segments to any quantum evolution without affecting physical results (since $U U^\dagger = \mathbb{1}$). We will explain elements of how these redundancies can be understood by working with different sets of correlation functions. Almost all of the analysis we undertake involves understanding different combinatorial (and kinematic) properties of OTO correlation functions.

---

[2] The connection between the process of thermalization and out-of-time-order observables dates back to the discussion of [15].

[3] Inspired by these developments, various authors have considered oto correlation functions in a variety of lattice models to probe thermalization and lack thereof (as occurs in many-body localized phases), see [19–28] for a sampling of these developments. Of related interest are the $k$-design networks studied in [29, 30]. In some of the quantum information literature, one computes an operator average oto correlator which can then be related to Rènyi entropy. See §8 for additional comments.

[4] That said, there is an active interest in measurement of such observables; see [31–33] for interesting proposals to experimentally measure scrambling and chaos in quantum models using oto observables and tricks to avoid the backwards evolution of the system. See also [34–36] for preliminary experiments on this front.

The outline of this paper is as follows. In §2 we will first describe the basic object of interest: the $k$-OTO path integral, and the natural sets of observables useful in different contexts. We then give a succinct summary of our results in §2.3. This essentially involves explaining how to use the path integral to compute OTO correlations. The simplest way to proceed is to compare different collections of correlators adapted to the OTO contours, which, as we will explain, can be interpreted as an upgrade of the usual Schwinger-Keldysh construction. The reader interested in the basic results is invited to consult §2, which explains the basic framework and summarizes the salient results, and the examples we present in §7.

The bulk of the paper is devoted to providing justifications of our statements and involves various combinatorial arguments. In §3 we explain how to map from the basis of OTO correlations onto the space of nested commutators and anti-commutators. This paves the way for §4 where we give an extension of the Keldysh rules for the computation of OTO correlations from the $k$-OTO contour we define below. Subsequently, §5 works out the canonical presentation of a particular OTO correlation in terms of a $k$-OTO functional integral, and enumerates the redundancies encountered in such embeddings. The general results are exemplified for low-point functions (up to 4-point functions) in §7. Some useful technical steps which aid our analysis are collected in the Appendices.

## 2 The $k$-OTO timefold path integral

Let us begin by defining the class of timefolded path integrals we wish to consider. Without loss of generality we assume that we have a quantum system prepared in an initial density matrix $\hat{\rho}_{\text{initial}}$ and then consider a timefolded evolution of this initial state. We define the $k$-OTO path integral by suitably generalizing the Schwinger-Keldysh contour, which we recall, computes singly out-of-time-order correlators.

There are two natural ways to view the $k$-OTO path integral. The first is to imagine the contour as a codimension-1 curve in complex time plane with imaginary excursions between the forward and backward evolutions as depicted in Fig. 1. This picture naturally implements the evolution made explicit in (1).

The second, which we prefer, is to view each evolution as the action on an element of the Hilbert space. More specifically, we orient the contours as in Fig. 2 and view each horizontal segment as an implementation of $U$, if directed right, or $U^\dagger$, if directed left. We then can visualize the contour as operating on the $2k$-fold tensor product of the original quantum Hilbert space $\mathcal{H}$ and its dual $\mathcal{H}^*$. In other words, we imagine working with an extended state space of the system

$$\mathcal{H}_{k-oto} = \otimes_{\alpha=1}^{k} \mathcal{H}_{\alpha\text{R}} \otimes_{\alpha=1}^{k} \mathcal{H}_{\alpha\text{L}}^* \tag{2}$$

Viewing the contour as acting on an extended Hilbert space accords us the freedom to also enlarge the operator algebra of the quantum system. The operators are indexed by $2k$ labels, $\alpha$R and $\alpha$L with $\alpha \in \{1, 2, \cdots, k\}$. We adhere to the convention used in [8]: the right (R) operators are inserted in the forward segments while the left (L) operators are inserted in the backward segments. The operators which act on each of the tensor components, $\mathcal{H}_{\alpha\text{R}}$, $\mathcal{H}_{\alpha\text{L}}$ will be indexed as $\mathbb{O}_{\text{R}}^\alpha$ and $\mathbb{O}_{\text{L}}^\alpha$, respectively. In addition we denote the elements of the operator algebra acting on $\mathcal{H}$ with a hat, i.e., $\widehat{\mathbb{O}}$, so as to keep them notationally distinct.

Our aim is to have a formalism that enables the computation of arbitrary $n$-point functions of the single-copy operators, viz., $\langle \widehat{\mathbb{O}}_1(t_1) \widehat{\mathbb{O}}_2(t_2) \cdots \widehat{\mathbb{O}}_n(t_n) \rangle$ for all possible orderings. To economize on notation, we a-priori pick a definite ordering of the temporal instances, say $t_1 > t_2 > t_3 > \cdots > t_n$ without loss of generality, and permute the operators to attain all orderings of interest. We will also often refrain from writing out the explicit arguments, leaving

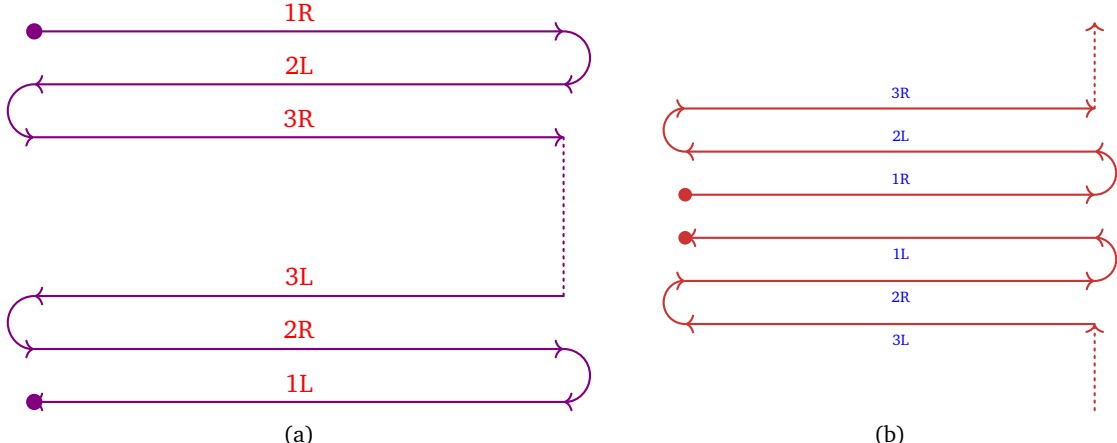

Figure 2: The k-OTO contour computing the out-of-time-ordered correlation functions encoded in the generating functional (4). (a) The contour drawn makes explicit the notion of *depth*; the segments with are nested inwards in order of increasing depth which is equivalent to the distance from the density matrix. (b) An alternate way of drawing the contour as e.g., used in [8] with contours of increasing depth going outwards from a central region. The second can be obtained from the first by turning the switchbacks inside-out.

it implicit for the single-copy operator algebra that the index of the operator corresponds to its temporal position, viz., $\widehat{\mathbb{O}}_j(t_j) \equiv \widehat{\mathbb{O}}_j$.[5]

We will first set forth the basic generating functional that will capture all of these OTO correlators. Once we have the basic functional integral of interest, we can then proceed to examine different sets of correlators and relations between them.

## 2.1 The $k$-OTO generating function

In order to facilitate the computation of correlation function we will allow ourselves the freedom of deforming the evolution by turning on external sources $\mathcal{J}$ that couple to the operators $\mathbb{O}$. The resulting evolution operator will be denoted as $U[\mathcal{J}]$ and is defined in terms of usual time ordered exponentials

$$U[\mathcal{J}] = \mathcal{T} \exp\left(-i \int_{t_i}^{t} dt\, H[\mathcal{J}]\right), \qquad (U[\mathcal{J}])^\dagger = \bar{\mathcal{T}} \exp\left(i \int_{t_i}^{t} dt\, H[\mathcal{J}]\right). \qquad (3)$$

We use the symbol $\mathcal{T}$ to denote time-ordering while $\bar{\mathcal{T}}$ denotes anti-time ordering. For any single horizontal segment of the contour these have conventional meaning. In the absence of sources, the unitaries reduce to the standard Heisenberg operators, e.g., for time independent Hamiltonians we would have $U = e^{-iHt}$.

To compute the out-of-time order correlation functions, we define the $k$-OTO generating function as follows [8]:

$$\mathcal{Z}_{k-oto}[\mathcal{J}_{\alpha R}, \mathcal{J}_{\alpha L}] = \text{Tr}\left(\cdots U[\mathcal{J}_{3R}](U[\mathcal{J}_{2L}])^\dagger U[\mathcal{J}_{1R}]\, \hat{\rho}_{\text{initial}}\, (U[\mathcal{J}_{1L}])^\dagger U[\mathcal{J}_{2R}](U[\mathcal{J}_{3L}])^\dagger \cdots\right), \qquad (4)$$

with $\alpha \in \{1, 2, \cdots, k\}$. As noted above, we have $k$ forward, and $k$ backward evolutions, in the $k$-OTO contour. The notation is meant to be suggestive; contours closer to the density matrix

---

[5] On occasion we will also find it convenient to write $\widehat{\mathbb{O}}_j(t_j) \equiv \widehat{\mathbb{O}}(j)$ (see §4.3) or even further simplified to $\widehat{\mathbb{O}}_j(t_j) \equiv j$ (cf., §3.2).

have lower value of the index. We will refer to the value of $\alpha$ as *depth*. In particular, will make a minor departure from the notation employed in [8] by declaring the contour with the highest index label to be nested innermost (furthest from $\hat{\rho}_{\text{initial}}$). This is perhaps easiest to visualize pictorially and is thus explicitly represented in Fig. 2.

Our terminology is meant to suggest the following interpretation: a 0-OTO contour is the standard Feynman path integral while a 1-OTO contour corresponds to the Schwinger-Keldysh contour which is usually invoked to compute time-ordered correlation functions. Aspects of the 2-OTO contours were recently discussed in [8,10] and we will review some of these results further.

The observables of interest are obtained by varying this generating function with respect to the sources. Following the usual discussion of the Schwinger-Keldysh formalism described in [8] we will implement a contour ordering procedure. Varying with respect to the sources, we will generate contour ordered correlators. We use $\mathcal{T}_C$ to denote the $k$-OTO contour ordering. The ordering is such that the 1R contour is past-most, the 1L is future-most, and the inner contours with $\alpha > 1$ will nest in between in the order they appear, viz., $1\text{R} < 2\text{L} < 3\text{R} < \cdots < 3\text{L} < 2\text{R} < 1\text{L}$. An explicit example is the following $(2k)$-point function

$$\langle \mathcal{T}_C \, \mathbb{O}^1_{\text{R},1}(t_1) \, \mathbb{O}^2_{\text{L},2}(t_2) \, \mathbb{O}^3_{\text{R},3}(t_3) \cdots \mathbb{O}^3_{\text{L},2k-2}(t_{2k-2}) \, \mathbb{O}^2_{\text{R},2k-1}(t_{2k-1}) \, \mathbb{O}^1_{\text{L},2k}(t_{2k}) \rangle, \tag{5}$$

with one operator on each of the $2k$ legs of a $k$-OTO contour. One can conveniently evaluate LR–correlation functions, which are adapted to the $k$-OTO contour, but the object of interest are the OTO correlations in the single copy theory. We should therefore find a way to evaluate them. This involves understanding useful classes of correlation functions, which will be our next focus.

## 2.2 Classes of OTO observables

With this background in place let us consider an $n$-point function $\langle \widehat{\mathbb{O}}_1(t_1)\widehat{\mathbb{O}}_2(t_2)\cdots\widehat{\mathbb{O}}_n(t_n) \rangle \equiv \langle \widehat{\mathbb{O}}_1\widehat{\mathbb{O}}_2\cdots\widehat{\mathbb{O}}_n \rangle$ in the single copy theory. While these are the primary objects which define our quantum theory, the $k$-OTO path integral contour is better adapted to a different set of objects involving the extended operator algebra. It is therefore useful, to consider the following collections of correlation functions.

**1. Wightman basis:** Let us without loss of generality fix the ordering of the temporal instances; say $t_1 > t_2 > \cdots > t_n$ for definiteness. Once we do this the space of $n$-point functions is simply spanned by the permutations of the operators $\widehat{\mathbb{O}}_i$. As noted before these act on the primary (single-copy) quantum Hilbert space $\mathcal{H}$.

The number of distinct $n$-point functions are easy to enumerate: we have $n!$ basic correlation functions to compute. Let us call the basis of observables which encode all of these $n!$ correlators as the *Wightman basis*. The elements of this basis can simply be taken to be

$$G_\sigma(t_1, t_2, \cdots, t_n) = \langle \widehat{\mathbb{O}}_{\sigma(1)} \widehat{\mathbb{O}}_{\sigma(2)} \cdots \widehat{\mathbb{O}}_{\sigma(n)} \rangle, \qquad \sigma \in S_n, \tag{6}$$

where $S_n$ denotes the group of permutations of $n$ objects.

There are now three other combinations of correlation functions that are interesting to consider. These are either, natural objects of interest physically, or aligned to the manner in which we evaluate the $k$-OTO functional integral.

**2. Nested correlators:** A second set of objects that is useful to consider is the space of nested commutators and anti-commutators of the $n$ operators $\widehat{\mathbb{O}}_i$. These are constructed in terms of the elementary building blocks which are commutators $[\cdot, \cdot]$ and anti-commutators $\{\cdot, \cdot\}$ of the operators.

Given these definitions, we can consider nesting a sequence of graded commutator, anti-commutators; for example

$$[\{[\widehat{\mathbb{O}}_1, \widehat{\mathbb{O}}_2], \widehat{\mathbb{O}}_3\}, \cdots], \tag{7}$$

which illustrates the general idea. We will enumerate this set to be spanned by $2^{n-2}n!$ correlators. We will implicitly assume that the operator algebra only has Grassmann even (bosonic) elements. It is straightforward to allow for both Grassmann even and odd elements by replacing the commutator/anti-commutator by the graded commutator/anti-commutator as in [8].[6] In the sequel we will refer to this space of correlation functions as *nested correlators* for brevity.

While this appears to be an added level of dressing atop the Wightman basis, this sequence of nested commutators has the utility of being more directly amenable to physical intuition. These objects, together with appropriate time-ordering step functions, form the basis of time-ordered response functions [9] (also see [8]). This statement should be familiar for 2-point functions, since the complete information of the propagator is contained in the commutator and anti-commutator. The reason for their importance can be traced to the fact that Lorentzian causal ordering ensures that the (graded) commutator of operators will vanish when the insertions are spacelike related.

**3. The LR correlators:** The third set of objects to consider is the space of correlation functions derived from the $k$-OTO contour. Since the $k$-OTO generating function (4) is a functional on the $2k$-fold tensor product operator algebra, one can imagine inserting on each leg of the contour, any of the $n$ operators of interest (or rather their images in $\mathcal{H}_{\alpha R}$ and $\mathcal{H}_{\alpha L}$, respectively). This leads to a total of $(2k)^n$ correlation functions, exemplified by (5).

This is however a vast over-determination, since many of these correlation functions can be collapsed to something simpler. For instance, by switching off or aligning some of the sources, we can collapse some of the timefolds using unitarity, viz., $U^\dagger U = \mathbb{1}$. In particular, $k$-OTO generating functional would collapse by aligning the inner most sources to a $j$-OTO with $j \leq k$. These result in localization limits, which were described in some detail for $k = 2$ in [8]. We will see soon that other alignments are also possible leading to a drastic reduction, down to the physical basis of $n!$ Wightman correlators.

**4. The Av-Dif correlators:** The final set of objects of interest involves a simple rotation of the LR-basis into the average-difference operator basis. This is done by a natural extension of the Keldysh basis used in the usual Schwinger-Keldysh formalism. We introduce:

$$\mathbb{O}_{av}^\alpha \equiv \frac{1}{2}\left(\mathbb{O}_R^\alpha + \mathbb{O}_L^\alpha\right) \equiv \underbrace{\left(\frac{1}{2} \quad \frac{1}{2}\right)}_{\xi^z}\underbrace{\begin{pmatrix}\mathbb{O}_R^\alpha \\ \mathbb{O}_L^\alpha\end{pmatrix}}_{\mathbb{O}_z^\alpha} = \xi^z \mathbb{O}_z^\alpha$$

$$\mathbb{O}_{dif}^\alpha \equiv (-1)^{\alpha+1}\left(\mathbb{O}_R^\alpha - \mathbb{O}_L^\alpha\right) \equiv (-1)^{\alpha+1}\underbrace{\left(1 \quad -1\right)}_{\eta}\begin{pmatrix}\mathbb{O}_R^\alpha \\ \mathbb{O}_L^\alpha\end{pmatrix} = \eta^z \mathbb{O}_z^\alpha \tag{8}$$

The matrices $\xi = \frac{1}{2}\begin{pmatrix}1 & 1\end{pmatrix}$ and $\eta = \begin{pmatrix}1 & -1\end{pmatrix}$, implement the linear transformation of import.

The averages and differences are taken for operators which are at the same depth or distance from the ends of the density matrix in the contour (see §2.4 and Fig. 2). The only novel element here is that even numbered legs have an relative sign in the definition of the difference operator to account for the fact that the backwards evolution precedes the forward evolution

---

[6] This generalization is most simply done by first converting all the Grassmann odd (fermionic) elements to be Grassmann even, e.g., by simple expedient of multiplying them by pure Grassmann odd numbers, say $\widehat{\mathbb{O}}_i \to \eta_i \widehat{\mathbb{O}}_i$. Extracting these $\eta(s)$ out of the nested correlator, we can then read off the signs for the graded brackets.

along the contour for such legs. Since this is a linear transformation on the set of LR correlators we also have $(2k)^n$ such objects. These also have to be suitably expressed in terms of the physical Wightman basis of correlation functions.

## 2.3 Summary of results

Now that we have identified the four classes of correlation functions that we will deal with, let us summarize the basic set of statements relating them to each other. We will justify these in the subsequent sections.

**1. The Wightman basis from the Euclidean correlator:** The Euclidean formulation of the QFT allows us to construct the Schwinger functions ($\tau$ denotes Euclidean time)

$$G_{\mathrm{E}}(\tau_1, \tau_2, \cdots, \tau_n) = \langle \widehat{\mathbb{O}}_1(\tau_1) \widehat{\mathbb{O}}_2(\tau_2) \cdots \widehat{\mathbb{O}}_n(\tau_n) \rangle. \tag{9}$$

The elements of the Wightman basis are then obtained by analytically continuing $\tau_i \to i\, t_i + \epsilon_i$, with $\epsilon_i$ ordered according to the permutation of interest, viz.,

$$G_\sigma(t_1, t_2, \cdots, t_m) = \lim_{\epsilon_i \to 0} G_{\mathrm{E}}(\tau_1, \tau_2, \cdots, \tau_n)\big|_{\tau_i = i\, t_i + \epsilon_i} \tag{10}$$

$$\epsilon_{\sigma(1)} > \epsilon_{\sigma(2)} > \cdots > \epsilon_{\sigma(n)}$$

Thus the $n!$ temporal orderings nicely translates in to the $n!$ orderings for $i\epsilon$ prescription. This is explained for instance in [4] and is nicely summarized in [37]. We will exemplify this with some low-point examples in §7.

**2. Nested correlators and Wightman basis §3:** While the nested commutator/anti-commutator correlation functions of the single-copy (operator algebra) operators are physically interesting, for reasons explained above, they are however a vastly redundant set.

Let us first count the elements of this set: given any of the $n!$ elements of the Wightman basis, we can partition them into binary sets for nesting in $(n-1)$ ways, e.g., by simply putting commas in-between the operators. For each such comma placement, we get to choose to enclose the relevant pair in a commutator or anti-commutator. This clearly amount to a set of $2^{n-1}$ choices. Not all of these are however independent, since we can use the anti-symmetry of the commutator to offset some permutations. One can quickly check that half of the original permutations can be thus accounted for, as we can restrict attention to even permutations $\sigma \in S_n$ with $\mathrm{sign}(\sigma) = 1$. All told, the number of nested commutator/anti-commutator correlators are then simply enumerated to be

$$\underbrace{2^{n-1}}_{\text{binary choice: } [\cdot,\cdot]_\pm, \{\cdot,\cdot\}_\pm} \times \underbrace{\frac{1}{2}\, n!}_{\text{permutations}} = 2^{n-2}\, n!. \tag{11}$$

The $2^{n-2}n!$ correlation functions spanned by nested correlators cannot be linearly independent; they should be expressible in terms of the $n!$ elements of the Wightman basis. The $(2^{n-2}-1)n!$ relations which we need, are best thought of as a set of generalized Jacobi identities involving commutators and anti-commutators.[7] Since the relations involve both the commutator and anti-commutator bracketing relation we need to go beyond standard Jacobi identities in constructing the desired relations.

---

[7] It is worthwhile noting that there are no redundancies at the level of 2-point functions, which is in accord with our physical intuition.

The standard operator algebra of a QFT from which $\widehat{\mathbb{O}}_i$ are drawn has two natural brackets $[\cdot,\cdot]$ and $\{\cdot,\cdot\}$. The commutator, of course, satisfies the familiar Jacobi identity:

$$[[\widehat{\mathbb{A}},\widehat{\mathbb{B}}],\widehat{\mathbb{C}}] = [\widehat{\mathbb{A}},[\widehat{\mathbb{B}},\widehat{\mathbb{C}}]] - [\widehat{\mathbb{B}},[\widehat{\mathbb{A}},\widehat{\mathbb{C}}]]. \tag{12}$$

However, this is one of many identities. It transpires that the full set of relations involves generalizations which increase the level of nesting and also use the second bracketing operation $\{\cdot,\cdot\}$. We will refer to these identities as *generalized super-Jacobi identities* (abbreviated to sJacobi).

Of these we will see that $(2^{n-2}-2)n!$ will be improper sJacobi identities, while $n!$ of them will be proper sJacobi identities. The distinction will lie in the action of $S_n$ on the set of sJacobi identities. Firstly, we note that the Wightman basis transforms in a regular representation $\mathcal{R}$ of $S_n$, while the nested correlator basis lies in $2^{n-2}\mathcal{R}$ representation. To count the family of improper sJacobi identities, we need to understand how to embed proper $m$ sJacobis for $m \leq n$ (thus consider possible subgroups of the permutation group, $S_m \subset S_n$ for $m < n$). We will show that these transform in $2^{n-k}\mathcal{R}$ of $S_n$, by suitably inducing the representations. Working inductively we then isolate the proper sJacobi identities on $n$-operators.

One advantage of invoking $S_n$ representation theory, is that, we can isolate a master sJacobi identity for each $m$. All other identities can be obtained by permutations and nestings of these master sJacobi identities.

**3. LR and Av-Dif correlators:**    The relation between the LR and Av-Dif correlators is a simple linear transformation that generalizes the usual construction of Keldysh basis [9]. This is already manifest from (8). Given that the LR-correlators arise from the $k$-OTO contour, we end with $(2k)^n$ $n$-point functions.

**4. OTO Keldysh rules §4:**    It is easiest to first relate the Av-Dif correlators to the physical set of nested $n$-point functions. This is a map from a $(2k)^n$ dimensional space into a space spanned by $2^{n-2}n!$ elements. The basic element of the construction should be familiar from the Schwinger-Keldysh formalism ($k = 1$). The Keldysh rules [8,9] give an explicit map using the contour ordering prescription to express a string of average-difference operator correlator in terms of nested commutator/anti-commutators of single copy operators, dressed with suitable time-ordering step functions. We need here a generalization of this construction, which turns out to be relatively easy.

Effectively, one isolates segments of the $k$-OTO contour that are part of a forward/backward or LR pair, equidistant from the density matrix (same depth). On odd numbered segments we apply the standard Keldysh rules, while on the even numbered ones we employ the CPT conjugate version to account for the reversed trajectory. Given that contour ordering defines for us the precise out-of-time-order, we just apply these rules sequentially starting from the outermost contours $1_R - 1_L$ and work our way into the deeper segments. This application results in an OTO Keldysh prescription which is given in Eq. (40).

**5. LR correlators and the Wightman basis §5:**    The last set of relations we describe is to map the $k$-OTO correlation functions to the basis of single copy correlators defining the theory. This is best described by giving a map, expressing the $(2k)^n$ LR-correlators in terms of the Wightman basis of $n!$ elements.

To explain the map, we first should realize that the minimal number of timefolds necessary to obtain every single $n$-point function is a $\lfloor\frac{n+1}{2}\rfloor$-OTO contour where $\lfloor x \rfloor$ denotes the integer part of $x$ [8]. This is easy to intuit, as the configuration with the most number of

timefolds involves a sawtooth pattern of operator insertions. Given the temporal ordering $t_1 > t_2 > \cdots > t_n$ this is attained for example in the sequence $\widehat{\mathbb{O}}_1 \widehat{\mathbb{O}}_n \widehat{\mathbb{O}}_2 \widehat{\mathbb{O}}_{n-1} \widehat{\mathbb{O}}_3 \cdots$.[8]

If $k > \lfloor \frac{n+1}{2} \rfloor$ then we clearly have a large degree of degeneracy, since we should be able to slide the operator along the contour, like beads on an abacus, to concatenate the $k$-OTO contour to a smaller contour. Even for $k < \lfloor \frac{n+1}{2} \rfloor$ we may have a simpler presentation for a particular time-ordering. For example, a completely time-ordered correlator of $n$-operators can be obtained from the $k = 0$ Feynman contour.

These observations motivate us to define a sequence of primitive contours, which compute particular orderings of $n$-point functions. We introduce in the course of our discussion, a notion of proper $q$-OTO, which allows us to give a canonical presentation of a given element of the Wightman basis in the timefolded functional integral. A proper $q$-OTO simply refers to the fact that we need a minimum of $q$ timefolds to represent the particular correlator.

Given then a $k$-OTO functional integral, there are two steps involved in ascertaining the desired map. First we construct all the proper $q$-OTO contours with $q = 1, 2, \cdots, \lfloor \frac{n+1}{2} \rfloor$. Such proper $q$-OTO contours, we show, compute $g_{n,q}$ of the $n!$ time-ordering correlators. The counts $g_{n,q}$ are given in (51). These numbers form interesting arithmetic sequences: in special case they are related in turn to tangent numbers (coefficients in the Taylor expansion of $\tan x$), which themselves are closely related to tremelo partitions.

The second step is to investigate the number of ways a proper $q$-OTO correlator embeds into our $k$-OTO contour. This involves a second counting problem, which can be shown to be related to the problem of computing the coordination sequence of a cubic lattice in Euclidean space. We will show that the counts are given then by $h_{n,k}^{(q)}$, see (54). Essentially this two-step procedure gives us a breakdown of the total set of $n$-point correlation functions into proper $q$-OTO-subsets. We have

$$n! = \sum_{q=1}^{\lfloor \frac{n+1}{2} \rfloor} g_{n,q}, \qquad (2k)^n = \sum_{q=1}^{\lfloor \frac{n+1}{2} \rfloor} g_{n,q} \, h_{n,k}^{(q)}. \tag{13}$$

The explicit expressions for $g_{n,q}$ and $h_{n,k}^{(q)}$ are given in §5, cf., Eqs. (51) and (54), respectively.

**6. Some low-point examples §7:** Let us record some useful facts based on the above discussion for $n \leq 4$. We remind the reader of our convention $t_1 > t_2 > \ldots > t_n$.

- One-point functions are clearly computed by the 0-OTO or Feynman contour. For convenience we will refrain from distinguishing this from the 1-OTO Schwinger-Keldysh contour.

- Two-point functions involve two orderings and thus require the Schwinger-Keldysh contour which is 1-OTO. We need a single timefold to compute the anti-time-ordered correlator $G(t_2, t_1) = \langle \widehat{\mathbb{O}}_2 \widehat{\mathbb{O}}_1 \rangle$. Proper 1-OTOs suffice for computing two-point functions. Given a $k$-OTO contour, the $(2k)^2$ LR-correlators split into $2k^2$ time-ordered and $2k^2$ anti-time ordered correlators. That is to say, $g_{2,1} = 2$ and $h_{2,k}^{(1)} = 2k^2$.

- Three-point functions can be obtained from at most 2-OTO contours. Of the 6 elements of the Wightman basis, 4 can be computed by proper 1-OTOs:

$$\langle \widehat{\mathbb{O}}_1 \widehat{\mathbb{O}}_2 \widehat{\mathbb{O}}_3 \rangle, \quad \langle \widehat{\mathbb{O}}_2 \widehat{\mathbb{O}}_1 \widehat{\mathbb{O}}_3 \rangle, \quad \langle \widehat{\mathbb{O}}_3 \widehat{\mathbb{O}}_2 \widehat{\mathbb{O}}_1 \rangle, \quad \langle \widehat{\mathbb{O}}_3 \widehat{\mathbb{O}}_1 \widehat{\mathbb{O}}_2 \rangle, \tag{14}$$

while 2 correlators require a proper 2-OTO contour:

$$\langle \widehat{\mathbb{O}}_1 \widehat{\mathbb{O}}_3 \widehat{\mathbb{O}}_2 \rangle, \quad \langle \widehat{\mathbb{O}}_2 \widehat{\mathbb{O}}_3 \widehat{\mathbb{O}}_1 \rangle. \tag{15}$$

---

[8] Permutations of objects which follow such a sawtooth pattern are referred to as *tremelo* permutations.

Thus $g_{3,1} = 4$ and $g_{3,2} = 2$. The degeneracies can be shown to be: $h_{3,k}^{(1)} = \frac{2k(2k^2+1)}{3}$ and $h_{3,k}^{(2)} = \frac{4k(k^2-1)}{3}$.

- Four-point functions are spanned by the basis of 24 correlators. Of these 8 are realized in a proper 1-OTO contour, while the remaining 16 require use of a proper 2-OTO contour. The former are enumerated to be the following $g_{4,1} = 8$ correlators:

$$\begin{aligned}
&\langle \widehat{\mathbb{O}}_1 \widehat{\mathbb{O}}_2 \widehat{\mathbb{O}}_3 \widehat{\mathbb{O}}_4 \rangle, \quad \langle \widehat{\mathbb{O}}_2 \widehat{\mathbb{O}}_1 \widehat{\mathbb{O}}_3 \widehat{\mathbb{O}}_4 \rangle, \quad \langle \widehat{\mathbb{O}}_3 \widehat{\mathbb{O}}_1 \widehat{\mathbb{O}}_2 \widehat{\mathbb{O}}_4 \rangle, \quad \langle \widehat{\mathbb{O}}_3 \widehat{\mathbb{O}}_2 \widehat{\mathbb{O}}_1 \widehat{\mathbb{O}}_4 \rangle, \\
&\langle \widehat{\mathbb{O}}_4 \widehat{\mathbb{O}}_1 \widehat{\mathbb{O}}_2 \widehat{\mathbb{O}}_3 \rangle, \quad \langle \widehat{\mathbb{O}}_4 \widehat{\mathbb{O}}_2 \widehat{\mathbb{O}}_1 \widehat{\mathbb{O}}_3 \rangle, \quad \langle \widehat{\mathbb{O}}_4 \widehat{\mathbb{O}}_3 \widehat{\mathbb{O}}_1 \widehat{\mathbb{O}}_2 \rangle, \quad \langle \widehat{\mathbb{O}}_4 \widehat{\mathbb{O}}_3 \widehat{\mathbb{O}}_2 \widehat{\mathbb{O}}_1 \rangle,
\end{aligned} \tag{16}$$

while the latter are spanned by the $g_{4,2} = 16$ combinations

$$\begin{aligned}
&\langle \widehat{\mathbb{O}}_1 \widehat{\mathbb{O}}_2 \widehat{\mathbb{O}}_4 \widehat{\mathbb{O}}_3 \rangle, \langle \widehat{\mathbb{O}}_1 \widehat{\mathbb{O}}_3 \widehat{\mathbb{O}}_2 \widehat{\mathbb{O}}_4 \rangle, \langle \widehat{\mathbb{O}}_1 \widehat{\mathbb{O}}_3 \widehat{\mathbb{O}}_4 \widehat{\mathbb{O}}_2 \rangle, \langle \widehat{\mathbb{O}}_1 \widehat{\mathbb{O}}_4 \widehat{\mathbb{O}}_2 \widehat{\mathbb{O}}_3 \rangle, \langle \widehat{\mathbb{O}}_1 \widehat{\mathbb{O}}_4 \widehat{\mathbb{O}}_3 \widehat{\mathbb{O}}_2 \rangle, \\
&\langle \widehat{\mathbb{O}}_2 \widehat{\mathbb{O}}_1 \widehat{\mathbb{O}}_4 \widehat{\mathbb{O}}_3 \rangle, \langle \widehat{\mathbb{O}}_2 \widehat{\mathbb{O}}_3 \widehat{\mathbb{O}}_1 \widehat{\mathbb{O}}_4 \rangle, \langle \widehat{\mathbb{O}}_2 \widehat{\mathbb{O}}_3 \widehat{\mathbb{O}}_4 \widehat{\mathbb{O}}_1 \rangle, \langle \widehat{\mathbb{O}}_2 \widehat{\mathbb{O}}_4 \widehat{\mathbb{O}}_1 \widehat{\mathbb{O}}_3 \rangle, \langle \widehat{\mathbb{O}}_2 \widehat{\mathbb{O}}_4 \widehat{\mathbb{O}}_3 \widehat{\mathbb{O}}_1 \rangle, \\
&\langle \widehat{\mathbb{O}}_3 \widehat{\mathbb{O}}_1 \widehat{\mathbb{O}}_4 \widehat{\mathbb{O}}_2 \rangle, \langle \widehat{\mathbb{O}}_3 \widehat{\mathbb{O}}_2 \widehat{\mathbb{O}}_4 \widehat{\mathbb{O}}_1 \rangle, \langle \widehat{\mathbb{O}}_3 \widehat{\mathbb{O}}_4 \widehat{\mathbb{O}}_1 \widehat{\mathbb{O}}_2 \rangle, \langle \widehat{\mathbb{O}}_3 \widehat{\mathbb{O}}_4 \widehat{\mathbb{O}}_2 \widehat{\mathbb{O}}_1 \rangle, \langle \widehat{\mathbb{O}}_4 \widehat{\mathbb{O}}_1 \widehat{\mathbb{O}}_3 \widehat{\mathbb{O}}_2 \rangle, \langle \widehat{\mathbb{O}}_4 \widehat{\mathbb{O}}_2 \widehat{\mathbb{O}}_3 \widehat{\mathbb{O}}_1 \rangle.
\end{aligned} \tag{17}$$

In a $k$-OTO contour, each proper 1-OTO combination occurs with degeneracy $h_{4,k}^{(1)} = \frac{2k^2(k^2+2)}{3}$, whereas each proper 2-OTO combination occurs $h_{4,k}^{(2)} = \frac{2k^2(k^2-1)}{3}$ times.

It is useful to note that a $k$-OTO contour is only required for the computation of $2k-1$ or $2k$-point correlation functions. For lower point functions they are an overkill, and thus we note that there must be some intrinsic redundancy built into the construction. This statement is very familiar in the context of the Schwinger-Keldysh formalism, as reviewed in [8]. As discussed, there are various localizations of the $k$-OTO correlation function. As in the 1-OTO case we anticipate that there is an underlying BRST symmetry that controls such localizations and leads to the myriad relations detailed above. For the rest of this paper we will focus on justifying the statements we have summarized above. A separate publication will detail how to view these in terms of various BRST Ward identities along the lines of [8].

## 2.4 $k$-OTO contour nomenclature

As we go through the discussion of the $k$-OTO contour, it will be useful to refer to various elements of the contour depicted in Fig. 2. to this end we introduce some nomenclature which will help in identifying elements of the contour, and also the operator insertions on it.

**Depth:** In referring to the individual legs of the contour, we will use the notion of depth. A segment indexed by $\alpha$ is said to be deeper for larger values of $\alpha$. In terms of the trace representation (4) deeper segments are further from the initial density matrix $\hat{\rho}_{\text{initial}}$. This way the index ordering directly gives us the depth which will prove useful in relating the $k$-OTO contour correlators to the physical single copy theory. See Fig. 2 for an illustration.

**Proper OTO number:** This was alluded to above, and is defined as the minimal OTO number required to reproduce a particular element of Wightman basis. It is important to note that we will take both, the fully time ordered correlator, and the fully anti time ordered correlator, to have the proper OTO number as 1.

**Future turning-point:** The future turning-point is defined as the junction between $\{(2j-1)\,\text{R}, (2j)\,\text{L}\}$ or between $\{(2j)\,\text{R}, (2j-1)\,\text{L}\}$ with $j \geq 1$) segments of the path integral contour. That is to say, future turning-points are the turning-points at the right ends of Fig. (2)(a). There are $q$ such future turning-points in a proper $q$-OTO correlator.

**Past turning-point:** The past turning-point is defined as the junction between $\{(2j)\mathrm{R}, (2j+1)\mathrm{L}\}$ or between $\{(2j+1)\mathrm{L}, (2j)\mathrm{R}\}$ with $j \geq 1$ segments of the path integral contour. Thus past turning-points are the turning-points at the left ends of Fig. (2)(a). There are $q-1$ such future turning-points in a proper $q$-OTO correlator.

**Turning-point operator:** An operator inserted just before a turning-point will be referred to as such. Often we will prefix this with noting whether we are describing a future or a past turning-point operator.

**Wings and wing operators:** The segment of the $k$-OTO contour between a particular future turning-point operator, and its nearest neighbour past turning-point on either side will be referred to a the *wing* of the future turning-point operator in question. The non turning-point operators along this wing, will be called the *wing operators*. While implementing some elements of the counting, we will further define the notion of wing neighbours, wing-spread, and wing position, which will serve to provide us with a useful way to package the sliding rules along the contour.

**Symbol for the OTO contour:** While it is easy to draw a particular OTO contour, we find it convenient to introduce a compact symbol which essentially gives a pictorial representation. The representation involves denoting the density matrix as $\circ$, at its past/future ends, the future turning-points as $)$, past turning-points as $($, and the operator insertions as numbers $1, 2, \cdots, n$. With an understanding that $t_1 > t_2 > \cdots > t_n$, a string of these symbols represents a contour-ordered $k$-OTO correlator. For example $\langle \widehat{\mathbb{O}}_2 \widehat{\mathbb{O}}_3 \widehat{\mathbb{O}}_1 \rangle = \circ 1)3(2)\circ$ symbolizes a 2-OTO correlator. This is explained in greater detail in §5.

# 3 Reducing nested correlators

We begin our discussion by describing the physical basis of nested correlation functions and its relation to the basis of time-ordered correlation functions. As described earlier, the reason to be interested in the nested commutator/anti-commutators has to do with the fact that they naturally map, in some circumstances, to response functions that are of physical interest.

The objects we are interested in are basically a sequence of nested commutators and anti-commutators of $n$-operators. The latter are elements of the operator algebra that act on the physical Hilbert space $\mathcal{H}$. In a certain sense, we can think of the nested operators as a construction of the free algebra generated by the elements of the physical operator algebra, given two brackets, the commutator and anti-commutator, which map pairs of operators into a new element.[9]

Let us introduce a convenient notation unifying the brackets in question. Define

$$[\widehat{\mathbb{O}}_1, \widehat{\mathbb{O}}_2]_\varepsilon = \begin{cases} [\widehat{\mathbb{O}}_1, \widehat{\mathbb{O}}_2], & \varepsilon = 1 \\ \{\widehat{\mathbb{O}}_1, \widehat{\mathbb{O}}_2\}, & \varepsilon = -1 \end{cases}. \tag{18}$$

Consider first the Wightman basis of correlation functions, whose $n!$ elements we denote for brevity as $G_\sigma = G(t_{\sigma(1)}, t_{\sigma(2)}, \cdots, t_{\sigma(n)})$, viz.,

$$G_\sigma = \langle \widehat{\mathbb{O}}_{\sigma(1)} \widehat{\mathbb{O}}_{\sigma(2)} \cdots \widehat{\mathbb{O}}_{\sigma(n)} \rangle \qquad \sigma \in S_n. \tag{19}$$

---

[9] Conventionally a free Lie algebra is the set of elements generated by the basic Lie commutator action on the elements of an algebra. This is very similar in spirit to the notion of a free group, where we construct elements as words built out the alphabets (the group generators) The main difference from a free Lie algebra is that we have two brackets and only one of them is anti-symmetric. Somewhat curiously, we have not been able to find a discussion of such constructs in the mathematics literature.

These elements span a vector space, and our interest is in identifying interesting members of the resulting space, and give useful expressions for them, in terms of these basis elements. We will use $\sigma$ to also index the $n!$ elements of $S_n$ so as to write compact formulae below.

Given $n$ elements $\widehat{\mathbb{O}}_i$ of the operator algebra,[10] we can form $2^{n-2}n!$ combinations

$$\mathfrak{C}_I = \langle [ \cdots [[[\widehat{\mathbb{O}}_{\pi(1)}, \widehat{\mathbb{O}}_{\pi(2)}]_{\varepsilon_1}, \widehat{\mathbb{O}}_{\pi(3)}]_{\varepsilon_2}, \cdots ]_{\varepsilon_{n-1}} \rangle \\ \text{with } I \equiv \{\pi, (\varepsilon_1, \varepsilon_2, \ldots, \varepsilon_{n-1})\}, \quad \pi \in S_n^+, \quad \varepsilon_1, \ldots \varepsilon_{n-1} \in \{+, -\}. \tag{20}$$

Note that allowing any permutation $\pi$ would naively give $2^{n-1}n!$ different multi-indices $I$; however, by restricting to even permutations $S_n^+$ we consider those which only differ by a swap of $i_1$ and $i_2$ as being the same because their associated correlators at most differ by an overall sign. This leaves us with $2^{n-2}n!$ possibilities. The total number of $\mathfrak{C}_I$ follows from the counting argument given in §2.2 where we enumerate the $2^{n-2}n!$ possibilities, which far exceeds the set of time-ordering correlators which only amount to $n!$ correlators. The index $I$ collectively encodes both the choice of permutation and the various choices of brackets involved, as indicated.

This then implies that we should exhibit $(2^{n-2}-1)n!$ relations amongst the nested correlation functions. These relations should be purely algebraic in nature and their origins are easily intuited. Expanding out the brackets in $\mathfrak{C}_I$ we have

$$\mathfrak{C}_I = \sum_\sigma \mathfrak{M}_{I\sigma} G_\sigma, \qquad \mathfrak{M}_{I\sigma} = \pm 1. \tag{21}$$

The matrix of coefficients $\mathfrak{M}$ is $2^{n-2}n! \times n!$ in size, with its rank being $n!$.

From this matrix we can determine the relations between the nested correlators. Moreover, it is possible to directly construct a projector onto the subspace of relations $\mathfrak{J}_p$. This is done by finding the kernel of its transpose, $\mathfrak{M}^T$, which defines a matrix of relations, $\mathfrak{J}^T$. Equivalently, we directly define $\mathfrak{J}$ as annihilating the $\mathfrak{M}$ from the left, viz.,

$$\mathfrak{J} \cdot \mathfrak{M} = 0, \qquad \dim(\mathfrak{J}) = 2^{n-2}n! \times (2^{n-2}-1)n!. \tag{22}$$

One can rotate this matrix $\mathfrak{J}$ to obtain a projector onto the space of relations, $\mathfrak{J}_p$. This can for instance be done using singular value decomposition of $\mathfrak{J} = \mathfrak{u}_{\mathfrak{J}} \mathfrak{s}_{\mathfrak{J}} \mathfrak{v}_{\mathfrak{J}}^\dagger$. The projector is then given by taking the first $(2^{n-2}-1)n!$ rows of $\mathfrak{v}_{\mathfrak{J}}^\dagger$, viz., $\mathfrak{J}_p = \mathfrak{v}_{\mathfrak{J}} \mathfrak{v}_{\mathfrak{J}}^\dagger$, which satisfies $\mathfrak{J}_p^2 = \mathfrak{J}_p$. We will think of these relations as a set of generalized super-Jacobi identities (sJacobi), and will justify the counting below.

## 3.1 Proper and improper sJacobi identities

We claim that the set of sJacobi identities captured by $\mathfrak{J}_p$ (or equivalently $\mathfrak{J}$) naturally splits into two classes: a class of proper sJacobi, $\mathfrak{P}_P$ which are $n!$ in number, and the remainder $\mathfrak{I}_P$ which amount to $(2^{n-2}-2)n!$. We have $\mathfrak{J}_p = \mathfrak{P}_P + \mathfrak{I}_P$ with each matrix being a projector onto the appropriate subspace of sJacobi identities. These manipulations are easy to carry out explicitly to check that the dimensions of the spaces are as quoted. The improper sJacobi identities refer to relations that are inherited from lower order sJacobi involving $j < n$ operators. We will now give a more abstract group theoretic proof of this decomposition.

**Regular representations of finite groups:** Recall that left multiplication in group $G$ permutes the elements of the group, thus giving rise to a permutation representation called the *regular representation* $\mathcal{R}(G)$. We will need the following group theory lemma (the proof can

---

[10] Recall that by convention $\widehat{\mathbb{O}}_i$ are Heisenberg operators inserted at time $t_i$.

be found in Appendix C.1):

*Lemma:* The regular representation of the group is induced by regular representation of a subgroup.

**Decomposition of nested correlator relations:**   The relevance of the above observations stems from the fact that $n$-pt Wightman correlators lie in the regular representation $\mathcal{R}(S_n)$. Nested correlators on the other hand are of the form

$$[\dots[\{\widehat{\mathbb{O}}_1,\widehat{\mathbb{O}}_2\},\widehat{\mathbb{O}}_3]_{\varepsilon_1}\dots,\widehat{\mathbb{O}}_n]_{\varepsilon_{n-2}}\,,\quad [\dots[[\widehat{\mathbb{O}}_1,\widehat{\mathbb{O}}_2],\widehat{\mathbb{O}}_3]_{\varepsilon_1}\dots,\widehat{\mathbb{O}}_n]_{\varepsilon_{n-2}}\,. \tag{23}$$

Equivalently we can consider the linear combinations

$$\begin{aligned}&[\dots[\{\widehat{\mathbb{O}}_1,\widehat{\mathbb{O}}_2\},\widehat{\mathbb{O}}_3]_{\varepsilon_1}\dots,\widehat{\mathbb{O}}_n]_{\varepsilon_{n-2}}+[\dots[[\widehat{\mathbb{O}}_1,\widehat{\mathbb{O}}_2],\widehat{\mathbb{O}}_3]_{\varepsilon_1}\dots,\widehat{\mathbb{O}}_n]_{\varepsilon_{n-2}}\,,\\&[\dots[\{\widehat{\mathbb{O}}_1,\widehat{\mathbb{O}}_2\},\widehat{\mathbb{O}}_3]_{\varepsilon_1}\dots,\widehat{\mathbb{O}}_n]_{\varepsilon_{n-2}}-[\dots[[\widehat{\mathbb{O}}_1,\widehat{\mathbb{O}}_2],\widehat{\mathbb{O}}_3]_{\varepsilon_1}\dots,\widehat{\mathbb{O}}_n]_{\varepsilon_{n-2}}\,,\end{aligned} \tag{24}$$

which together transform in the regular representation $\mathcal{R}(S_n)$, for a given set of $(n-2)$ sign choices $\{\varepsilon_\alpha\}$. Taking every sign choice into account, one gets $2^{n-2}$ copies of $\mathcal{R}(S_n)$. Since the $2^{n-2}n!$ nested correlators can be constructed by taking the direct sum of the vector space of $n!$ Wightman correlators along with the vector space of sJacobi relations. The sJacobis then have to lie in $(2^{n-2}-1)$ copies of $\mathcal{R}(S_n)$. This justifies our count for the rank of $\mathfrak{J}$ (and $\mathfrak{J}_{\mathrm{p}}$).

We will now study the structure of these sJacobis in more detail. Given a sJacobi with $k$ operators with $3 \leq k < n$, there is a way to lift it to a sJacobi with $n$ operators: we simply nest the sJacobi with $k$ operators within $(n-k)$ number of commutators/anti-commutators to get a sJacobi with $n$ operators. The sJacobis with $n$ operators obtained this way will be called improper $n$ sJacobis, whereas an sJacobi with $n$ operators which cannot be formed this way, will be called proper $n$ sJacobi. Thus improper $n$ sJacobis are formed from all proper $k$ sJacobis with $k < n$.

*Theorem 1:* Proper $n$ sJacobi identities lie in the regular representation $\mathcal{R}(S_n)$.

Equivalently, we can assert that improper sJacobis lie in $(2^{n-2}-2)$ copies of $\mathcal{R}(S_n)$, since the representation of sJacobis should be a direct sum of these two. We prove the latter statement by induction in Appendix C.2.

## 3.2   The master proper sJacobi identity

The above abstract result in representation theory is very useful in studying the structure of sJacobis. It implies that there is a single 'master' sJacobi relation for each $n$. All other sJacobi relations between the nested correlator are generated from it in the following sense. The proper $n$ sJacobis are generated by permutations of the master $n$ sJacobi. All improper n sJacobis are generated from all proper $k$ sJacobis with $k < n$ by nesting. Thus, it suffices to write down a single relation for every $n$.

Unfortunately, while we have proved the existence of such a relation, we have not yet found an efficient way to construct these relations for arbitrary $n$. The master sJacobi for $n = 3, 4$ can however be worked out by trial and error and we will report on them below.[11]

---

[11] The explicit computation can be done by working out the projector matrices $\mathfrak{J}_{\mathrm{p}}$, $\mathfrak{P}_{\mathrm{p}}$, and $\mathfrak{I}_{\mathrm{p}}$ explicitly for any $n$. The tricky part is then to identify combinations of the $n!$ elements of $\mathfrak{P}_{\mathrm{p}}$ that actually results in the master identity.

**sJacobi for n=3:** The master sJacobi relation for $n = 3$ is given by (writing $\widehat{\mathbb{O}}_j \equiv j$, and dropping the $\langle \cdot \rangle$ for conciseness)

$$[\{1,2\},3] + [[1,2],3] - \{\{2,3\},1\} - \{[2,3],1\} + \{\{3,1\},2\} + \{[3,1],2\} = 0 \,. \qquad (25)$$

This means that a complete basis of proper sJacobis at $n = 3$ can be obtained by applying various permutations to the above identity. There are no improper sJacobis at $n = 3$. Thus, for every element of the permutation group $S_3$, we have an identity:[12]

$$
\begin{aligned}
\text{Id}: \quad & [\{1,2\},3] + [[1,2],3] - \{\{2,3\},1\} - \{[2,3],1\} + \{\{3,1\},2\} + \{[3,1],2\} = 0\,, \\
(12): \quad & [\{2,1\},3] + [[2,1],3] - \{\{1,3\},2\} - \{[1,3],2\} + \{\{3,2\},1\} + \{[3,2],1\} = 0\,, \\
(23): \quad & [\{1,3\},2] + [[1,3],2] - \{\{3,2\},1\} - \{[3,2],1\} + \{\{2,1\},3\} + \{[2,1],3\} = 0\,, \\
(31): \quad & [\{3,2\},1] + [[3,2],1] - \{\{2,1\},3\} - \{[2,1],3\} + \{\{1,3\},2\} + \{[1,3],2\} = 0\,, \\
(123): \quad & [\{2,3\},1] + [[2,3],1] - \{\{3,1\},2\} - \{[3,1],2\} + \{\{1,2\},3\} + \{[1,2],3\} = 0\,, \\
(321): \quad & [\{3,1\},2] + [[3,1],2] - \{\{1,2\},3\} - \{[1,2],3\} + \{\{2,3\},1\} + \{[2,3],1\} = 0\,.
\end{aligned}
\qquad (26)
$$

These identities are linearly independent and hence furnish a basis for the six dimensional vector space of sJacobi relations at $n = 3$. In the form written above, they also manifestly lie in the regular representation of $S_3$ that permutes the three operators.

By standard representation theory, the regular representation of $S_3$ breaks up into irreps as $\mathbf{6} = \mathbf{1} + \mathbf{1}' + 2 \times \mathbf{2}$. Here $\mathbf{1}$ is the trivial irrep, $\mathbf{1}'$ is the sign irrep where the odd permutations are represented by $(-1)$, and $\mathbf{2}$ is the standard irrep of $S_3$. We can thus use the representation theory techniques to project out simpler sJacobi relations from the above. We first begin by projecting to the subspace where the exchanges act symmetrically. We get

$$
\begin{aligned}
\tfrac{1}{2}[\text{Id} + (12)]: \quad & [\{1,2\},3] - \{\{2,3\},1\} + \{\{3,1\},2\} = 0\,, \\
\tfrac{1}{2}[(23) + (321)]: \quad & [\{1,3\},2] - \{\{3,2\},1\} + \{\{2,1\},3\} = 0\,, \\
\tfrac{1}{2}[(31) + (123)]: \quad & [\{3,2\},1] - \{\{2,1\},3\} + \{\{1,3\},2\} = 0\,.
\end{aligned}
\qquad (27)
$$

These three identities manifestly form a 3 dimensional sub-representation of proper sJacobis. This sub-representation, in turn breaks into $\mathbf{1} + \mathbf{2}$. By taking a sum of all three, we get a sJacobi in the trivial irrep $\mathbf{1}$:

$$\tfrac{1}{2}[\text{Id} + (12) + (23) + (31) + (123) + (321)]: \quad [\{1,2\},3] + [\{1,3\},2] + [\{3,2\},1] = 0\,. \quad (28)$$

We recognize this as the sJacobi relation arising from the associativity of a supersymmetry (or BRST) action. The other two linear combinations then form the standard irrep $\mathbf{2}$.

The rest of the decomposition follows similarly: projecting onto the subspace where the exchanges act anti-symmetrically, we obtain

$$
\begin{aligned}
\tfrac{1}{2}[\text{Id} - (12)]: \quad & [[1,2],3] - \{\{2,3\},1\} + \{\{3,1\},2\} = 0\,, \\
\tfrac{1}{2}[(321) - (23)]: \quad & [[3,1],2] - \{\{1,2\},3\} + \{\{2,3\},1\} = 0\,, \\
\tfrac{1}{2}[(123) - (31)]: \quad & [[2,3],1] - \{\{3,1\},2\} + \{\{1,2\},3\} = 0\,.
\end{aligned}
\qquad (29)
$$

---

[12] We use the standard cycle notation to denote elements of $S_3$ and as per convention do not explicitly show 1-cycles.

By taking a sum of all three, we get a sJacobi in the sign irrep $\mathbf{1}'$ :

$$\frac{1}{2}[2\,\mathrm{Id}-(12)-(23)-(31)+(123)+(321)]: \quad [[1,2],3]+[[3,1],2]+[[2,3],1]=0. \quad (30)$$

This as the standard Jacobi identity. The other two linear combinations form another copy of the standard irrep $\mathbf{2}$. One can also project into the two copies of standard irrep $\mathbf{2}$ by taking the combination $\frac{1}{2}[\mathrm{Id}-(123)-(321)]$.

**sJacobi for $n = 4$:** We next move to $n = 4$. First of all, there improper sJacobis are in $(2^{n-2}-2)=2$ copies of $\mathcal{R}(S_4)$. These are obtained respectively by nesting the $n = 3$ sJacobis inside a commutator and an anti-commutator. Thus, $\mathfrak{I}_{\mathrm{P}}$ for $n = 4$ is generated by the two sJacobi relations

$$
\begin{aligned}
&[[\{1,2\},3],4]+[[[1,2],3],4]-[[\{2,3\},1],4]-[\{[2,3],1\},4]\\
&\qquad\qquad +[\{\{3,1\},2\},4]+[\{[3,1],2\},4]=0,\\
&\{[\{1,2\},3],4\}+\{[[1,2],3],4\}-\{\{\{2,3\},1\},4\}-\{\{[2,3],1\},4\}\\
&\qquad\qquad +\{\{\{3,1\},2\},4\}+\{\{[3,1],2\},4\}=0,
\end{aligned}
\quad (31)
$$

and their $4! = 24$ permutations. Each set transforms in $\mathcal{R}(S_4)$, and together, they give a complete basis for 48 improper sJacobis at $n = 4$.

By trial and error, one can work out the master sJacobi at $n = 4$. It is given by the somewhat complicated expression:

$$
\begin{aligned}
&-[[[1,2],3],4]]-[[[1,3],2],4]]\\
&-\{[[1,2],3],4\}-\{[[1,2],4],3\}-\{[[1,4],2],3\}+\{[[2,3],1],4\}\\
&+[\{[1,2],3\},4]+[\{[1,2],4\},3]-[\{[1,4],2\},3]+[\{[2,3],1\},4]\\
&+[[\{1,2\},3],4]-[[\{1,2\},4],3]+[[\{1,4\},2],3]-[[\{2,3\},1],4]\\
&-\{\{\{1,2\},3\},4\}+\{\{\{1,2\},4\},3\}+\{\{\{1,4\},2\},3\}-\{\{\{2,3\},1\},4\}\\
&-\{\{[1,2],3\},4\}-\{\{[1,3],4\},2\}-\{\{[3,4],1\},2\}+\{\{[3,4],2\},1\}\\
&-\{[\{1,3\},2],4\}-\{[\{1,3\},4],2\}+\{[\{2,3\},1],4\}+\{[\{2,3\},4],1\}\\
&\qquad -\{[\{2,4\},3],1\}+\{[\{3,4\},1],2\}\\
&+[\{\{1,2\},3\},4]+[\{\{1,2\},4\},3]+[\{\{1,3\},2\},4]+[\{\{1,3\},4\},2]\\
&\qquad +[\{\{1,4\},2\},3]+[\{\{1,4\},3\},2]+[\{\{2,4\},1\},3]+[\{\{3,4\},1\},2]=0.
\end{aligned}
\quad (32)
$$

Its $4! = 24$ permutations gives set of all proper sJacobis at $n = 4$ lying in $\mathcal{R}(S_4)$. All told we have 72 sJacobi relations between (31) and (32) and their permutations.[13]

**Note added:** In the upcoming publication [38] we construct a canonical basis of $n!$ nested correlators such that the sJacobi relations are already implemented. This ensures that every Wightman $n$-point function can be expressed in terms of these without the need to explicitly construct sJacobi relations.

# 4 $k$-OTO Keldysh rules

We next turn to the question of relating a $k$-OTO correlation function, which we view as being obtained from insertions of operators $\mathbb{O}_{\mathrm{R}}^{\alpha}$ and $\mathbb{O}_{\mathrm{L}}^{\alpha}$, to the physical observables. This requires

---

[13] By standard representation theory, the regular representation of $S_4$ breaks up into irreps as $\mathbf{24} = \mathbf{1} + \mathbf{1}' + 2 \times \mathbf{2} + 3 \times \mathbf{3} + 3 \times \mathbf{3}'$. We can then recombine permutations of master sJacobi to get identities transforming in the irreps (as was done for $n = 3$).

that we map the $(2k)^n$ correlation functions in the $2k$-fold extended operator algebra. To this end we first ask how to map the $(2k)^n$ correlators obtained from the $k$-OTO contour onto the set of $2^{n-2} n!$ nested correlation functions. This will simultaneously provide us a map from the observables in the $2k$-fold tensor product algebra onto the operator algebra acting on $\mathcal{H}$,

For the case of Schwinger-Keldysh (1-OTO) correlators it is for instance well known that the average-difference correlation functions can be expressed in terms of nested commutators and anti-commutators of operators with time-ordering step-functions. This procedure goes by the name of Keldysh rules [9]. We will derive analogous expressions for the higher OTOs simply by iterating the standard discussion for the 1-OTO case. Given that the LR correlators are obtained by a simple linear combination of the Av-Dif correlation functions, it is then a simple matter to carry out a basis rotation to extract a map from the LR correlators to the nested observables.

### 4.1 Preliminaries: Keldysh basis and notation

In order to write down the expressions of interest we need to take care of some minor technicalities and introduce some notion. Firstly, to keep track of time ordering we will employ step-functions and adhere to the conventions described in [8]. We define $\Theta_{\mathbb{A}\mathbb{B}} = \Theta_{\mathbb{A}>\mathbb{B}}$ to be unity $\mathbb{A}$ lies in the causal future of $\mathbb{B}$ and vanishing if it is the causal past. Similarly we define $\Theta_{\mathbb{B}\mathbb{A}} = \Theta_{\mathbb{A}<\mathbb{B}}$. Should the causal relation be indeterminate we democratically decide to fix $\Theta_{\mathbb{A}\mathbb{B}} = \Theta_{\mathbb{B}\mathbb{A}} = \frac{1}{2}$. These functions satisfy the normalization condition $\Theta_{\mathbb{A}\mathbb{B}} + \Theta_{\mathbb{B}\mathbb{A}} = 1$.

Multi-argument step functions are easily obtained by stringing together products of these basic step functions. We find it useful to define combinations for both time-ordering and anti-time-ordering as follows:

$$
\begin{aligned}
\Theta_{\mathbb{A}_1\cdots\mathbb{A}_n} &= \Theta_{\mathbb{A}_1>\mathbb{A}_2}\Theta_{\mathbb{A}_2>\mathbb{A}_3}\cdots\Theta_{\mathbb{A}_{n-1}>\mathbb{A}_n} = \Theta_{\mathbb{A}_1\mathbb{A}_2}\Theta_{\mathbb{A}_2\mathbb{A}_3}\cdots\Theta_{\mathbb{A}_{n-1}\mathbb{A}_n}\,, \\
\overline{\Theta}_{\mathbb{A}_1\cdots\mathbb{A}_n} &= \Theta_{\mathbb{A}_1<\mathbb{A}_2}\Theta_{\mathbb{A}_2<\mathbb{A}_3}\cdots\Theta_{\mathbb{A}_{n-1}<\mathbb{A}_n} = \Theta_{\mathbb{A}_n\mathbb{A}_{n-1}}\cdots\Theta_{\mathbb{A}_3\mathbb{A}_2}\Theta_{\mathbb{A}_2\mathbb{A}_1}\,.
\end{aligned}
\tag{33}
$$

The normalization condition for these is that the sum of all permutations of the arguments of the multi step-functions is unity i.e.,

$$
\sum_{\sigma\in S_n}\Theta_{\mathbb{A}_{\sigma(1)}\cdots\mathbb{A}_{\sigma(n)}} = \sum_{\sigma\in S_n}\overline{\Theta}_{\mathbb{A}_{\sigma(1)}\cdots\mathbb{A}_{\sigma(n)}} = 1\,.
\tag{34}
$$

Note that the normalization condition involves a sum over all $n!$ permutation of the $n$-labels and we have used $\sigma$ to denote the element of the symmetric group $S_n$. To retain the spirit of the discussion of [8], we recall our definition of the graded commutator and anti-commutator defined there. They are simply usual commutators and anti-commutators with an additional sign that accounts for the Grassmann statistics of our operators and were defined by[14]

$$
\begin{aligned}
\left[\widehat{\mathbb{A}},\widehat{\mathbb{B}}\right]_{\pm} &= \widehat{\mathbb{A}}\,\widehat{\mathbb{B}} - (-)^{\widehat{\mathbb{A}}\widehat{\mathbb{B}}}\,\widehat{\mathbb{B}}\,\widehat{\mathbb{A}}\,, \\
\left\{\widehat{\mathbb{A}},\widehat{\mathbb{B}}\right\}_{\pm} &= \frac{1}{2}\left(\widehat{\mathbb{A}}\,\widehat{\mathbb{B}} + (-)^{\widehat{\mathbb{A}}\widehat{\mathbb{B}}}\,\widehat{\mathbb{B}}\,\mathbb{A}\right)\,.
\end{aligned}
\tag{35}
$$

The other element we need is the generalized Keldysh rotation given in (8). The latter, we recall, defines the Keldysh basis of Av-Dif operators, which we reproduce here for convenience:

---

[14] The factor of half in the definition of the graded anticommutator is useful to prevent proliferation of factors of 2 in subsequent manipulations. Readers should exercise care when using this definition for the sJacobi identities of §3. The latter, we recall, were computed with the conventional definition of the anti-commutator.

$$\mathbb{O}_{av}^{\alpha} \equiv \frac{1}{2}\left(\mathbb{O}_{R}^{\alpha} + \mathbb{O}_{L}^{\alpha}\right) \equiv \underbrace{\left(\tfrac{1}{2} \quad \tfrac{1}{2}\right)}_{\xi^z}\underbrace{\begin{pmatrix}\mathbb{O}_{R}^{\alpha}\\\mathbb{O}_{L}^{\alpha}\end{pmatrix}}_{\mathbb{O}_z^{\alpha}} = \xi^z \mathbb{O}_z^{\alpha},$$

$$\mathbb{O}_{dif}^{\alpha} \equiv (-1)^{\alpha+1}\left(\mathbb{O}_{R}^{\alpha} - \mathbb{O}_{L}^{\alpha}\right) \equiv (-1)^{\alpha+1}\underbrace{\left(1 \quad -1\right)}_{\eta}\begin{pmatrix}\mathbb{O}_{R}^{\alpha}\\\mathbb{O}_{L}^{\alpha}\end{pmatrix} = \eta^z\,\mathbb{O}_z^{\alpha}. \tag{36}$$

The constant matrices $\xi = \frac{1}{2}\begin{pmatrix}1 & 1\end{pmatrix}$ and $\eta = \begin{pmatrix}1 & -1\end{pmatrix}$ implement the basis transform.

The difference operators $\mathbb{O}_{dif}^{\alpha}$ are defined with a sign depending on the odd/even parity of $\alpha$ to account for the fact that while the odd numbered contours are time-ordered (the right contour precedes the left), the even numbered contours are anti time-ordered (the left contour is encountered first).

Finally, we introduce the *Keldysh bracket* $(\,\cdot\,,\cdot\,)_{SK}$ which [9] maps the average and difference operators to their counterparts in the underlying single-copy operator algebra, suitably combined into graded commutators and anti-commutators. To wit,

$$\left(\widehat{\mathbb{A}}\,,\mathbb{B}_{dif}\right)_{SK} \equiv \widehat{\mathbb{A}}\,\widehat{\mathbb{B}} - (-)^{\mathbb{A}\mathbb{B}}\,\widehat{\mathbb{B}}\,\widehat{\mathbb{A}} \equiv \left[\widehat{\mathbb{A}},\widehat{\mathbb{B}}\right]_{\pm},$$

$$\left(\widehat{\mathbb{A}}\,,\mathbb{B}_{av}\right)_{SK} \equiv \frac{1}{2}\left(\widehat{\mathbb{A}}\,\widehat{\mathbb{B}} + (-)^{\mathbb{A}\mathbb{B}}\,\widehat{\mathbb{B}}\,\widehat{\mathbb{A}}\right) \equiv \left\{\widehat{\mathbb{A}},\widehat{\mathbb{B}}\right\}_{\pm}. \tag{37}$$

In particular, if $\widehat{\mathbb{I}}$ is the identity operator then we have

$$\left(\widehat{\mathbb{I}}\,,\mathbb{A}_{dif}\right)_{SK} = 0\,, \qquad \left(\widehat{\mathbb{I}}\,,\mathbb{A}_{av}\right)_{SK} = \widehat{\mathbb{A}}. \tag{38}$$

While (37) takes Grassmann parity of the operators into account, we will in the following assume for simplicity that all operators are Grassmann-even. Generalizations of the various equations are straightforward to write down.

## 4.2 $k$-OTO correlation functions

With these preliminaries in place let us now consider the following correlation function of contour-ordered operators

$$\begin{aligned}
G_{k-oto}(t_1, t_2, \cdots t_{n_k}) =&\langle\mathcal{T}_{\mathcal{C}}\left(\mathbb{O}_{av}^1(1)\ldots\mathbb{O}_{av}^1(m_1)\,\mathbb{O}_{dif}^1(m_1+1)\ldots\mathbb{O}_{dif}^1(n_1)\right)\\
&\times\left(\mathbb{O}_{av}^2(n_1+1)\ldots\mathbb{O}_{av}^2(m_2)\,\mathbb{O}_{dif}^2(m_2+1)\ldots\mathbb{O}_{dif}^2(n_2)\right)\times\cdots \\
&\cdots\times\left(\mathbb{O}_{av}^k(n_{k-1}+1)\ldots\mathbb{O}_{av}^k(m_k)\,\mathbb{O}_{dif}^k(m_k+1)\ldots\mathbb{O}_{dif}^k(n_k)\right)\rangle.
\end{aligned} \tag{39}$$

In writing the above, we have employed a short-hand notation where the argument of the operator both indexes the operator and its time argument, viz., $\mathbb{O}_{av,m}^{\alpha}(t_m) \equiv \mathbb{O}_{av}^{\alpha}(m)$ and analogously for $dif$-type operators.

We claim that this correlation function can be expressed using the Keldysh bracket as a sequence of nested (graded) commutators and anti-commutators with suitable dressing by the time ordering step functions. To write readable expressions, we now go even further in compactifying notation by introducing indices $\boldsymbol{\alpha} \in \{av, dif\}$ labelling the contour type. We can then write each operator in (39) as a symbol of the form $\mathbb{O}_{\boldsymbol{\alpha}_m,m}^{\alpha}(t_m) \equiv \mathbb{O}_{\boldsymbol{\alpha}_m}^{\alpha}(m) \equiv \mathbb{O}_m^{\alpha}$. We then define the *generalized Keldysh rule*, which computes (39) in terms of Keldysh brackets

(and hence in terms of nested correlators). We start with the innermost segment $\alpha = 1$ and work our way outward in the trace iteratively, viz.,

$$
\begin{aligned}
& G_{k-oto}(t_1, t_2, \cdots t_{n_k}) \\
&= \sum_{\sigma_1 \in S_{n_1}} \Theta_{\sigma_1(1) \cdots \sigma_1(n_1)} \langle (\cdots (\mathcal{X}_2, \mathbb{O}^1_{\sigma_1(1)})_{SK}, \cdots \mathbb{O}^2_{\sigma_1(n_1)})_{SK} \rangle \\
&= \sum_{\sigma_1 \in S_{n_1}} \Theta_{\sigma_1(1) \cdots \sigma_1(n_1)} \sum_{\sigma_2 \in S_{n_2-n_1}} \overline{\Theta}_{\sigma_2(n_1+1) \cdots \sigma_2(n_2)} \\
& \quad \times \langle (\cdots (\cdots (\cdots (\mathcal{X}_3, \mathbb{O}^2_{\sigma_2(n_1+1)})_{SK}, \cdots \mathbb{O}^2_{\sigma_2(n_2)})_{SK}, \cdots \mathbb{O}^1_{\sigma_1(1)})_{SK}, \cdots \mathbb{O}^1_{\sigma_1(n_1)})_{SK} \rangle .
\end{aligned}
\tag{40}
$$

In the above, we have introduced $\mathcal{X}_j$ which denotes the set of operators inserted in contours indexed by $\alpha = j, j+1, \cdots, k$, and permutations thereof.

We have shown how the iterative scheme works by exhibiting the first two levels; the first line takes care of $\alpha = 1$, and in the second line we carry out the extension to the second level $\alpha = 2$. As we have only made explicit the operators in the first two segments we have at the innermost level of the nesting $\mathcal{X}_3$ which captures all the operators with $j \geq 3$. Recursively, we have for $j = 1, \ldots, k$:

$$
\mathcal{X}_j = \begin{cases}
\displaystyle \sum_{\sigma_j \in S_{n_j - n_{j-1}}} \Theta_{\sigma_j(n_{j-1}+1) \cdots \sigma_j(n_j)} (\cdots (\mathcal{X}_{j+1}, \mathbb{O}^j_{\sigma_j(n_{j-1}+1)})_{SK}, \cdots \mathbb{O}^j_{\sigma_j(n_j)})_{SK} & (j \text{ odd}) \\
\displaystyle \sum_{\sigma_j \in S_{n_j - n_{j-1}}} \overline{\Theta}_{\sigma_j(n_{j-1}+1) \cdots \sigma_j(n_j)} (\cdots (\mathcal{X}_{j+1}, \mathbb{O}^j_{\sigma_j(n_{j-1}+1)})_{SK}, \cdots \mathbb{O}^j_{\sigma_j(n_j)})_{SK} & (j \text{ even})
\end{cases}
$$
$$
\mathcal{X}_{k+1} = \mathbb{1}.
\tag{41}
$$

We first apply the Keldysh rules as stated to the segment $\alpha = 1$, then apply nested within this the rules for $\alpha = 2$. The new wrinkle is the even index contours are anti-time-ordered and hence we see the appearance of $\overline{\Theta}$ for $\alpha = 2$. The procedure continues till we reach the $k^{\text{th}}$ level. We will give explicit examples in what follows. The argument deriving this is very similar to the implementation of the Keldysh rules in [9] and is described in Appendix A.

## 4.3 Exemplifying Keldysh rules

Let us first consider the situation for two and three point functions using 2-OTOs and then generalize to give expressions for other cases.[15]

**Two-point functions:** The space of two-point functions is completely captured by the Schwinger-Keldysh 1-OTO correlation functions. This case has already been described in [8] but we will use it to first illustrate the general ideas explained above.

To get oriented let us record the standard expressions for two-point functions that are well known from the Schwinger-Keldysh formalism. We can obtain these by simply placing all operators in the first contour $\alpha = 1$. Applying the Keldysh rule (40) we find

$$
\begin{aligned}
\langle \mathcal{T}_{\mathcal{C}} \, \mathbb{O}^1_{av}(1) \mathbb{O}^1_{av}(2) \rangle &= \langle \{ \widehat{\mathbb{O}}(1), \widehat{\mathbb{O}}(2) \}_{\pm} \rangle, \\
\langle \mathcal{T}_{\mathcal{C}} \, \mathbb{O}^1_{av}(1) \mathbb{O}^1_{dif}(2) \rangle &= \Theta_{12} \, \langle [ \widehat{\mathbb{O}}(1), \widehat{\mathbb{O}}(2) ]_{\pm} \rangle, \\
\langle \mathcal{T}_{\mathcal{C}} \, \mathbb{O}^1_{dif}(1) \mathbb{O}^1_{av}(2) \rangle &= \langle \mathcal{T}_{\mathcal{C}} \, \mathbb{O}^1_{av}(2) \mathbb{O}^1_{dif}(1) \rangle = -\Theta_{21} \, \langle [ \widehat{\mathbb{O}}(1), \widehat{\mathbb{O}}(2) ]_{\pm} \rangle, \\
\langle \mathcal{T}_{\mathcal{C}} \, \mathbb{O}^1_{dif}(1) \mathbb{O}^1_{dif}(2) \rangle &= 0.
\end{aligned}
\tag{42}
$$

---

[15] To avoid proliferation of subscripts and superscripts, we will in this subsection use the notation $\widehat{\mathbb{O}}_j(t_j) \equiv \widehat{\mathbb{O}}(j)$ and similarly for the Av-Ret operators.

As promised there are only two linearly independent correlators corresponding to the temporal ordering $t_1 > t_2$ and $t_2 > t_1$. We can conveniently pick a basis of 2-point functions to be given by symmetrized Keldysh correlator and the commutator.

$$
\begin{aligned}
\langle \mathcal{T}_{\mathcal{C}}\, \mathbb{O}^1_{av}(1)\mathbb{O}^1_{av}(2)\rangle &= \langle \{\widehat{\mathbb{O}}(1),\widehat{\mathbb{O}}(2)\}_{\pm}\rangle\,, \\
\langle \mathcal{T}_{\mathcal{C}}\left(\mathbb{O}^1_{av}(1)\mathbb{O}^1_{dif}(2) - \mathbb{O}^1_{dif}(1)\mathbb{O}^1_{av}(2)\right)\rangle &= \langle [\widehat{\mathbb{O}}(1),\widehat{\mathbb{O}}(2)]_{\pm}\rangle\,.
\end{aligned}
\tag{43}
$$

Let us try to examine what happens if we evaluate 2-point functions using the 2-OTO contour instead. Now we have two segments $\alpha = 1, 2$ where we can insert the operators. This leads to various cross-contour possibilities; these however have no new information as all the desired two-point functions can be computed using the Schwinger-Keldysh 1-OTO contour. We have explicitly

$$
\begin{aligned}
\langle \mathcal{T}_{\mathcal{C}}\, \mathbb{O}^1_{av}(1)\mathbb{O}^2_{av}(2)\rangle &= \langle \{\widehat{\mathbb{O}}(1),\widehat{\mathbb{O}}(2)\}_{\pm}\rangle\,, \\
\langle \mathcal{T}_{\mathcal{C}}\, \mathbb{O}^1_{dif}(1)\mathbb{O}^2_{av}(2)\rangle &= \langle [\widehat{\mathbb{O}}(2),\widehat{\mathbb{O}}(1)]_{\pm}\rangle\,, \\
\langle \mathcal{T}_{\mathcal{C}}\, \mathbb{O}^1_{av}(1)\mathbb{O}^2_{dif}(2)\rangle &= \langle \mathcal{T}_{\mathcal{C}}\, \mathbb{O}^1_{dif}(1)\mathbb{O}^2_{dif}(2)\rangle = 0\,,
\end{aligned}
\tag{44}
$$

which reduced to the basis correlators (43). Likewise the correlators of operators inserted on the second contour $\alpha = 2$ lead to

$$
\begin{aligned}
\langle \mathcal{T}_{\mathcal{C}}\, \mathbb{O}^2_{av}(1)\mathbb{O}^2_{av}(2)\rangle &= \langle \{\widehat{\mathbb{O}}(1),\widehat{\mathbb{O}}(2)\}_{\pm}\rangle\,, \\
\langle \mathcal{T}_{\mathcal{C}}\, \mathbb{O}^2_{av}(1)\mathbb{O}^2_{dif}(2)\rangle &= \Theta_{21}\langle [\widehat{\mathbb{O}}(1),\widehat{\mathbb{O}}(2)]_{\pm}\rangle\,, \\
\langle \mathcal{T}_{\mathcal{C}}\, \mathbb{O}^2_{dif}(1)\mathbb{O}^2_{av}(2)\rangle &= \langle \mathcal{T}_{\mathcal{C}}\, \mathbb{O}^2_{av}(2)\mathbb{O}^2_{dif}(1)\rangle = -\Theta_{12}\langle [\widehat{\mathbb{O}}(1),\widehat{\mathbb{O}}(2)]_{\pm}\rangle\,, \\
\langle \mathcal{T}_{\mathcal{C}}\, \mathbb{O}^2_{dif}(1)\mathbb{O}^2_{dif}(2)\rangle &= 0\,.
\end{aligned}
\tag{45}
$$

The only feature of interest here is the fact that the time-ordering is reversed.

The reason behind the simplification in the two-point functions owes to the fact that the there is a large degree of redundancy built into the $k$-OTO contour as presaged in §2.

**Three-point Functions:** Let us now consider 3-point functions. We have seen that the Wightman basis has $3! = 6$ elements, while the Av-Dif correlators are enumerated by the $k$-OTO contour to be $(2k)^3$. The nested correlators however are $2 \times 3! = 12$ in number. We will later give an explicit embedding of all the Av-Dif correlators for $k = 2$ in §7.

For now we note the following: we can collapse the nested correlators to a simpler set stripping off the permutations. Say we have three operators $\widehat{\mathbb{O}}(1)$, $\widehat{\mathbb{O}}(2)$ and $\widehat{\mathbb{O}}(3)$. We can pick the first two and decide to pair them into a commutator or an anti-commutator. Following this choice, we can take the composite object thus constructed and pair it with the third operator into another commutator or anti-commutator. There are 2 partitions involved with a pair of choices for each partitioning leading to $2^2 = 4$ choices. The remaining choices are obtained by permuting the operators. Sticking to this limited set, and examining the Keldysh rules we derive a simple set of identities:

$$
\begin{aligned}
\langle \{\{\widehat{\mathbb{O}}(1),\widehat{\mathbb{O}}(2)\}_{\pm},\widehat{\mathbb{O}}(3)\}_{\pm}\rangle &= \langle \mathcal{T}_{\mathcal{C}}\, \mathbb{O}^2_{av}(1)\mathbb{O}^2_{av}(2)\mathbb{O}^1_{av}(3)\rangle\,, \\
\langle [\{\widehat{\mathbb{O}}(1),\widehat{\mathbb{O}}(2)\}_{\pm},\widehat{\mathbb{O}}(3)]_{\pm}\rangle &= \langle \mathcal{T}_{\mathcal{C}}\, \mathbb{O}^2_{av}(1)\mathbb{O}^2_{av}(2)\mathbb{O}^1_{dif}(3)\rangle\,, \\
\langle \{[\widehat{\mathbb{O}}(1),\widehat{\mathbb{O}}(2)]_{\pm},\widehat{\mathbb{O}}(3)\}_{\pm}\rangle &= \langle \mathcal{T}_{\mathcal{C}}\left(\mathbb{O}^2_{av}(1)\mathbb{O}^2_{dif}(2) - \mathbb{O}^2_{dif}(1)\mathbb{O}^2_{av}(2)\right)\mathbb{O}^1_{av}(3)\rangle\,, \\
\langle [[\widehat{\mathbb{O}}(1),\widehat{\mathbb{O}}(2)]_{\pm},\widehat{\mathbb{O}}(3)]_{\pm}\rangle &= \langle \mathcal{T}_{\mathcal{C}}\left(\mathbb{O}^2_{av}(1)\mathbb{O}^2_{dif}(2) - \mathbb{B}_{dif}(1)\mathbb{O}^2_{av}(2)\right)\mathbb{O}^1_{dif}(3)\rangle\,.
\end{aligned}
\tag{46}
$$

### 4.4 Simplifying Keldysh basis correlators

To give compact expressions for all the correlators in the Keldysh basis in terms of nested commutators and anti-commutators we employ the following simplifying notational device.

- Fix the time ordering to be $t_1 > t_2 > t_3 > \cdots t_n$.

- Let the average and difference operators be indexed by a set of binary valued symbols $\{\alpha, \beta, \gamma\}$ etc..

$$\alpha, \beta, \gamma \in \{av, dif\}. \tag{47}$$

- Introduce a bracket $(\!(\cdot, \cdot)\!)_\alpha$ and a binary constant $\mathfrak{d}_\alpha$

$$(\!(\mathbb{A}, \mathbb{B})\!)_\alpha = \left\{ \begin{array}{ll} [\mathbb{A}, \mathbb{B}]_\pm, & \alpha = av \\ \{\mathbb{A}, \mathbb{B}\}_\pm, & \alpha = dif \end{array} \right., \qquad \mathfrak{d}_\alpha = \left\{ \begin{array}{ll} 1, & \alpha = av \\ 0, & \alpha = dif \end{array} \right.. \tag{48}$$

We claim that armed with this notation we can succinctly encode all the relations obtained by employing the $k$-OTO Keldysh rules. Consider the 2-OTO results for the 2-point functions. We can simply write

$$\begin{aligned} \langle \mathcal{T}_{\mathcal{C}}\, \mathbb{O}_\gamma^\alpha(1)\mathbb{O}_\alpha^1(2) \rangle &= \mathfrak{d}_\gamma \langle (\!(\widehat{\mathbb{O}}(1), \widehat{\mathbb{O}}(2))\!)_\alpha \rangle, \\ \langle \mathcal{T}_{\mathcal{C}}\, \mathbb{O}_\alpha^\alpha(1)\mathbb{O}_\gamma^2(2) \rangle &= \mathfrak{d}_\gamma \langle (\!(\widehat{\mathbb{O}}(2), \widehat{\mathbb{O}}(1))\!)_\alpha \rangle. \end{aligned} \tag{49}$$

The reader can verify that this captures all $2^4$ correlators that have been described earlier. The counting follows since $\alpha = 1, 2$ while $\alpha$ and $\gamma$ are each binary valued.

It is easy to generalize this to higher OTO 2-point functions. For the $k$-OTO theory the reader can verify the relations:

$$\begin{aligned} \langle \mathcal{T}_{\mathcal{C}}\, \mathbb{O}_\gamma^\alpha(1)\mathbb{O}_\alpha^\beta(2) \rangle &= \mathfrak{d}_\gamma \langle (\!(\widehat{\mathbb{O}}(1), \widehat{\mathbb{O}}(2))\!)_\alpha \rangle, & \alpha &> \beta, \\ \langle \mathcal{T}_{\mathcal{C}}\, \mathbb{O}_\gamma^\alpha(1)\mathbb{O}_\alpha^\alpha(2) \rangle &= \mathfrak{d}_\gamma \langle (\!(\widehat{\mathbb{O}}(1), \widehat{\mathbb{O}}(2))\!)_\alpha \rangle, & \alpha &= 2k+1, \\ \langle \mathcal{T}_{\mathcal{C}}\, \mathbb{O}_\alpha^\alpha(1)\mathbb{O}_\gamma^\alpha(2) \rangle &= \mathfrak{d}_\gamma \langle (\!(\widehat{\mathbb{O}}(2), \widehat{\mathbb{O}}(1))\!)_\alpha \rangle, & \alpha &= 2k, \\ \langle \mathcal{T}_{\mathcal{C}}\, \mathbb{O}_\alpha^\alpha(1)\mathbb{O}_\gamma^\beta(2) \rangle &= \mathfrak{d}_\gamma \langle (\!(\widehat{\mathbb{O}}(2), \widehat{\mathbb{O}}(1))\!)_\alpha \rangle, & \alpha &< \beta. \end{aligned} \tag{50}$$

These capture all $4k^2$ 2-point correlation functions which can be computed from this contour. Similar results hold for higher point functions. For explicit expressions in case of $n = 3, 4$ we refer to the examples in §6.3.

## 5 LR correlators and the Wightman basis

Our final remaining task is to show how to map the $(2k)^n$ correlation functions that can be evaluated from a $k$-OTO contour onto the Wightman basis. We have outlined the basic decomposition in §2.3, and now give a more detailed explanation of the same. Before getting into the details however, let us first record here the general results which we derive below:

- To obtain all of the $n!$ Wightman basis elements, we need to consider proper $q$-OTO contours, with $q = 1, 2, \cdots, \lfloor \frac{n+1}{2} \rfloor$.

- A proper $q$-OTO computes $g_{n,q}$ of these basis elements where

$$\begin{aligned} g_{n,q} &= \text{Coefficient of } \mu^q \text{ in } \mathcal{G}_n(\mu), \\ \mathcal{G}_n(\mu) &\equiv \left( 2\sqrt{1-\mu} \right)^{n+1} \text{Li}_{-n}\!\left( \frac{2}{1 + \sqrt{1-\mu}} - 1 \right), \end{aligned} \tag{51}$$

where $\text{Li}_{-n}(z) \equiv \sum_{k=1}^\infty k^n z^k$ is the polylogarithm function.

- In a $k$-OTO contour, each of these $g_{n,q}$ proper $q$-OTO correlators can be represented in $h_{n,k}^{(q)}$ ways. It should be apparent that $h_{n,k}^{(q)} = 0$ for $k < q$ or $n < 2q-1$. For larger values of $q$, there is a non-trivial degeneracy, which can be obtained from the generating function

$$\mathcal{H}_q(z,t) = \sum_{n=2q-1}^{\infty} \sum_{k=q}^{\infty} h_{n,k}^{(q)} z^n t^k = \left(\frac{2z}{1-t}\right)^{2q-1} \frac{t^q}{1-(z+t+zt)}. \tag{52}$$

It is also convenient to consider a related generating function

$$\mathcal{H}_{q,n}(t) = \sum_{k=q}^{\infty} h_{n,k}^{(q)} t^k = 2^{2q-1} t^q \frac{(1+t)^{n-(2q-1)}}{(1-t)^{n+1}} \Theta(n-(2q-1)), \tag{53}$$

which will play a role in the course of our analysis below. Thus, we finally note that the degeneracy factor for a each particular Wightman basis element computed on a proper $q$-OTO contour when embedded in a $k$-OTO contour is given by

$$\begin{aligned} h_{n,k}^{(q)} &= \text{Coefficient of } z^n t^k \text{ in } \left(\frac{2z}{1-t}\right)^{2q-1} \frac{t^q}{1-(z+t+zt)} \\ &= \text{Coefficient of } t^k \text{ in } 2^{2q-1} t^q \frac{(1+t)^{n-(2q-1)}}{(1-t)^{n+1}} \Theta(n-2q+1). \end{aligned} \tag{54}$$

- As a simple check to ascertain the veracity of these statements, we enumerate all the $k$-OTO correlators, for:

$$\begin{aligned} \sum_{q=1}^{\lfloor \frac{n+1}{2} \rfloor} g_{n,q} h_{n,k}^{(q)} &= \text{Coefficient of } t^k \text{ in } \sum_{q=1}^{\lfloor \frac{n+1}{2} \rfloor} g_{n,q} 2^{2q-1} t^q \frac{(1+t)^{n-(2q-1)}}{(1-t)^{n+1}} \\ &= \text{Coefficient of } t^k \text{ in } \frac{1}{2}\left(\frac{1+t}{1-t}\right)^{n+1} \times \sum_{q=1}^{\lfloor \frac{n+1}{2} \rfloor} g_{n,q} \left(\frac{4t}{(1+t)^2}\right)^q \\ &= \text{Coefficient of } t^k \text{ in } \frac{1}{2}\left(\frac{1+t}{1-t}\right)^{n+1} \mathcal{G}_n(\mu) \Big|_{\mu=\frac{4t}{(1+t)^2}} \\ &= \text{Coefficient of } t^k \text{ in } 2^n \text{Li}_{-n}(t) = (2k)^n. \end{aligned} \tag{55}$$

- Thus, we see that the $g_{n,q}$ proper $q$-OTO correlator each occurring $h_{n,k}^{(q)}$ times accounts for all the $(2k)^n$ contour correlators, as indicated. A few low-lying values are provided in Tables 1 and 2, respectively.

## 5.1 Canonical representation of time-ordering correlator

In order to describe the results above, we will need to understand various facets of the $k$-OTO contour. Some of the salient features have been partly summarized in §2.4; these will prove helpful in streamlining the argument.

The first task at hand is to identify the minimal presentation of a timefolded contour that computes for us a particular element of the Wightman basis. We will refer to this as the canonical form of the contour for a given correlator. Say we are interested in a specific element $G_\sigma$, which is described by a permutation $\sigma \in S_n$ of the $n$-labels. We pick a time interval ranging from the initial time when the density matrix is prepared $t_0$ to the largest time $t_1$, and

Table 1: The decomposition of the $n!$ Wightman basis correlators into the proper $q$-OTO correlators for low-lying values of $n$.

| $g_{n,q}$ | $q = 1$ | 2 | 3 |
|---|---|---|---|
| $n = 1$ | 1 | 0 | 0 |
| 2 | 2 | 0 | 0 |
| 3 | 4 | 2 | 0 |
| 4 | 8 | 16 | 0 |
| 5 | 16 | 88 | 16 |
| 6 | 32 | 416 | 272 |

Table 2: The degeneracies encountered in embedding a particular $n$-point function of proper $q$-OTO type, into a generic $k$-OTO contour for low-lying values of $n$.

| $h^{(q)}_{n,k}$ | $q = 1$ | 2 | 3 |
|---|---|---|---|
| $n = 1$ | $2k$ | 0 | 0 |
| 2 | $2k^2$ | 0 | 0 |
| 3 | $\frac{2}{3}k(2k^2+1)$ | $\frac{4}{3}k(k^2-1)$ | 0 |
| 4 | $\frac{2}{3}k^2(k^2+2)$ | $\frac{2}{3}k^2(k^2-1)$ | 0 |
| 5 | $\frac{2}{15}k(2k^4+10k^2+3)$ | $\frac{4}{15}k(k^4-1)$ | $\frac{4}{15}k(k^2-1)(k^2-4)$ |
| 6 | $\frac{2}{45}k^2(2k^4+20k^2+23)$ | $\frac{4}{45}k^2(2k^4+5k^2-7)$ | $\frac{4}{45}k^2(k^2-1)(k^2-4)$ |

subdivide this at the operator insertion locations $t_i$, with $i = 2, 3, \cdots, n$. One can then insert the operators at their appropriate temporal locale, and insert a forward/backward switchback whenever we need to reverse the flow of time from one operator to the next. This is easy to do pictorially and the reader is invited to try out various examples given our contour conventions in Fig. 2.

One can give a more compact abstract symbolic representation, which may be useful to build intuition. Given a permutation $\sigma \in S_n$, consider the string $\sigma(1)\sigma(2)\sigma(3)\cdots\sigma(n)$. It suffices for some purposes to insert a future turn, denoted ) or a past turn denoted ( into this string. The density matrix itself will be denoted by ∘ at either end of the string. For example a time-ordered four-point function would simply be ∘4321)∘ while an anti-time ordered one would be ∘)1234∘. The 2-OTO correlator $G(t_4, t_1, t_3, t_2)$ for instance would then be written as

$$G(t_4, t_1, t_3, t_2) = \circ 2)3(1)4\circ \equiv$$



$$t_4 \ t_3 \quad t_2 \ t_1$$

The main drawback about this representation is that it requires a moment's thought to visualize the temporal ordering, which is more clearly manifest in the contour picture. However, with the understanding that we will always assume $t_1 > t_2 > t_3 > \cdots t_n$, this shorthand notation captures all the information about the time-ordering structure of a correlator.

However, already at this level we can see that there are certain degeneracies involved in the representation of the correlators. A time-ordered correlator can be given a 1-OTO representation as in ∘4321)∘, but also as a 2-OTO, e.g., ∘)(4321)∘, or indeed as any other $k$-OTO. This simply follows from the fact that unitarity allows us to concatenate away the string )( us-

ing $U U^\dagger = \mathbb{1}$. The canonical presentation of the correlator is one which has minimal number of timefolds, which prompts thence our notion of the *proper OTO contour*.

A proper OTO is one where all the switchbacks of the timefolded contour are necessary to preserve the temporal ordering of the correlator. In particular, no pair of the segments of a proper OTO contour can be contracted away by unitarity. It is important to note that while we can find a proper OTO presentation of a given $G_\sigma$, this is not necessarily unique, since one may still have the freedom to slide the operators without changing the OTO number. For instance, correlator $G(t_4, t_1, t_3, t_2)$, which has a canonical representation as a 2-OTO, can have multiple realizations, e.g., either as ∘2)3(1)4∘, or as ∘2)(3)14∘, which are related by sliding the operators around (cf., Fig. 3 later).

Once we understand this idea of the proper OTO representation, it is immediately clear that the we need to consider no more than $\lfloor \frac{n+1}{2} \rfloor$ timefolds. A given $q$-OTO contour, has $q$ future turning points ), and $q-1$ past turns (, leading to an insertion of $2q-1$ switchbacks between the operators. Amongst permutations of $n$ operators, we can at most encounter a completely oscillating or tremelo permutation, which is a sequence where insertion times alternately increase/decrease along the contour. Counting the future/past turns we can insert in this sequence gives us the maximum proper $q$-OTO number.

## 5.2 Time ordering correlators from proper $q$-OTO

We now understand how to canonically represent a given time-ordering correlator $G_\sigma$ as a proper $q$-OTO contour. Let us then ask given a proper $q$-OTO contour, with $q = 1, 2, \cdots, \lfloor \frac{n+1}{2} \rfloor$, how many of the correlators, call it $g_{n,q}$, can be realized on it?

It is easy to come up with an argument for the counting by proceeding inductively. To be concrete, let us assume that we have been handed a proper OTO for $(n-1)$-point function. We now wish to add an additional operator, so that we upgrade this to an $n$-point correlator. We can assume that that last operator we add is to the future of the original $(n-1)$ operators. We are interested in inserting this future-most operator it in such a way that the proper OTO number of the resultant $n$-point correlator is not more than the given number $q$. There are two ways to do this:

- The first way is to start with a $(n-1)$-point correlator whose proper OTO number is less than $q$ and then increase it. A little thought reveals that an insertion of one operator cannot increase the proper OTO number by more than one. Thus, we can insert the new operator arbitrarily into any one of the $n$ intervals that exist between previous $(n-1)$ insertions. Since the operator we are inserting is the future-most, many of these insertions need us to pull out a timefold out of the intervals into which the future-most operator can be inserted. Sometimes, an additional timefold is not necessary.

  In any case, if we start with the $(n-1)$-point correlator whose proper OTO number is less than $q$, we will end with $n$ different $n-$point correlators whose proper OTO number is less than or equal to $q$.

- The second way to proceed is to start with a proper $q$-OTO $(n-1)$-point correlator and add an operator without increasing the proper OTO number. To see how this can be done recall the definition of the *future turning-point* introduced in §2.4. There are $q$ such future turning-points in a proper $q$-OTO correlator. If the future-most operator is inserted at these future turning-points, then the proper OTO number stays $q$.

  There is however an additional degeneracy to account for. In each of these $q$ future turning-points, the future-most operator can be inserted in two distinct ways: either by keeping the next future-most operator near the future turning-point to be before the

turning-point or after the turning-point. Thus, in total, for every proper $q$-OTO $(n-1)$-point correlator, we can generate $2q$ number of proper $q$-OTO $n$-point correlators

Thus, putting together all these ways of generating an $n$-point correlator whose proper OTO number is less than or equal to $q$ from an $(n-1)$-point correlator whose proper OTO number is less than or equal to $q$, we obtain the following recurrence relation for the number of proper $q$-OTO $n$-point functions denoted by $g_{n,q}$:

$$\sum_{j=1}^{q} g_{n,j} = n \sum_{j=1}^{q-1} g_{n-1,j} + (2q) g_{n-1,q}. \tag{56}$$

In Appendix C.3 we solve this recursion relation and prove the following result:

*Theorem 2:* The number of proper $q$-OTO $n$-point functions is given by

$$\boxed{\begin{aligned} g_{n,q} &= \text{Coefficient of } \mu^q \text{ in } \left(2\sqrt{1-\mu}\right)^{n+1} \text{Li}_{-n}\left(\frac{2}{1+\sqrt{1-\mu}} - 1\right) \\ &= \text{Coefficient of } \frac{z^n}{n!}\mu^q \text{ in } \mu\left\{\left(1 - \frac{\tan(z\sqrt{\mu-1})}{\sqrt{\mu-1}}\right)^{-1} - 1\right\}. \end{aligned}} \tag{57}$$

As a consistency check, using the fact the polylogarithm function has an $n+1$ order pole at at $x = 1$, viz.,

$$\text{Li}_{-n}(x) = \frac{n!}{(1-x)^{n+1}} + \zeta(-n) + \mathcal{O}(1-x),$$

we can immediately see that

$$\sum_{q=1}^{\lfloor\frac{n+1}{2}\rfloor} g_{n,q} = \lim_{\mu\to 1} \mathcal{G}_n(\mu) = n!. \tag{58}$$

This shows that our counting of $g_{n,q}$ does account for all the $n!$ time-orderings.

There are some interesting observations to make about the counts $g_{n,q}$, especially in connection with general properties of permutations.

- Given that we want the proper OTO number to be $q$, and have a set of $n$ objects, the counting problem is the same as counting the subset of $n!$ permutations which have a fixed number of maxima. The latter are nothing but the future turning points, which has been fixed to be $q$. This problem is described combinatorially in [39] and leads to the same result quoted above.[16]

- In the case where we only have turning-point operators, i.e., when $n = 2q-1$, $g_{2q-1,q}$ measure the number of so called tremolo permutations of the turning-point operators (these are permutations which zig-zag to keep the proper OTO number to be $q$). The numbers $g_{2q-1,q}$ are moreover also known as the zag or tangent numbers. The latter name comes from the fact that

$$\tan z = \sum_{q=1}^{\infty} g_{2q-1,q}\frac{z^{2q-1}}{(2q-1)!},$$

$$\frac{\tan z - z}{z^2} = \sum_{q=1}^{\infty} g_{n=2q,q}\frac{z^{2q-1}}{(2q+1)!}. \tag{59}$$

---

[16] The counting problem can also be phrased in terms of enumerating partitions with a given run structure which has been considered in the combinatorics literature earlier; see e.g., [40] for recent developments.

- This identification in turn leads to an expression in terms of Bernoulli numbers:

$$
\begin{aligned}
g_{2q-1,q} &= \frac{2^{2q-1}(2^{2q}-1)}{q}(-1)^{q+1}B_{2q} = \frac{2(2^{2q}-1)}{\pi^{2q}}(2q-1)!\zeta(2q), \\
g_{2q,q} &= \frac{2^{2q+1}(2^{2q+2}-1)}{q+1}(-1)^{q}B_{2q+2} = \frac{2(2^{2q+2}-1)}{\pi^{2q+2}}(2q+1)!\zeta(2q+2),
\end{aligned}
\tag{60}
$$

- There also appears to be an interesting relation between these numbers and the representation theory of $\mathfrak{su}(1,1)$ Lie algebra, [41,42]. This supergroup structure is tantalizing given the observations made in [8] relating to the BRST symmetries inherent in the $k$-OTO (and thus in the proper $q$-OTO) functional integral contours.

### 5.3 $k$-OTOs and degeneracy factor

The discussion above makes clear that insofar as $n$-point functions are concerned, we only need to consider proper $q$-OTOs with $q \leq \lfloor \frac{n+1}{2} \rfloor$. The original question we wanted to address was how to think about these correlation functions when we are given a $k$-OTO generating function. We now turn to the question: given a $k$-OTO functional integral contour, in how many ways can we find an embedding for a particular time-ordering correlation function? In answering this question we will see how to bring the $k$-OTO contour to a canonical form.

To get some intuition, let us begin by looking at some simple cases with the minimum allowed $k$'s (number of contours) and $n$'s (number of operator insertions) to get a proper $q$-OTO correlator.

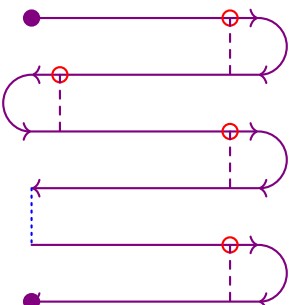

Figure 3: The allowed flips (denoted by vertical dashed lines) that give rise to $2^{2q-1}$ choices of placing the turning-point operators. There are $q$ future turning-point operators and $q-1$ past turning-point operators. Each of them can be chosen to come before or after the turning-point leading to an irreducible degeneracy of $2^{2q-1}$ contour correlators which evaluate to same single time correlator. In the 'canonical' arrangement we fix these $2^{2q-1}$ choices by demanding that the turning-point operators be placed always before the turning-point.

**Case 1** ($k = q$, $n$ **arbitrary**): Consider the case $k = q$, and assuming that the $n$-point function of interest is obtained from a proper $q$-OTO we can ask how many different contours would lead to the same result. We claim that there are as many as:

$$
h^{(q)}_{n,k=q} = 2^{2q-1},
\tag{61}
$$

distinct proper $q$-OTOs which result in the same $n$-point function.

This can be understood straightforwardly: since the proper OTO number coincides with the number of contours $q = k$, the only freedom in such a correlator is to choose the future-most



(past-most) operator on the future (past) turning-points to be on either side of the turning-point. The total number of such turning-points are $2q-1$ and for each turning-point we have 2 choices and hence are led to (61). Fig. 3 shows the $2^{2q-1}$ choices one can make with regards to turning-point operators.

**Case 2 ($n = 2q-1$, $k$ arbitrary):**  Another simple case is the minimal value of $n$ for a fixed $q$, i.e., $n = 2q-1$. In this case we claim

$$h^{(q)}_{n=2q-1,k} = 2^{2q-1} \frac{(k+q-1)!}{(k-q)!(2q-1)!} \, . \tag{62}$$

We first note that when one has a proper $q-$ OTO with $n = 2q-1$ insertions, then the $(2q-1)$ insertions necessarily zig-zag in time, viz., as we move from one insertion to next insertion the time reduces or increases alternately. One way to think about this is that, given $q$ timefolds with $q$ future turning-points and $q-1$ past turning-points, one has an operator which lives near each of these $2q-1$ turning-points. The number $h^{(q)}_{n=2q-1,k}$ counts how many ways this structure can be embedded into $k$ timefolds.

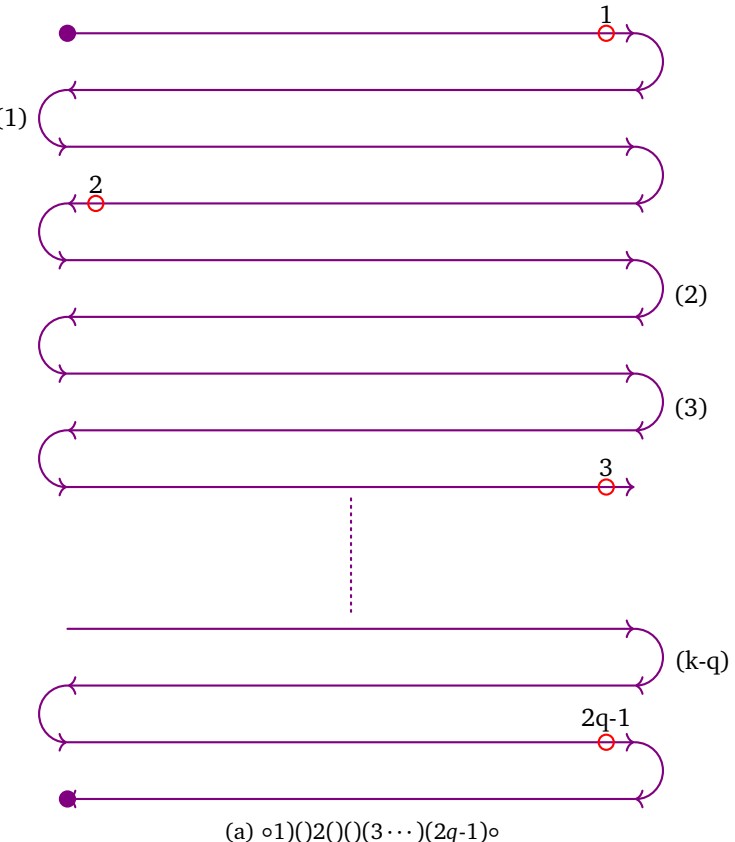

(a) ∘1)()2()()(3⋯)(2q-1)∘

Figure 4: Arranging $(k-q)$ empty timefolds between $(2q-1)$ turning-point operators (which are in the canonical arrangement). We have labeled the $i^{th}$ turning-point operator and the $j^{th}$ timefold is denoted by $(j)$. For the case of minimal $n$, i.e., $n = 2q-1$ with all operators being turning-point operators, the counting reduces to the number of ways $(k-q)$ empty timefolds can be put into $2q$ boxes created by the legs with turning-point operators (and the last empty leg). This gives $\frac{(k-q+2q-1)!}{(k-q)!(2q-1)!} = \frac{(k+q-1)!}{(k-q)!(2q-1)!}$ number of ways of arranging empty timefolds.

First of all, as we have seen before, even staying within $q$-timefolds the turning-point operators can jump across the turning-points giving rise to $2^{2q-1}$ possibilities. Having counted

this, let us fix this flip degree of freedom, by demanding that operators near turning-points are always fixed to be before the turning-points. This ensures that each of the $2q - 1$ insertions are on a different leg. We will call this arrangement as the *canonical* arrangement.

Any embedding of the canonical arrangement within $k$ timefolds is built with alternating future operator legs and past operator legs, interspersed by integer number of timefolds. Note further that any canonical arrangement starts with a future operator leg and ends with a future operator leg. In addition, along with full timefolds, the last leg of the $k$ timefolds is always free, because of our prescription to fix all operator insertions to be before the turning-points in the canonical arrangement. Thus, there are $q$ future operator legs , $(q - 1)$ past operator legs, $(k - q)$ total number of timefolds, giving $2(k - q)$ empty legs. Finally we have to account for the last empty leg of the contour. In total, these add up to $q + (q - 1) + 2(k - q) + 1 = 2k$ legs as they should. The situation is illustrated in Fig. 4.

We then have to count how many ways we can get such an arrangement. This reduces to the following counting problem: in how many ways can $k - q$ timefolds be put into $2q$ gaps created by the $2q - 1$ occupied legs? This is quite standard (familiar say from counting bosonic multi-particle states) and leads to

$$\frac{(k - q + 2q - 1)!}{(k - q)!(2q - 1)!} = \frac{(k + q - 1)!}{(k - q)!(2q - 1)!} \,.\tag{63}$$

Together with the $2^{2q-1}$ jumps of operators across turning-points, we get the number of $k$ timefold arrangements giving the same proper $q$-OTO correlator with $n = 2q - 1$ insertions:

$$h^{(q)}_{n=2q-1,k} = 2^{2q-1} \frac{(k + q - 1)!}{(k - q)!(2q - 1)!} \,.\tag{64}$$

**General situation ($n, k, q$ arbitrary):**  Having explored two special cases, to build intuition, we can now consider the problem in earnest and compute the degeneracy factor $h^{(q)}_{n,k}$ for the general $n, k, q$. Readers interested in the main result are invited to skip the derivation and proceed to the final answer in (70).

We now have the benefit of the two above examples, so we will start by considering a canonical arrangement of these $n$ operators. Specifically, $q$ of these operators live near future turning-points, $q - 1$ live near past turning-points, and all these turning-point operators are fixed to be before their respective turning-points. This enables us fix the $2^{2q-1}$ turning-point degeneracy, which we will fold into the analysis at the very end.

This implies that we are now left with $n - (2q - 1)$ extra operators to sprinkle across the $k$-OTO contour. They must clearly be punctuated by the turning-point operators, owing to the arrangement chosen. It will be convenient to think of these non-turning-point operators as divided into different groups by the $2q - 1$ turning-point operators.

Pick two of the neighbouring past turning-points (or before the first or after the last past turning-point). Between them lies a single future turning-point operator along with a set of non-turning-point operators on either side of it. As presaged in §2.4 we will call these as making up the *wing* of the future turning-point operator in question. There are $q$ such wings, one for each future turning-point. For a given time-ordering, the $n - (2q - 1)$ *wing operators* are distributed among these $q$ wings in a fixed way.

By convention we choose the wing to start from the timefold containing first wing operator before encountering the future turning-point operator in question. The wing likewise ends with the timefold containing last wing operator after the future turning-point operator. Every wing operator has a neighbour which is closer to the future turning point operator (which may be the future turning-point operator itself) and another neighbour which is farther from the future turning point operator. We will call the former the *near-wing neighbour* and the latter

as the *far-wing neighbour*. We would like to count the number of ways how these $q$ wings can be accommodated into $k$-timefolds.

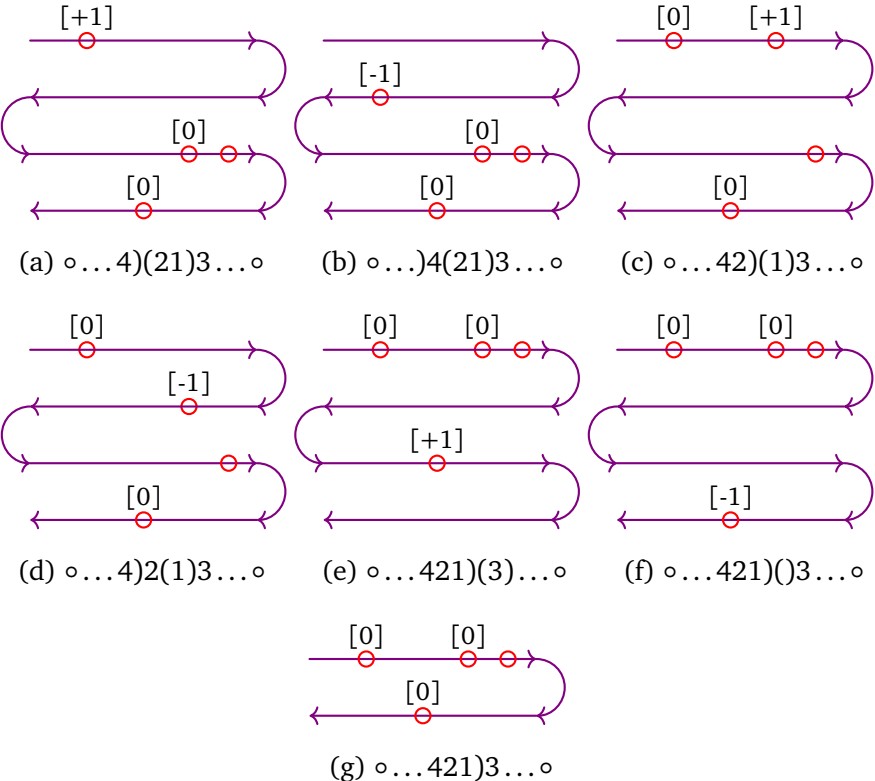

Figure 5: Wing spread ($w = 1$) configurations of the basic wing shown in fig (g) with the wing positions marked for every wing operator. Note that the future turning-point operator is always placed before the future turning-point,i.e., in canonical arrangement. Each wing configuration is completely specified by wing positions $\{x_1, x_2, x_3\}$ of the three wing operators. The wing-spread can be computed from the wing positions by using the formula $w = |x_1| + |x_2| + |x_3|$.

In order to do this counting, we will introduce the notion of a *wing-spread w* which is defined as the amount of extra timefolds occupied by the wing. Note that, by definition, any wing can be compressed within a single timefold, by successive sliding of wing operators without changing their ordering. We are interested in how many *additional* timefolds, over and above this minimum of one, has the wing-spread into. Thus, $w + 1$ is the total number of timefolds a wing occupies.

In order to facilitate the computation of wing-spread, we will begin by assigning a *wing position* to each of the wing operators. The *wing position* is the distance to the near-wing neighbour measured in number of timefolds. Alternately, it is the number of past turning-points one has to cross to reach the near-wing neighbour.

In order to completely specify the position of wing operators, we will also include a sign in the wing position: if the near-wing neighbour is in the same type of contour (among R and L) as the wing operator, the wing position is taken to be positive. If the near-wing neighbour is in the opposite type of contour (among R and L) as the wing operator, the wing position is taken to be negative instead. Alternately, when the wing position is positive, the number of future and past turning-points one has to cross to reach the near-wing neighbour are equal. It is negative if they are not equal.

The wing-spread $w_i$ of $i^{th}$ wing can then be computed as the sum of the magnitudes of all

the wing positions $x_\alpha^{(i)}$, i.e., $w_i = \sum_\alpha |x_\alpha^{(i)}|$ where the sum is over all the wing operators.

Let us illustrate these rules of assigning wing positions with an example shown in Fig. 5. As shown in Fig. 5(g), we consider the case with a future turning-point operator, along with two wing operators coming before it in the contour, and one operator coming afterwards. When all these are accommodated within a single time fold (and hence wing-spread is zero), and when the future turning-point operator is placed before the turning-point, we say that the wing is in the canonical arrangement. The reader can convince itself that for a given time ordering there is a unique configuration of the wing that satisfies these properties. Computing the wing positions for each wing operator, one sees that all of them turn out to be zero.

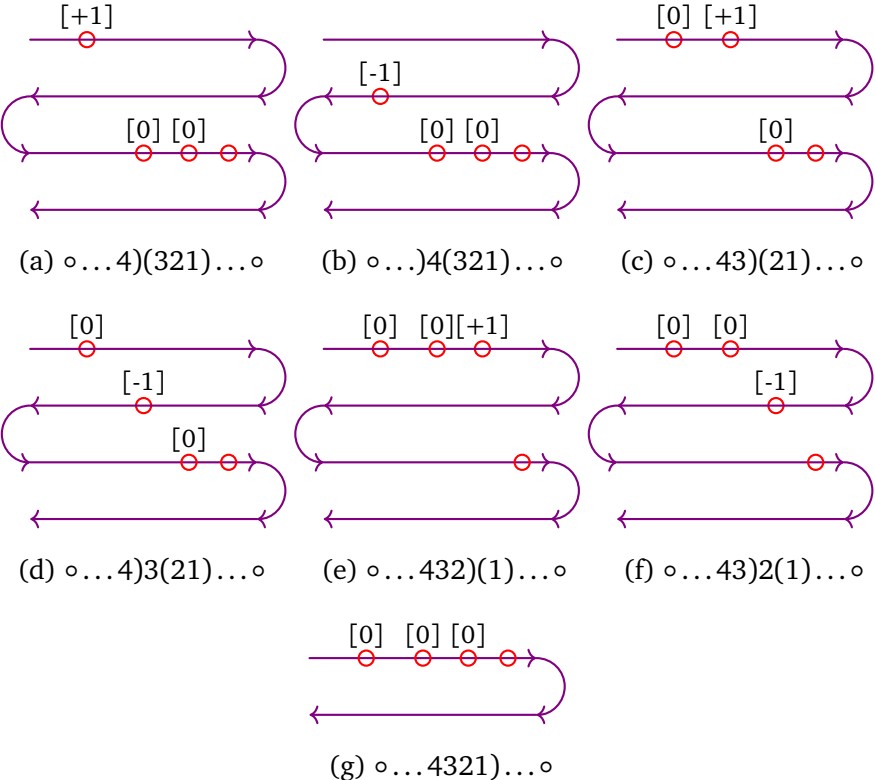

Figure 6: Same as Fig. 5, but with a different time ordering. Note that wing position assignments are the same as Fig. 5 even though the time-orderings are different. The number of wing-spread ($w = 1$) configurations is still six, and is independent of the details of time-ordering.

We then consider the next simplest case, where the wing-spreads and covers one more time fold. There are six such one wing-spread configurations as shown in Fig. 5 corresponding to six wing positions: $\{x_1, x_2, x_3\} = \{\pm1, 0, 0\}$, $\{0, \pm1, 0\}$, $\{0, 0, \pm1\}$, respectively. Using the formula $w_i = \sum_\alpha |x_\alpha^{(i)}|$, we can check that all these correspond to a wing-spread of one. Another example appears in Fig. 6 where we consider the case with a future turning-point operator along with three wing operators coming before it. The considerations here are very similar and we again arrive at six one wing-spread configurations. A few more examples are depicted in Figs. 7, 8, and 9, where we also display configurations with wing-spread 2.

From these examples, it is clear that to count the wing configurations of $i^{th}$ wing with a given wing-spread $w_i$ and $m_i$ number of wing operators, we need to count the number of possible wing positions. In other words, we need to count the number of integer $x_\alpha$'s such

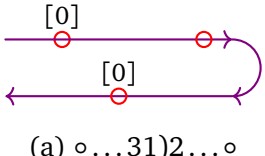

(a) ∘...31)2...∘

Figure 7: Three operators in wing configurations of spread 0.

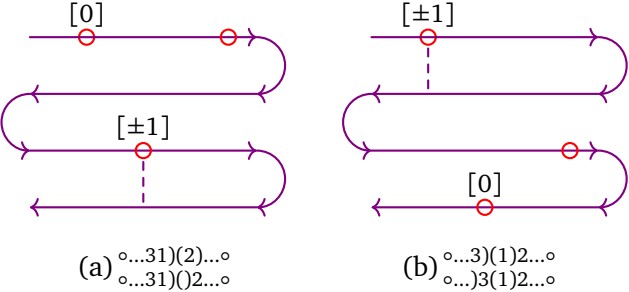

(a) ∘...31)(2)...∘
    ∘...31)()2...∘

(b) ∘...3)(1)2...∘
    ∘...)3(1)2...∘

Figure 8: Same three operators in Fig. 7, but now in wing configurations of spread 1. The dashed lines denote the possible positions of operators. There are 4 wing configurations of spread 1.

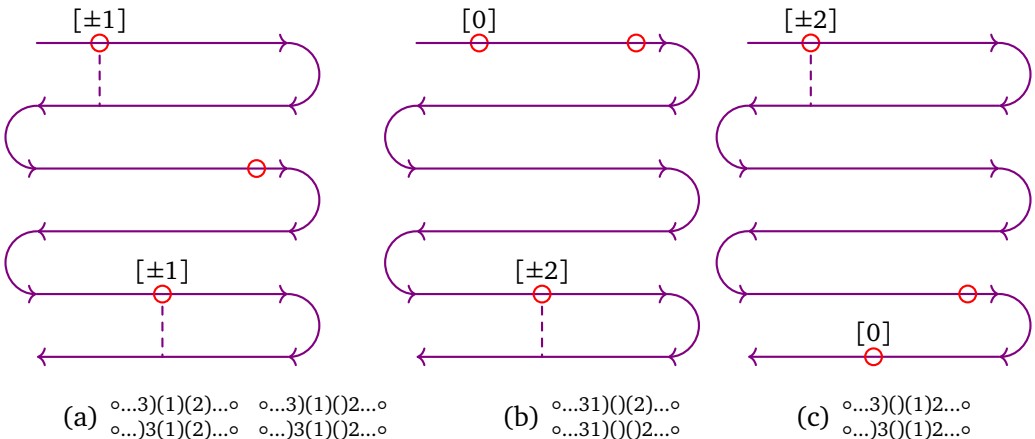

(a) ∘...3)(1)(2)...∘   ∘...3)(1)()2...∘
    ∘...)3(1)(2)...∘   ∘...)3(1)()2...∘

(b) ∘...31)()(2)...∘
    ∘...31)()()2...∘

(c) ∘...3)()(1)2...∘
    ∘...)3()(1)2...∘

Figure 9: Same three operators in Fig. 7, but now in wing configurations of spread 2. The dashed lines denote the possible positions of operators. There are $4 + 2 + 2 = 8$ wing configurations of spread 3.

that

$$\sum_{\alpha=1}^{m_i} |x_\alpha| = w_i. \tag{65}$$

Geometrically, this is the problem of counting the number of $w^{\text{th}}$ nearest neighbours of a site in a cubic lattice of dimension $m_i$, with the distances measured using $L_1$-metric (or Manhattan metric). This is a standard problem whose result is called the co-ordination sequence of the cubic lattice. To see how one might go about computing this sequence, let us start with a $1d$ lattice of points and write down a generating function for the number of $w^{\text{th}}$ nearest neighbours. A simple computation leads to

$$1 + 2t + 2t^2 + \ldots = \frac{1+t}{1-t}. \tag{66}$$

This can easily be generalized to a cubic lattice of dimension $n_i$, whereby the number of $w_i^{\text{th}}$ nearest neighbours is given by the coefficient of $t^{w_i}$ in

$$\left(\frac{1+t}{1-t}\right)^{n_i}. \tag{67}$$

We can now count how many wing configurations of all the wings are there with total wing-spread $w = \sum_{i=1}^{q} w_i$ covered by the $(n - 2q + 1) = \sum_i n_i$ number of wing operators. We obtain

# of configurations of total wing-spread $w = $ Coefficient of $t^w$ in $\left(\frac{1+t}{1-t}\right)^{n-2q+1}$. $\tag{68}$

Now the total number of timefolds, which are occupied by wings with total wing-spread $w = \sum_{i=1}^{q} w_i$, is $\sum_{i=1}^{q}(w_i + 1) = w + q$, where we have also counted the timefolds in which the future turning-point operators lie. The remaining number of empty timefolds is then given by $(k - w - q)$ which need to be put into $2q$ boxes between $q$ wings and $q - 1$ past turning-point operators. The number of ways to rearrange the empty timefolds is then given by the binomial coefficient

$$\left(\begin{array}{c} k - w - q + 2q - 1 \\ k - w - q \end{array}\right).$$

Finally, we have to account for the turning-point degeneracy of $2^{2q-1}$, which arises if we move out of the canonical configuration of the turning-point operators.

Putting together the turning-point degeneracy, empty timefold configurations, and the number of wing configurations, we finally get

$$\begin{aligned}
\text{Total degeneracy } h_{n,k}^{(q)} &= \sum_{w=0}^{k-q} 2^{2q-1} \times \left(\begin{array}{c} k - w - q + 2q - 1 \\ k - w - q \end{array}\right) \\
&\quad \times \text{Coefficient of } t^w \text{ in } \left[\left(\frac{1+t}{1-t}\right)^{n-2q+1}\right] \\
&= \sum_{w=0}^{k-q} \text{Coefficient of } t^{k-w} \text{ in } \left[2^{2q-1}\frac{t^q}{(1-t)^{2q}}\right] \\
&\quad \times \text{Coefficient of } t^w \text{ in } \left[\left(\frac{1+t}{1-t}\right)^{n-2q+1}\right] \\
&= \text{Coefficient of } t^k \text{ in } \left[2^{2q-1}\frac{t^q}{(1-t)^{2q}}\left(\frac{1+t}{1-t}\right)^{n-2q+1}\right].
\end{aligned} \tag{69}$$

We combine this result into a generating function for the degeneracy factor. Define

$$\boxed{\mathcal{H}_{q,n}(t) = \sum_{k=q}^{\infty} h_{n,k}^{(q)} t^k = 2^{2q-1} t^q \frac{(1+t)^{n-(2q-1)}}{(1-t)^{n+1}} \Theta(n - (2q-1)).} \tag{70}$$

The coefficients, one can check, satisfy the recursion relation

$$h_{n+1,k+1}^{(q)} - h_{n+1,k}^{(q)} = h_{n,k+1}^{(q)} + h_{n,k}^{(q)}. \tag{71}$$

Multiplying by $t^k$ and summing $k$ from $q$ to $\infty$, we get the following recurrence relation for the generating functions

$$\mathcal{H}_{q,n+1}(t) = \left(\frac{1+t}{1-t}\right)\mathcal{H}_{q,n}(t) + t^q \frac{h_{n+1,q}^{(q)} - h_{n,q}^{(q)}}{(1-t)}. \tag{72}$$

We now use $h^{(q)}_{n+1,q} = h^{(q)}_{n,q} = 2^{2q-1}$ as well as our previous result that

$$\mathcal{H}_{q,n=(2q-1)}(t) = 2^{2q-1} t^q \frac{1}{(1-t)^{2q}}$$

to completely solve for $\mathcal{H}_{q,n}(t)$. This gives us back the result quoted in (70). In the course of the derivation, we found it useful to derive various recursion relations satisfied by $h^{(q)}_{n,k}$. These are collected for reference in Appendix B.

# 6 Low-point functions exemplified

This section serves as an illustration of the ideas and countings presented in the paper for the case of $n$-point functions with $n \leq 4$. Clearly, $n = 1$ is trivial, so we start with 2-point functions.

## 6.1 Two-point functions

The $n!$ Wightman basis correlators can be combined with suitable step functions to express the correlators in any other basis. Let us now demonstrate this explicitly for $n = 2$:

- **Wightman basis ($n! = 2$ correlators):** These are labeled by permutations $\sigma \in S_2 = \{\mathrm{Id}, (12)\}$:

$$\begin{aligned} G_{\mathrm{Id}}(t_1, t_2) &\equiv G(t_1, t_2) = \langle \widehat{\mathbb{O}}_1 \widehat{\mathbb{O}}_2 \rangle, \\ G_{(12)}(t_1, t_2) &\equiv G(t_2, t_1) = \langle \widehat{\mathbb{O}}_2 \widehat{\mathbb{O}}_1 \rangle. \end{aligned} \tag{73}$$

These Wightman functions can be obtained from the Euclidean correlator $G_{\mathrm{E}}(\tau_1, \tau_2)$ by the two distinct choices for analytic continuation in (10), viz., either $\epsilon_1 > \epsilon_2$ or $\epsilon_1 < \epsilon_2$:

$$G_{\mathrm{E}}(\tau_1 = i t_1 + \epsilon_1, \tau_2 = i t_2 + \epsilon_2) = \begin{cases} \langle \widehat{\mathbb{O}}_1(t_1) \widehat{\mathbb{O}}_2(t_2) \rangle & \text{if } \epsilon_1 > \epsilon_2 \\ \langle \widehat{\mathbb{O}}_2(t_2) \widehat{\mathbb{O}}_1(t_1) \rangle & \text{if } \epsilon_2 > \epsilon_1 \end{cases}. \tag{74}$$

- **Nested correlators ($2^{n-2} n! = 2$ correlators):** Since there are no sJacobi identities for $n = 2$, relating the nested correlators back to the Wightman basis is trivial:

$$\begin{aligned} \langle [\widehat{\mathbb{O}}_1, \widehat{\mathbb{O}}_2] \rangle &= G(t_1, t_2) - G(t_2, t_1), \\ \langle \{\widehat{\mathbb{O}}_1, \widehat{\mathbb{O}}_2\} \rangle &= G(t_1, t_2) + G(t_2, t_1). \end{aligned} \tag{75}$$

- **Av-Dif correlators ($(2k)^n = 4k^2$ correlators on a $k$-OTO contour):** A 1-OTO (Schwinger-Keldysh) contour is sufficient to compute all 2-point correlators. The Keldysh rules of §4.2 allow us to relate contour-ordered Av-Dif correlators to nested correlators. In the present case, we have $(2k)^n = 4$ and find the following associated contour-ordered correlation functions:[17]

$$\begin{aligned} \langle \mathcal{T}_{\mathcal{C}} \, \mathbb{O}_{av}(1) \mathbb{O}_{av}(2) \rangle &= \langle \{\widehat{\mathbb{O}}(1), \widehat{\mathbb{O}}(2)\}_{\pm} \rangle = \frac{1}{2} \left( G(t_1, t_2) + (-1)^{\widehat{\mathbb{O}}_1 \widehat{\mathbb{O}}_2} G(t_2, t_1) \right), \\ \langle \mathcal{T}_{\mathcal{C}} \, \mathbb{O}_{av}(1) \mathbb{O}_{dif}(2) \rangle &= \Theta_{12} \langle [\widehat{\mathbb{O}}(1), \widehat{\mathbb{O}}(2)]_{\pm} \rangle = \Theta_{12} \left( G(t_1, t_2) - (-1)^{\widehat{\mathbb{O}}_1 \widehat{\mathbb{O}}_2} G(t_2, t_1) \right), \\ \langle \mathcal{T}_{\mathcal{C}} \, \mathbb{O}_{dif}(1) \mathbb{O}_{av}(2) \rangle &= -\Theta_{21} \langle [\widehat{\mathbb{O}}(1), \widehat{\mathbb{O}}(2)]_{\pm} \rangle = -\Theta_{21} \left( G(t_1, t_2) - (-1)^{\widehat{\mathbb{O}}_1 \widehat{\mathbb{O}}_2} G(t_2, t_1) \right), \\ \langle \mathcal{T}_{\mathcal{C}} \, \mathbb{O}_{dif}(1) \mathbb{O}_{dif}(2) \rangle &= 0. \end{aligned} \tag{76}$$

---

[17] We note that the extra factor of $\frac{1}{2}$ in the first line has to do with our definition of the graded anticommutator. We allow here also for Grassmann-odd operators.

The second and third lines are otherwise also known as ($i$ times the) retarded and advanced Green's functions, respectively. The first line is sometimes called as ($i$ times the) Keldysh function.

- **LR correlators ($(2k)^n = 4k^2$ elements on a $k$-OTO contour):** The contour ordering can be explicitly implemented by inspection of the corresponding picture. A 1-OTO contour suffices to generate all correlators, which are then easily related to the Wightman basis:

$$
\begin{aligned}
\langle \mathcal{T}_C \, \mathbb{O}_{\mathrm{R}}^1(1)\mathbb{O}_{\mathrm{R}}^1(2)\rangle &\equiv \circ 21)\circ = \langle \widehat{\mathbb{O}}_1\widehat{\mathbb{O}}_2\rangle = G(t_1,t_2), \\
\langle \mathcal{T}_C \, \mathbb{O}_{\mathrm{L}}^1(1)\mathbb{O}_{\mathrm{R}}^1(2)\rangle &\equiv \circ 2)1\circ = \langle \widehat{\mathbb{O}}_1\widehat{\mathbb{O}}_2\rangle = G(t_1,t_2), \\
\langle \mathcal{T}_C \, \mathbb{O}_{\mathrm{R}}^1(1)\mathbb{O}_{\mathrm{L}}^1(2)\rangle &\equiv \circ 1)2\circ = \langle \widehat{\mathbb{O}}_2\widehat{\mathbb{O}}_1\rangle = G(t_2,t_1), \\
\langle \mathcal{T}_C \, \mathbb{O}_{\mathrm{L}}^1(1)\mathbb{O}_{\mathrm{L}}^1(2)\rangle &\equiv \circ )12\circ = \langle \widehat{\mathbb{O}}_2\widehat{\mathbb{O}}_1\rangle = G(t_2,t_1).
\end{aligned}
\tag{77}
$$

We remind the reader that we always assume $t_1 > t_2 > \ldots > t_n$. Alternatively, we can easily go back and forth between LR correlators and Av-Dif correlators by the simple basis rotation (8). Of course, we could choose to redundantly represent these correlators in a $k$-OTO contour with $k > 1$. For illustration, let us use our symbolic notation explained in §5.1 to demonstrate the relations among all correlators for $n = 2$ and $k = 2$:

$$
\begin{aligned}
G(t_1,t_2) &= \circ 21()()\circ = \circ 2)1()\circ = \circ 2)(1)\circ = \circ 2)()1\circ = \circ )2(1)\circ = \circ )2()1\circ = \circ )(21)\circ = \circ )(2)1\circ, \\
G(t_2,t_1) &= \circ 1)2()\circ = \circ 1)(2)\circ = \circ 1)()2\circ = \circ )12()\circ = \circ )1(2)\circ = \circ )1()2\circ = \circ )(1)2\circ = \circ )()12\circ.
\end{aligned}
\tag{78}
$$

The degeneracy of 8 in each of the two lines corresponds to the general result $h_{2,k}^{(1)} = 2k^2$ for proper 1-OTO 2-point functions embedded in a ($k = 2$)-OTO contour (see Table 2).

## 6.2 Three-point functions

Let us repeat the analysis for 3-point functions. The new element in this discussion will be the necessity to involve 2-OTO contours.

- **Wightman basis ($n! = 6$ correlators):** These are labeled by permutations of three objects. Explicitly, the six independent Wightman correlators are:

$$
\begin{aligned}
G_\sigma(t_1,t_2,t_3) &\equiv G(t_{\sigma(1)},t_{\sigma(2)},t_{\sigma(3)}) = \langle \widehat{\mathbb{O}}_{\sigma(1)}\widehat{\mathbb{O}}_{\sigma(2)}\widehat{\mathbb{O}}_{\sigma(3)}\rangle \\
&\text{for } \sigma \in S_3 = \{\mathrm{id},(12),(23),(13),(123),(132)\}.
\end{aligned}
\tag{79}
$$

Let us again note how these can all be obtained from the Euclidean correlator $G_{\mathrm{E}}(\tau_1,\tau_2,\tau_3)$, by the analytic continuation $\tau_i = it_i + \epsilon_i$, c.f., (10). There are now $3! = 6$ choices for the ordering of the $\epsilon_i$ which lead to

$$
G_{\mathrm{E}}(\tau_1,\tau_2,\tau_3)\big|_{\tau_i=it_i+\epsilon_i} =
\begin{cases}
\langle \widehat{\mathbb{O}}_1(t_1)\widehat{\mathbb{O}}_2(t_2)\widehat{\mathbb{O}}_3(t_3)\rangle & \text{if } \epsilon_1 > \epsilon_2 > \epsilon_3 \\
\langle \widehat{\mathbb{O}}_1(t_1)\widehat{\mathbb{O}}_3(t_3)\widehat{\mathbb{O}}_2(t_2)\rangle & \text{if } \epsilon_1 > \epsilon_3 > \epsilon_2 \\
\langle \widehat{\mathbb{O}}_2(t_2)\widehat{\mathbb{O}}_3(t_3)\widehat{\mathbb{O}}_1(t_1)\rangle & \text{if } \epsilon_2 > \epsilon_3 > \epsilon_1 \\
\langle \widehat{\mathbb{O}}_2(t_2)\widehat{\mathbb{O}}_1(t_1)\widehat{\mathbb{O}}_3(t_3)\rangle & \text{if } \epsilon_2 > \epsilon_1 > \epsilon_3 \\
\langle \widehat{\mathbb{O}}_3(t_3)\widehat{\mathbb{O}}_1(t_1)\widehat{\mathbb{O}}_2(t_2)\rangle & \text{if } \epsilon_3 > \epsilon_1 > \epsilon_2 \\
\langle \widehat{\mathbb{O}}_3(t_3)\widehat{\mathbb{O}}_2(t_2)\widehat{\mathbb{O}}_1(t_1)\rangle & \text{if } \epsilon_3 > \epsilon_2 > \epsilon_1
\end{cases}.
\tag{80}
$$

- **Nested correlators ($2^{n-2}n! = 12$ correlators):** a-priori, there are 12 nested correlators which can be succinctly written as

$$
\langle (\!(\,(\!(\widehat{\mathbb{O}}_{\sigma(1)},\widehat{\mathbb{O}}_{\sigma(2)})\!)_\alpha,\widehat{\mathbb{O}}_{\sigma(3)})\!)_\beta\rangle
\tag{81}
$$

for all 4 choices $\alpha, \beta \in \{av, dif\}$ and $\frac{3!}{2} = 3$ inequivalent permutations $\sigma \in S_3$ which do not just permute the innermost operators in (81) (this would at most change the sign of the correlator). More explicitly, we can write these 12 choices as

$$\langle[[\widehat{\mathbb{O}}_1, \widehat{\mathbb{O}}_2], \widehat{\mathbb{O}}_3]\rangle, \quad \langle[\{\widehat{\mathbb{O}}_1, \widehat{\mathbb{O}}_2\}, \widehat{\mathbb{O}}_3]\rangle, \quad \langle\{[\widehat{\mathbb{O}}_1, \widehat{\mathbb{O}}_2], \widehat{\mathbb{O}}_3\}\rangle, \quad \langle\{\{\widehat{\mathbb{O}}_1, \widehat{\mathbb{O}}_2\}, \widehat{\mathbb{O}}_3\}\rangle,$$

$$\langle[[\widehat{\mathbb{O}}_1, \widehat{\mathbb{O}}_3], \widehat{\mathbb{O}}_2]\rangle, \quad \langle[\{\widehat{\mathbb{O}}_1, \widehat{\mathbb{O}}_3\}, \widehat{\mathbb{O}}_2]\rangle, \quad \langle\{[\widehat{\mathbb{O}}_1, \widehat{\mathbb{O}}_3], \widehat{\mathbb{O}}_2\}\rangle, \quad \langle\{\{\widehat{\mathbb{O}}_1, \widehat{\mathbb{O}}_3\}, \widehat{\mathbb{O}}_2\}\rangle, \quad (82)$$

$$\langle[[\widehat{\mathbb{O}}_2, \widehat{\mathbb{O}}_3], \widehat{\mathbb{O}}_1]\rangle, \quad \langle[\{\widehat{\mathbb{O}}_2, \widehat{\mathbb{O}}_3\}, \widehat{\mathbb{O}}_1]\rangle, \quad \langle\{[\widehat{\mathbb{O}}_2, \widehat{\mathbb{O}}_3], \widehat{\mathbb{O}}_1\}\rangle, \quad \langle\{\{\widehat{\mathbb{O}}_2, \widehat{\mathbb{O}}_3\}, \widehat{\mathbb{O}}_1\}\rangle.$$

By explicitly expanding out the commutators and anti-commutators, it is straightforward to write these in terms of the six basic functions (79). More abstractly, to see that only 6 of the above 12 correlators are independent, we observe that there are 6 independent sJacobi identities for $n = 3$, which we can write as:

$$\langle[[\widehat{\mathbb{O}}_1, \widehat{\mathbb{O}}_2], \widehat{\mathbb{O}}_3] + \{\{\widehat{\mathbb{O}}_1, \widehat{\mathbb{O}}_3\}, \widehat{\mathbb{O}}_2\} - \{\{\widehat{\mathbb{O}}_2, \widehat{\mathbb{O}}_3\}, \widehat{\mathbb{O}}_1\}\rangle = 0,$$

$$\langle[[\widehat{\mathbb{O}}_3, \widehat{\mathbb{O}}_1], \widehat{\mathbb{O}}_2] + \{\{\widehat{\mathbb{O}}_2, \widehat{\mathbb{O}}_3\}, \widehat{\mathbb{O}}_1\} - \{\{\widehat{\mathbb{O}}_1, \widehat{\mathbb{O}}_2\}, \widehat{\mathbb{O}}_3\}\rangle = 0,$$

$$\langle[[\widehat{\mathbb{O}}_2, \widehat{\mathbb{O}}_3], \widehat{\mathbb{O}}_1] + \{\{\widehat{\mathbb{O}}_2, \widehat{\mathbb{O}}_1\}, \widehat{\mathbb{O}}_3\} - \{\{\widehat{\mathbb{O}}_3, \widehat{\mathbb{O}}_1\}, \widehat{\mathbb{O}}_2\}\rangle = 0,$$

$$\langle[\{\widehat{\mathbb{O}}_1, \widehat{\mathbb{O}}_2\}, \widehat{\mathbb{O}}_3] - [\{\widehat{\mathbb{O}}_2, \widehat{\mathbb{O}}_3\}, \widehat{\mathbb{O}}_1] + [\{\widehat{\mathbb{O}}_3, \widehat{\mathbb{O}}_1\}, \widehat{\mathbb{O}}_2]\rangle = 0,$$

$$\langle[\{\widehat{\mathbb{O}}_1, \widehat{\mathbb{O}}_3\}, \widehat{\mathbb{O}}_2] - [\{\widehat{\mathbb{O}}_3, \widehat{\mathbb{O}}_2\}, \widehat{\mathbb{O}}_1] + [\{\widehat{\mathbb{O}}_2, \widehat{\mathbb{O}}_1\}, \widehat{\mathbb{O}}_3]\rangle = 0,$$

$$\langle[\{\widehat{\mathbb{O}}_3, \widehat{\mathbb{O}}_2\}, \widehat{\mathbb{O}}_1] - [\{\widehat{\mathbb{O}}_2, \widehat{\mathbb{O}}_1\}, \widehat{\mathbb{O}}_3] + [\{\widehat{\mathbb{O}}_1, \widehat{\mathbb{O}}_3\}, \widehat{\mathbb{O}}_2]\rangle = 0.$$

$$(83)$$

The first three lines transform into each other under action of $S_3$. Adding them up yields the standard Jacobi identity. Similarly for the last three lines.

- **Av-Dif correlators ($(2k)^n = 8k^3$ correlators on a $k$-OTO contour):** For 3-point functions, we need to consider 2-OTO contours (i.e., $k = 2$). Using the notation of §4.4, we use labels $\boldsymbol{\alpha}, \boldsymbol{\beta}, \boldsymbol{\gamma} \in \{av, dif\}$ to denote the operator type and integers $\alpha, \beta, \gamma \in \{1, 2\}$ to label the contour. The 3-point functions in the Av-Dif correlators can then all be written as

$$\langle\mathcal{T}_{\mathcal{C}} \ \mathbb{O}_{\boldsymbol{\alpha}}^{\alpha}(1)\mathbb{O}_{\boldsymbol{\beta}}^{\beta}(2)\mathbb{O}_{\boldsymbol{\gamma}}^{\gamma}(3)\rangle \quad (84)$$

for $(2 \times 2)^3 = 64$ choices of $\boldsymbol{\alpha}, \boldsymbol{\beta}, \boldsymbol{\gamma}$ and $\alpha, \beta, \gamma$. Similar to the case of 2-point functions, these can be related to the nested correlators basis by making use of the double bracket (see §4.4):

$$\langle\mathcal{T}_{\mathcal{C}} \ \mathbb{O}_{\boldsymbol{\gamma}}^{\alpha}(1)\mathbb{O}_{\boldsymbol{\alpha}}^{1}(2)\mathbb{O}_{\boldsymbol{\beta}}^{1}(3)\rangle = \mathfrak{d}_{\boldsymbol{\gamma}}\langle(\!(\!(\widehat{\mathbb{O}}(1), \widehat{\mathbb{O}}(2))\!)_{\boldsymbol{\alpha}}, \widehat{\mathbb{O}}(3))\!)_{\boldsymbol{\beta}}\rangle,$$

$$\langle\mathcal{T}_{\mathcal{C}} \ \mathbb{O}_{\boldsymbol{\beta}}^{\alpha}(1)\mathbb{O}_{\boldsymbol{\alpha}}^{2}(2)\mathbb{O}_{\boldsymbol{\gamma}}^{2}(3)\rangle = \mathfrak{d}_{\boldsymbol{\gamma}}\langle(\!(\!(\widehat{\mathbb{O}}(3), \widehat{\mathbb{O}}(2))\!)_{\boldsymbol{\alpha}}, \widehat{\mathbb{O}}(1))\!)_{\boldsymbol{\beta}}\rangle,$$

$$\langle\mathcal{T}_{\mathcal{C}} \ \mathbb{O}_{\boldsymbol{\alpha}}^{\alpha}(1)\mathbb{O}_{\boldsymbol{\gamma}}^{2}(2)\mathbb{O}_{\boldsymbol{\beta}}^{1}(3)\rangle = \mathfrak{d}_{\boldsymbol{\gamma}}\langle(\!(\!(\widehat{\mathbb{O}}(2), \widehat{\mathbb{O}}(1))\!)_{\boldsymbol{\alpha}}, \widehat{\mathbb{O}}(3))\!)_{\boldsymbol{\beta}}\rangle,$$

$$\langle\mathcal{T}_{\mathcal{C}} \ \mathbb{O}_{\boldsymbol{\alpha}}^{\alpha}(1)\mathbb{O}_{\boldsymbol{\beta}}^{1}(2)\mathbb{O}_{\boldsymbol{\gamma}}^{2}(3)\rangle = \mathfrak{d}_{\boldsymbol{\gamma}}\langle(\!(\!(\widehat{\mathbb{O}}(3), \widehat{\mathbb{O}}(1))\!)_{\boldsymbol{\alpha}}, \widehat{\mathbb{O}}(2))\!)_{\boldsymbol{\beta}}\rangle.$$

$$(85)$$

- **LR correlators ($(2k)^n = 8k^3$ elements on a $k$-OTO contour):** We work again on the 2-OTO contour (higher $k$ can be considered and leads to more redundancy). The counting here is the same as in the Av-Dif correlators . Each correlation function can be written as

$$\langle\mathcal{T}_{\mathcal{C}} \ \mathbb{O}_{\bar{\boldsymbol{\alpha}}}^{\alpha}(1)\mathbb{O}_{\bar{\boldsymbol{\beta}}}^{\beta}(2)\mathbb{O}_{\bar{\boldsymbol{\gamma}}}^{\gamma}(3)\rangle \quad (86)$$

for 64 choices of $\bar{\boldsymbol{\alpha}}, \bar{\boldsymbol{\beta}}, \bar{\boldsymbol{\gamma}} \in \{R, L\}$ and $\alpha, \beta, \gamma \in \{1, 2\}$. The non-trivial part is to figure out the $64 - 6 = 58$ relations giving these RL-correlators on the 2-OTO contour in terms of 6 Wightman functions. The simplest way to give these relations is by using again a

pictorial representation. The following relations are those among the $g_{3,1} = 4$ proper 1-OTO 3-point functions:

$$
\begin{aligned}
G(t_1, t_2, t_3) &= \circ 321)()\circ = \circ 32)1()\circ = \circ 32)(1)\circ = \circ 32)()1\circ = \circ 3)2(1)\circ = \circ 3)2()1\circ \\
&= \circ 3)(21)\circ = \circ 3)(2)1\circ = \circ )3(21)\circ = \circ )3(2)1\circ = \circ )(321)\circ = \circ )(32)1\circ , \\
G(t_2, t_1, t_3) &= \circ 31)2()\circ = \circ 31)(2)\circ = \circ 31)()2\circ = \circ 3)12()\circ = \circ 3)1(2)\circ = \circ 3)1()2\circ \\
&= \circ 3)(1)2\circ = \circ 3)()12\circ = \circ )3(1)2\circ = \circ )3()12\circ = \circ )(31)2\circ = \circ )(3)12\circ , \\
G(t_3, t_1, t_2) &= \circ 21)3()\circ = \circ 21)(3)\circ = \circ 21)()3\circ = \circ 2)13()\circ = \circ 2)1(3)\circ = \circ 2)1()3\circ \\
&= \circ 2)(1)3\circ = \circ 2)()13\circ = \circ )2(1)3\circ = \circ )2()13\circ = \circ )(21)3\circ = \circ )(2)13\circ , \\
G(t_3, t_2, t_1) &= \circ 1)23()\circ = \circ 1)2(3)\circ = \circ 1)2()3\circ = \circ 1)(2)3\circ = \circ 1)()23\circ = \circ )123()\circ \\
&= \circ )12(3)\circ = \circ )12()3\circ = \circ )1(2)3\circ = \circ )1()23\circ = \circ )(1)23\circ = \circ )()123 \circ .
\end{aligned}
\tag{87}
$$

The degeneracy in each case is $h_{3,2}^{(1)} = 12$. Similarly, there are relations among the $g_{3,2} = 2$ proper 2-OTO 3-point functions:

$$
\begin{aligned}
G(t_1, t_3, t_2) &= \circ 2)3(1)\circ = \circ 2)3()1\circ = \circ 2)(31)\circ = \circ 2)(3)1\circ \\
&= \circ )23(1)\circ = \circ )23()1\circ = \circ )2(31)\circ = \circ )2(3)1\circ , \\
G(t_2, t_3, t_1) &= \circ 1)3(2)\circ = \circ 1)3()2\circ = \circ 1)(32)\circ = \circ 1)(3)2\circ \\
&= \circ )13(2)\circ = \circ )13()2\circ = \circ )1(32)\circ = \circ )1(3)2 \circ .
\end{aligned}
\tag{88}
$$

For these, the degeneracy for the representation of each Wightman correlator is $h_{3,2}^{(2)} = 8$.

## 6.3 Four-point functions

Finally, let us discuss four-point functions and how various correlators are related to each other.

- **Wightman basis ($n! = 24$ correlators):** These are labeled by permutations of four objects. Explicitly, the 24 independent Wightman correlators are:

$$
G_\sigma(t_1, t_2, t_3, t_4) \equiv G(t_{\sigma(1)}, t_{\sigma(2)}, t_{\sigma(3)}, t_{\sigma(4)}) = \langle \widehat{\mathbb{O}}_{\sigma(1)} \widehat{\mathbb{O}}_{\sigma(2)} \widehat{\mathbb{O}}_{\sigma(3)} \widehat{\mathbb{O}}_{\sigma(4)} \rangle \qquad \sigma \in S_4 .
\tag{89}
$$

Out of these, there are $g_{4,1} = 8$ proper 1-OTO correlators, and $g_{4,2} = 16$ proper 2-OTO correlators (see (16) and (17)). As before, one can obtain all 24 Wightman correlators from analytic continuation of $G_E(\tau_1, \tau_2, \tau_3, \tau_4)$, by setting $\tau_i = it_i + \epsilon_i$ and making choices for the relative ordering of the $\epsilon_i$.

- **Nested correlators ($2^{n-2} n! = 96$ correlators):** these are all of the general form

$$
\langle (((( \widehat{\mathbb{O}}_{\sigma(1)}, \widehat{\mathbb{O}}_{\sigma(2)} ))_{\alpha}, \widehat{\mathbb{O}}_{\sigma(3)} ))_{\beta}, \widehat{\mathbb{O}}_{\sigma(4)} ))_{\gamma} \rangle
\tag{90}
$$

for $2^3 \times 4! = 192$ choices of $\alpha, \beta, \gamma \in \{av, dif\}$ and permutations $\sigma \in S_4$. Note that we get the same correlators up to signs for any two permutations which only exchange the innermost arguments in (90) (or are distinct from other permutations only by such an operation). The total number of correlators is hence $\frac{192}{2} = 96$ as expected. We can reduce these to a basis of 24 independent correlators by using 24 proper and 48 improper sJacobi identities, see (31), (32) and permutations thereof.

- **Av-Dif correlators ($(2k)^n = 16k^4$ correlators on a $k$-OTO contour):** For 4-point functions, a 2-OTO contour is sufficient to capture all correlators. For $k = 2$ the 256 Av-Dif-type correlators can be written as

$$\langle \mathcal{T}_C\, \mathbb{O}^\alpha_\alpha(1)\mathbb{O}^\beta_\beta(2)\mathbb{O}^\gamma_\gamma(3)\mathbb{O}^\delta_\delta(4)\rangle. \tag{91}$$

We can again use the double bracket to relate these to the nested correlators:

$$
\begin{aligned}
\langle \mathcal{T}_C\, \mathbb{O}^\alpha_\delta(1)\mathbb{O}^1_\alpha(2)\mathbb{O}^1_\beta(3)\mathbb{O}^1_\gamma(4)\rangle &= \mathfrak{d}_\delta \langle (\!(\!(\!(\!(\widehat{\mathbb{O}}(1),\widehat{\mathbb{O}}(2))\!)_\alpha,\widehat{\mathbb{O}}(3))\!)_\beta,\widehat{\mathbb{O}}(4))\!)_\gamma\rangle, \\
\langle \mathcal{T}_C\, \mathbb{O}^\alpha_\alpha(1)\mathbb{O}^2_\delta(2)\mathbb{O}^1_\beta(3)\mathbb{O}^1_\gamma(4)\rangle &= \mathfrak{d}_\delta \langle (\!(\!(\!(\!(\widehat{\mathbb{O}}(2),\widehat{\mathbb{O}}(1))\!)_\alpha,\widehat{\mathbb{O}}(3))\!)_\beta,\widehat{\mathbb{O}}(4))\!)_\gamma\rangle, \\
\langle \mathcal{T}_C\, \mathbb{O}^\alpha_\alpha(1)\mathbb{O}^1_\delta(2)\mathbb{O}^2_\delta(3)\mathbb{O}^1_\gamma(4)\rangle &= \mathfrak{d}_\delta \langle (\!(\!(\!(\!(\widehat{\mathbb{O}}(3),\widehat{\mathbb{O}}(1))\!)_\alpha,\widehat{\mathbb{O}}(2))\!)_\beta,\widehat{\mathbb{O}}(4))\!)_\gamma\rangle, \\
\langle \mathcal{T}_C\, \mathbb{O}^\alpha_\alpha(1)\mathbb{O}^1_\beta(2)\mathbb{O}^1_\gamma(3)\mathbb{O}^2_\delta(4)\rangle &= \mathfrak{d}_\delta \langle (\!(\!(\!(\!(\widehat{\mathbb{O}}(4),\widehat{\mathbb{O}}(1))\!)_\alpha,\widehat{\mathbb{O}}(2))\!)_\beta,\widehat{\mathbb{O}}(3))\!)_\gamma\rangle, \\
\langle \mathcal{T}_C\, \mathbb{O}^\alpha_\beta(1)\mathbb{O}^2_\alpha(2)\mathbb{O}^2_\delta(3)\mathbb{O}^1_\gamma(4)\rangle &= \mathfrak{d}_\delta \langle (\!(\!(\!(\!(\widehat{\mathbb{O}}(3),\widehat{\mathbb{O}}(2))\!)_\alpha,\widehat{\mathbb{O}}(1))\!)_\beta,\widehat{\mathbb{O}}(4))\!)_\gamma\rangle, \\
\langle \mathcal{T}_C\, \mathbb{O}^\alpha_\beta(1)\mathbb{O}^2_\alpha(2)\mathbb{O}^1_\gamma(3)\mathbb{O}^2_\delta(4)\rangle &= \mathfrak{d}_\delta \langle (\!(\!(\!(\!(\widehat{\mathbb{O}}(4),\widehat{\mathbb{O}}(2))\!)_\alpha,\widehat{\mathbb{O}}(1))\!)_\beta,\widehat{\mathbb{O}}(3))\!)_\gamma\rangle, \\
\langle \mathcal{T}_C\, \mathbb{O}^\alpha_\beta(1)\mathbb{O}^1_\gamma(2)\mathbb{O}^2_\alpha(3)\mathbb{O}^2_\delta(4)\rangle &= \mathfrak{d}_\delta \langle (\!(\!(\!(\!(\widehat{\mathbb{O}}(4),\widehat{\mathbb{O}}(3))\!)_\alpha,\widehat{\mathbb{O}}(1))\!)_\beta,\widehat{\mathbb{O}}(2))\!)_\gamma\rangle, \\
\langle \mathcal{T}_C\, \mathbb{O}^\alpha_\gamma(1)\mathbb{O}^2_\beta(2)\mathbb{O}^2_\alpha(3)\mathbb{O}^2_\delta(4)\rangle &= \mathfrak{d}_\delta \langle (\!(\!(\!(\!(\widehat{\mathbb{O}}(4),\widehat{\mathbb{O}}(3))\!)_\alpha,\widehat{\mathbb{O}}(2))\!)_\beta,\widehat{\mathbb{O}}(1))\!)_\gamma\rangle.
\end{aligned}
\tag{92}
$$

Each line encodes 32 equations, giving a total of $32 \times 8 = 256$ relations.

- **LR correlators ($(2k)^n = 16k^4$ elements on a $k$-OTO contour):** Taking $k = 2$, there are 256 LR correlators. As mentioned before, these can all be related to only 8 proper 1-OTO and 16 proper 2-OTO Wightman correlators. Writing all $256 - 8 - 16 = 232$ relations is tedious, but we wish to illustrate the degeneracy by giving one particular 1-OTO and one particular 2-OTO Wightman 4-point function and demonstrating their various representations in the $k = 2$ contour:

$$
\begin{aligned}
G(t_1, t_2, t_3, t_4) &= \circ4321)()\circ = \circ432)1()\circ = \circ432)(1)\circ = \circ432)()1\circ \\
&= \circ43)2(1)\circ = \circ43)2()1\circ = \circ43)(21)\circ = \circ43)(2)1\circ \\
&= \circ4)3(21)\circ = \circ4)3(2)1\circ = \circ4)(321)\circ = \circ4)(32)1\circ \\
&= \circ)4(321)\circ = \circ)4(32)1\circ = \circ)(4321)\circ = \circ)(432)1\circ, \\
G(t_1, t_3, t_2, t_4) &= \circ42)3(1)\circ = \circ42)3()1\circ = \circ42)(31)\circ = \circ42)(3)1\circ \\
&= \circ4)23(1)\circ = \circ4)23()1\circ = \circ4)2(31)\circ = \circ4)2(3)1\circ,
\end{aligned}
\tag{93}
$$

corresponding to degeneracies $h^{(1)}_{4,2} = 16$ and $h^{(2)}_{4,2} = 8$, respectively.

### 6.3.1 Chaos correlator

We wish to briefly discuss a particular 4-point function, which has been argued to be a measure of quantum chaos [14–16]. For consistency with the literature, we assume here that

$$t_1 = t_2 = t, \quad t_3 = t_4 = 0, \quad \widehat{\mathbb{O}}_1 = \widehat{\mathbb{O}}_2 \equiv \widehat{\mathbb{W}}, \quad \widehat{\mathbb{O}}_3 = \widehat{\mathbb{O}}_4 \equiv \widehat{\mathbb{V}}. \tag{94}$$

The correlator of interest (up to a sign) is $\mathcal{C}(t) \equiv \langle [\widehat{\mathbb{V}}(0), \widehat{\mathbb{W}}(t)]^2\rangle$ with $t > 0$, which clearly is a proper 2-OTO correlator. We can represent this object in the different representations described above as follows:

- **Wightman basis:** The correlator $\mathcal{C}(t)$ is a linear combination of 4 Wightman correlators (the first 3 of which are proper 2-OTO, and the last one is proper 1-OTO):

$$\mathcal{C}(t) = G(0, t, 0, t) + G(t, 0, t, 0) - G(t, 0, 0, t) - G(0, t, t, 0). \tag{95}$$

- **Nested correlators:** using nested commutators and anti-commutators, there are many ways to represent $\mathcal{C}(t)$. A particularly simple representation is the following:

$$\mathcal{C}(t) = \Big\langle \frac{1}{2}\{[[\widehat{\mathbb{V}}(0), \widehat{\mathbb{W}}(t)], \widehat{\mathbb{V}}(0)], \widehat{\mathbb{W}}(t)\} + \frac{1}{4}[[\{\widehat{\mathbb{W}}(t), \widehat{\mathbb{W}}(t)\}, \widehat{\mathbb{V}}(0)], \widehat{\mathbb{V}}(0)] \Big\rangle. \quad (96)$$

- **Av-Dif correlators** the representation in this form is again not unique. One particularly concise way to write $\mathcal{C}(t)$ is found to be

$$\mathcal{C}(t) = -\frac{1}{4}\Big\langle \mathcal{T}_C\Big( (\mathbb{V}^1_{av} - \mathbb{V}^2_{av})^2 - \frac{1}{4}(\mathbb{V}^1_{dif} - \mathbb{V}^2_{dif})^2 \Big)\Big( (\mathbb{W}^1_{av} + \mathbb{W}^2_{av})^2 - \frac{1}{4}(\mathbb{W}^1_{dif} + \mathbb{W}^2_{dif})^2 \Big) \Big\rangle. \quad (97)$$

We discuss the degeneracy of this representation in the following bullet point.

- **LR correlators:** again, the representation of $\mathcal{C}(t)$ in the LR representation is not unique. One way to write it is the following:

$$\begin{aligned}\mathcal{C}(t) = &\langle \mathcal{T}_C\, \mathbb{W}^1_{\text{R}}(t)\mathbb{V}^2_{\text{R}}(0)\mathbb{W}^2_{\text{R}}(t)\mathbb{V}^1_{\text{L}}(0)\rangle + \langle \mathcal{T}_C\, \mathbb{V}^1_{\text{R}}(0)\mathbb{W}^2_{\text{L}}(t)\mathbb{V}^2_{\text{R}}(0)\mathbb{W}^1_{\text{L}}(t)\rangle \\ &- \langle \mathcal{T}_C\, \mathbb{W}^1_{\text{R}}(t)\mathbb{V}^2_{\text{L}}(0)\mathbb{V}^2_{\text{R}}(0)\mathbb{W}^1_{\text{L}}(t)\rangle - \langle \mathcal{T}_C\, \mathbb{V}^1_{\text{R}}(0)\mathbb{W}^2_{\text{L}}(t)\mathbb{W}^2_{\text{R}}(t)\mathbb{V}^1_{\text{L}}(0)\rangle. \end{aligned} \quad (98)$$

The degeneracy of this representation is best described in symbolic notation and by noting that each of the four terms in (98) has some degeneracy by itself; for example

$$\begin{aligned}\langle \mathcal{T}_C\, \mathbb{W}^1_{\text{R}}(t)\mathbb{V}^2_{\text{L}}(0)\mathbb{W}^2_{\text{R}}(t)\mathbb{V}^1_{\text{L}}(0)\rangle = {}&\circ w)v(w)v\circ = \circ w)v()wv\circ = \circ)wv(w)v\circ = \circ)wv()wv\circ \\ = {}&\circ w)(vw)v\circ = \circ w)(v)wv\circ = \circ)w(vw)v\circ = \circ)w(v)wv\circ, \end{aligned} \quad (99)$$

where $w$ and $v$ denote insertions of $\mathbb{W}^\alpha_{\bar{\alpha}}$ and $\mathbb{V}^\alpha_{\bar{\alpha}}$ ($\bar{\alpha} \in \{\text{R},\text{L}\}$, $\alpha \in \{1,2\}$). Note that (98) has three pieces which are proper 2-OTO, and one piece which is proper 1-OTO. The full correlator $\mathcal{C}(t)$ hence has a degeneracy of $16 \times 8^3 = 8192$ in the LR correlators.

# 7 Applications to simple systems

We now exemplify the abstract discussion above with some explicit examples in simple quantum models. We will here only present only the basic results to illustrate the general features.

## 7.1 Example: Quantum harmonic oscillator

Let us begin with the case of a quantum harmonic oscillator. Evolution in one dimension admits a natural time-ordering (which we recall is not necessarily associated with relativistic invariance). We consider definite time-ordering of Heisenberg operators in correlation functions.

Let $X(t)$ denotes the position of a particle in a harmonic oscillator of frequency $\mu$. We present the results for the two and four point functions of $X(t)$ at various times in a $n^{\text{th}}$ excited state $|n\rangle$. We present both the Wightman functions and the nested correlators involving both commutators and anti-commutators.

The position operator $X(t)$ and Hamiltonian can written in terms of creation and annihilation operators as usual (with $m = 1$ for simplicity)

$$X(t) = \frac{1}{\sqrt{2\mu}}\Big( a\, e^{-i\mu t} + a^\dagger e^{i\mu t} \Big), \qquad H = \mu\Big( a^\dagger a + \frac{1}{2} \Big). \quad (100)$$

The action of creation and annihilation on energy eigenstates are given by the familiar expressions:

$$a \,|n\rangle = \sqrt{n}\,|n-1\rangle \qquad a^\dagger \,|n\rangle = \sqrt{n+1}\,|n+1\rangle. \tag{101}$$

It is of course straightforward to compute correlation functions in the quantum mechanical theory. The fastest way to proceed is to decompose the time evolution and insert a complete set of states between operators to reduce the computation into evaluating matrix elements of appropriate operators. For instance,

$$\langle n|\,X(t_1)X(t_2)\,|n\rangle = \sum_m \langle n|\,e^{iHt_1}X\,e^{-iHt_1}\,|m\rangle\,\langle m|\,e^{iHt_2}X\,e^{-iHt_2}\,|n\rangle.$$

Higher point functions can be obtained by iteration of this logic.

**Two-point functions:** Using the above definitions, it is easy to show that (with $t_{ij} \equiv t_i - t_j$)

$$2\mu\langle n|\,X(t_1)X(t_2)\,|n\rangle = e^{-i\mu t_{12}} + 2n\cos(\mu t_{12}), \tag{102}$$

from which one may deduce that

$$\begin{aligned}
2\mu\langle n|\,[X(t_1),X(t_2)]\,|n\rangle &= -2i\sin(\mu t_{12}),\\
2\mu\langle n|\,\{X(t_1),X(t_2)\}\,|n\rangle &= 2\cos(\mu t_{12})(1+2n).
\end{aligned} \tag{103}$$

It must be noted that the commutators have much simpler expressions than the corresponding Wightman functions. With suitable dressing by time-ordering step functions they end up giving the causal response functions of the theory.

**Four-point functions:** The explicit computation of the Heisenberg operators yields the 4-point function:

$$\begin{aligned}
(2\mu)^2\langle n|\,X(t_1)X(t_2)X(t_3)X(t_4)\,|n\rangle &= n(n-1)e^{i\mu(t_{13}+t_{24})} + n^2 e^{i\mu(t_{12}+t_{34})}\\
&+ (n+1)\Big(n\,e^{i\mu(t_{21}+t_{34})} + (n+1)e^{i\mu(t_{21}+t_{43})} + (n+2)e^{i\mu(t_{31}+t_{42})} + n\,e^{i\mu(t_{12}+t_{43})}\Big),
\end{aligned} \tag{104}$$

from which we can obtain the nested correlators:

$$\begin{aligned}
(2\mu)^2\langle n|\,\{[[X(t_1),X(t_2)],X(t_3)],X(t_4)\}\,|n\rangle &= 0,\\
(2\mu)^2\langle n|\,[[[X(t_1),X(t_2)],X(t_3)],X(t_4)]\,|n\rangle &= 0,\\
(2\mu)^2\langle n|\,[[\{X(t_1),X(t_2)\},X(t_3)],X(t_4)]\,|n\rangle &= -8\big[\sin(\mu t_{23})\sin(\mu t_{14}) + \sin(\mu t_{24})\sin(\mu t_{13})\big],\\
(2\mu)^2\langle n|\,[\{[X(t_1),X(t_2)],X(t_3)\},X(t_4)]\,|n\rangle &= -8\sin(\mu t_{12})\sin(\mu t_{34}),\\
(2\mu)^2\langle n|\,\{\{[X(t_1),X(t_2)],X(t_3)\},X(t_4)\}\,|n\rangle &= -8i(2n+1)\cos(\mu t_{34})\sin(\mu t_{12}),\\
(2\mu)^2\langle n|\,\{[\{X(t_1),X(t_2)\},X(t_3)],X(t_4)\}\,|n\rangle &= -8i(2n+1)\big(\cos(\mu t_{14})\sin(\mu t_{23}) + \cos(\mu t_{24})\sin(\mu t_{13})\big),\\
(2\mu)^2\langle n|\,[\{\{X(t_1),X(t_2)\},X(t_3)\},X(t_4)]\,|n\rangle &= -8i(2n+1)\big[\sin(\mu(t_{13}+t_{24})) + \sin(\mu(t_{12}+t_{34}))\\
&\qquad\qquad -\sin(\mu(t_{12}-t_{34}))\big],\\
(2\mu)^2\langle n|\,\{\{\{X(t_1),X(t_2)\},X(t_3)\},X(t_4)\}\,|n\rangle &= 8[1+2n(n+1)]\big[\cos(\mu(t_{13}+t_{24})) + \cos(\mu(t_{12}+t_{34}))\\
&\qquad\qquad + \cos(\mu(t_{12}-t_{34}))\big].
\end{aligned} \tag{105}$$

Once again we note a relative simplicity in the expressions for the nested correlators.

## 7.2 Scalar Field Theory

We now turn to a relativistic QFT. We will first exemplify the above statements with scalar field theory.[18] In fact, the non-trivial information we need is already in the free theory itself. Once

---

[18] For related studies of chaos in field theory, see for example **??**.

we isolate the pieces involving the propagators we can set up Feynman rules for computing $k$-OTO correlation functions in perturbation theory.

Say we want to compute the $k$-OTO correlation functions for a scalar $\phi^4$ theory with action (working with mostly plus signature):

$$S = \int d^d x\, \mathcal{L}[\phi(x)] = -\int d^d x \left( \frac{1}{2} \partial_\mu \phi\, \partial^\mu \phi + \frac{1}{2} m^2 \phi^2 + \lambda\, \phi^4 \right). \qquad (106)$$

Before writing down the rules for arbitrary $k$-OTO correlators, let us recall some well-known facts for the Schwinger-Keldysh 1-OTO case (see eg., [9]). In this case we have four possible two-point functions associated with the operators which we label by their location on the forward (1R) and backward (1L) segments of the contour, respectively. We arrange these four Green's functions into a $2 \times 2$ matrix, which can be easily evaluated to be (in the 1R,1L basis).

$$G(x,y) = \begin{bmatrix} \langle \mathcal{T}_\mathcal{C}\, \phi_{\text{R}}^1(x)\phi_{\text{R}}^1(y) \rangle & \langle \mathcal{T}_\mathcal{C}\phi_{\text{R}}^1(x)\phi_{\text{L}}^1(y) \rangle \\ \langle \mathcal{T}_\mathcal{C}\, \phi_{\text{L}}^1(x)\phi_{\text{R}}^1(y) \rangle & \langle \mathcal{T}_\mathcal{C}\, \phi_{\text{L}}^1(x)\phi_{\text{L}}^1(y) \rangle \end{bmatrix} = \begin{bmatrix} \langle \mathcal{T}\phi(x)\phi(y) \rangle & \langle \phi(y)\phi(x) \rangle \\ \langle \phi(x)\phi(y) \rangle & \langle \bar{\mathcal{T}}\phi(x)\phi(y) \rangle \end{bmatrix}. \qquad (107)$$

In momentum space[19], the above matrix can be evaluated to be

$$G(p) = \begin{pmatrix} \frac{-i}{p^2+m^2-i\epsilon} & 2\pi\delta(p^2+m^2)\theta(-p_0) \\ 2\pi\delta(p^2+m^2)\theta(p_0) & \frac{i}{p^2+m^2+i\epsilon} \end{pmatrix}, \qquad (109)$$

where we have used the $i\epsilon$ prescription for the time ordered correlator. The corresponding position space results can be found for example in [8] which we reproduce here for convenience:

$$
\begin{aligned}
\langle\, \mathcal{T}_{SK}\, \phi_{\text{R}}(x)\, \phi_{\text{R}}^\dagger(y) \,\rangle &= \int_p \left\{ \Theta_{xy}\, e^{ip.(x-y)} + \Theta_{yx}\, e^{-ip.(x-y)} \right\}, \\
\langle\, \mathcal{T}_{SK}\, \phi_{\text{R}}(x)\, \phi_{\text{L}}^\dagger(y) \,\rangle &= \int_p e^{-ip.(x-y)}, \\
\langle\, \mathcal{T}_{SK}\, \phi_{\text{L}}(x)\, \phi_{\text{R}}^\dagger(y) \,\rangle &= \int_p e^{ip.(x-y)}, \\
\langle\, \mathcal{T}_{SK}\, \phi_{\text{L}}(x)\, \phi_{\text{L}}^\dagger(y) \,\rangle &= \int_p \left\{ \Theta_{yx}\, e^{ip.(x-y)} + \Theta_{xy}\, e^{-ip.(x-y)} \right\}.
\end{aligned}
\qquad (110)
$$

where we have abbreviated the integral to represent $\int_p \mathfrak{I} \equiv \int \frac{1}{(2E_p)} \frac{d^{d-1}p}{(2\pi)^{d-1}} \mathfrak{I}$.

Armed with this information we can now write down the higher OTO two-point functions. To do so let us first introduce some notation. Define a collective index $\mathcal{I}$ such that $\phi_\mathcal{I} \equiv \phi_\alpha^I$, with $I = 1, 2, \cdots, k$ labeling the contours, while $\alpha \in \{\text{R}, \text{L}\}$ indexes the orientation. We also define an ordering relation on the contours: denote the occurrence of $\mathcal{I}$ before $\mathcal{J}$ along the contour as $\mathcal{I} > \mathcal{J}$. This takes care of implementing the contour-ordering prescription for us. With this at hand we can write:

$$\langle \mathcal{T}_\mathcal{C}\, \phi_\mathcal{I}(x)\phi_\mathcal{J}(y) \rangle = \begin{cases} \langle \phi(x)\phi(y) \rangle & \text{if } \mathcal{I} > \mathcal{J} \\ \langle \phi(y)\phi(x) \rangle & \text{if } \mathcal{J} > \mathcal{I} \\ \langle \mathcal{T}\, \phi(x)\phi(y) \rangle & \text{if } \mathcal{I} = \mathcal{J},\ \alpha = \text{R} \\ \langle \bar{\mathcal{T}}\, \phi(x)\phi(y) \rangle & \text{if } \mathcal{I} = \mathcal{J},\ \alpha = \text{L} \end{cases}. \qquad (111)$$

---

[19] We define Fourier transform via

$$\hat{\phi}(p) = \int d^d x\, e^{-ip.x}\phi(x) \qquad \phi(x) = \int \frac{d^d p}{(2\pi)^d}\, e^{ip.x}\hat{\phi}(p) \qquad (108)$$

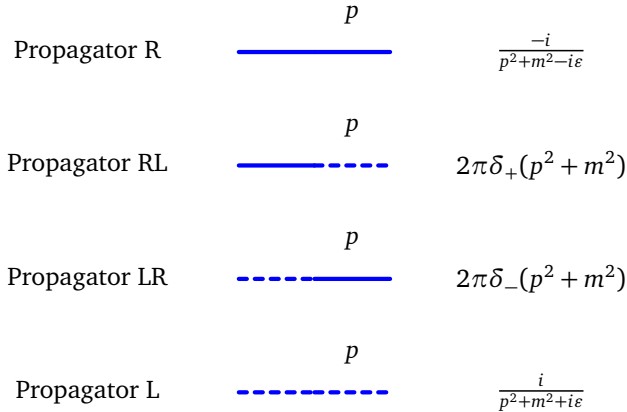

Figure 10: $k$-OTO propagator for a free scalar theory. Here, $\delta_\pm$ are the usual mass shell delta functions with $\theta(\pm p_0)$ as given in (109).

In frequency space, this can be summarized by the Feynman diagram rules given in Fig. 10.

With this information we have at our disposal the rules for carrying out perturbative computations of the correlators. We have the $(2k) \times (2k)$ matrix of propagators given by the above rules. The vertices for the interactions are localized to a given segment we simply have terms like $+\lambda\,\phi_{i\,\mathrm{R}}^4$ and $-\lambda\,\phi_{-i\,\mathrm{L}}^4$, respectively. We can then use this information to directly compute the $k$-OTO correlation functions with any desired time-ordering. This computation is well known in the case of Schwinger-Keldysh theory, and can be easily carried out at higher OTO order.

# 8 Discussion

The major part of our discussion has involved setting up a framework for the computation of general OTO correlation functions using a timefolded path integral. We have explained distinct collections of correlation functions, which are each adapted to either physically interesting observables (e.g., Wightman basis or nested correlators), or technical features of the OTO contour (LR or Av-Dif correlations). We have given rather explicit relations between the various collections, focusing in particular, on obtaining a canonical presentation of a given element of the Wightman basis in terms of the $k$-OTO contour.

Clearly, the reader will immediately appreciate, the analysis here is just the tip of the iceberg. Many interesting questions remain to be addressed, some of which we describe below. We also draw the attention of the reader to Section 11 of [8] where various physical questions involving 1-OTO contours were described. Many of those questions have natural analog in the $k$-OTO context ($k > 1$), and we elaborate on some of them in our discussion.

**BRST symmetries:** One of the underlying motivations for our analysis was to better understand the general set of constraints inherent in the $k$-OTO contours. As argued first in [44] and reviewed in some detail in [8], the Schwinger-Keldysh or 1-OTO contour has an underlying pair of BRST symmetries (see also [45, 46]). These symmetries are an efficient encoding of microscopic unitarity. In particular, this pair of BRST symmetries, called $\mathcal{Q}_{SK}$ and $\overline{\mathcal{Q}}_{SK}$ ensures that the relations between correlation functions in the LR or Av-Dif correlators are made manifest. The easy way to see these relations is to note the alignment, or topological limit. This pertains when we align the sources on the forward/backward legs of the contour, whence us-

ing $U U^\dagger = \mathbb{1}$, we will see that correlation functions involving Dif operators are constrained.[20] For instance, the correlator involving only Dif operators vanishes, as does a correlator when the Dif operator is futuremost.

For $k$-OTOs with $k > 1$, we have seen that we have many more relations which increase with $k$. This naturally suggests that the number of BRST charges should increase with $k$. As noted in [8] there are many distinct localizations for the case of $k = 2$. One can have *full localization*, in which the $k$-OTO generating function collapses to the trace over the initial density matrix, i.e., $\mathcal{Z}_{k-oto} = \text{Tr}(\hat{\rho}_{\text{initial}})$. These can be attained in $2k - 1$ distinct ways. One can also have *partial localization* whence a $k$-OTO contour collapses to a $j$-OTO contour for $j \leq k$. We have seen features of this in our discussion of finding a canonical presentation of an $n$-point function in terms of a proper $q$-OTO.

Based on the analysis of $k = 2$, it was conjectured in [8] that the $k$-OTO contour should have $2k$ BRST charges (which split into $k$ BRST charges and their CPT conjugates). We are currently investigating whether this structure suffices to obtain the various localizations and gives the captures the full set of redundancies inherent in the $k$-OTO contour. We hope to report on this issue in the near future.[21]

**Rényi entropies and replica:**    Consider the computation of Rényi entropy for a reduced density matrix in a time-evolving state using a real-time path integral. A canonical way to perform the computation involves stringing together various copies of the reduced density operator $(\rho_\mathcal{A})^k = \rho_\mathcal{A} \times \rho_\mathcal{A} \times \cdots \times \rho_\mathcal{A}$ and then taking the trace over the subsystem $\mathcal{A}$. So far this appears innocuous, but there is an important wrinkle in that we have to adhere to casual ordering of events. As argued in [47] we have to compute $\rho_\mathcal{A}$ using an appropriate Schwinger-Keldysh contour. Stringing together copies of $\rho_\mathcal{A}$ then involves $k$-copies of a Schwinger-Keldysh contour each with its associated forward/backward legs, which then leads to a $k$-OTO like contour, which was described in [47] (see their Fig. 5).

Thus the $k$-OTO contours find a natural home in implementing replica construction in real-time physics. There are two distinctions between our construction for correlators and that for Rényi entropies. One which is more or less obvious is that the gluing of the segments is different for the subsystem $\mathcal{A}$ and its complement at the future turning-point. The other, more important distinction, is that the structure of the past turning-points differs. While we glue segments by projecting the two ends of a turning point against a maximally entangled state in the two-copy Hilbert space, in the Rényi entropy computation, the past turning-points refer back to the density matrix the system was prepared in.[22]

**Gravity dual of timefolds:**    The OTO correlators as discussed pertain to non-gravitational quantum systems. However, via the holographic AdS/CFT correspondence they translate into questions that can be asked on the dual gravitational side. A general question then is how does one interpret a field theory $k$-OTO contour in the dual gravity variables? This issue has been addressed in different guises in the past, e.g., [5, 14] (see also [48] for a concrete proposal). One would like however to be able to directly find a covariant translation of these contours, keeping manifest perhaps some of the symmetries alluded to above. We consider this an interesting challenge, addressing which could help shed better light on the nature of the holographic map.

---

[20] It is important to note here that the difference operators couple to the average sources.

[21] We thank Michael Geracie and David Ramirez for extensive discussions and collaboration on understanding the BRST symmetries.

[22] The quantum information theoretic interpretation of oto correlators analyzed in [29, 30] (see also [23]), relates these observables to Rènyi entropies, by first averaging all operators in a given subsystem. One can argue that this averaging is tantamount to sampling over operator insertions on a contour, which can equivalently be recast in terms of inserting a copy of the reduced density operator at turning points.

**Perturbative QFT analysis:** On a perhaps more prosaic (albeit practically important) level, it would be interesting to develop perturbative QFT tools to tackle the $k$-OTO functional integral. The basic framework for such analysis has already been laid out in [10]. In particular, they have shown that the OTO chaos correlator satisfies a diffusion equation with non-linear dissipation (in the kinetic theory limit). It would be interesting to generalize this to other OTOs, and derive some effective Boltzmann type equation to capture their content. Another interesting question involves extending the known relation between Schwinger-Keldysh formalism and Veltman's cutting rules to the domain of higher OTO correlators.

# Acknowledgments

It is a great pleasure to thank Michael Geracie and David Ramirez for useful discussions and for ongoing collaboration on related ideas. We further thank Dan Roberts and Beni Yoshida for useful discussions on out-of-time-ordered correlators. FH gratefully acknowledges support through a fellowship by the Simons Collaboration 'It from Qubit'. RL gratefully acknowledges support from International Centre for Theoretical Sciences (ICTS), Tata institute of fundamental research, Bengaluru and Ramanujan fellowship from Govt. of India. PN and RL would also like to acknowledge their debt to the people of India for their steady and generous support to research in the basic sciences. MR would like to thank the organizers of Quantum Entanglement 2017 at Fudan University, Shanghai and NCTS, Taipei, and the organizers of Kavli Asian Winter School 2017 for their kind hospitality during the concluding stages of this project.

# A   Derivation of the $k$-OTO Keldysh rules

We now proceed to simplify (39). Let us first write out the expression for the correlation function by passing from the average-difference to the left-right basis. We find an expression of the form

$$
\left[\chi^{z_1}\chi^{z_2}\cdots\chi^{z_{n_k}}\right]\langle\mathcal{T}_{\mathcal{C}}\left\{\left[\mathbb{O}^1_{z_1}(1)\ldots\mathbb{O}^1_{z_{m_1}}(m_1)\mathbb{O}^1_{z_{m_1+1}}(m_1+1)\ldots\mathbb{O}^1_{z_{n_1}}(n_1)\right]\right.
$$
$$
\times\left[\mathbb{O}^2_{z_{n_1+1}}(n_1+1)\ldots\mathbb{O}^2_{z_{m_2}}(m_2)\mathbb{O}^2_{z_{m_2+1}}(m_2+1)\ldots\mathbb{O}^2_{z_{n_2}}(n_2)\right] \tag{112}
$$
$$
\left.\times\cdots\times\left[\mathbb{O}^k_{z_{n_{k-1}+1}}(n_{k-1}+1)\ldots\mathbb{O}^k_{z_{m_k}}(m_k)\mathbb{O}^k_{z_{m_k+1}}(m_k+1)\ldots\mathbb{O}^k_{z_{n_k}}(n_k)\right]\right\}\rangle.
$$

We simplified notation by combining the row matrices $\xi$ and $\eta$ into a single entity $\chi$ defined to be

$$
\chi^{z_i} = \begin{cases} \xi^{z_i}, & n_j+1 < i < m_j, \text{ for any } j \\ \eta^{z_i}, & m_j+1 < i < n_j, \text{ for any } j \end{cases}. \tag{113}
$$

We can now proceed to simplify the expression (112) into a single-copy correlation function as a sequence of nested (graded) commutators and anti-commutators. The logic is similar to the one described in [9], except that we have to employ it iteratively across the multiple segments of the $k$-OTO path integral. The recursion is however easy to set-up. We start with the contours closest to the density matrix, i.e., those indexed by $\alpha = 1$, and work our way further out. At each stage we use the step function normalization condition (34) to write the expression.

Let us see how this works starting with the segments labeled with $\alpha = 1$, i.e., the operators $\mathbb{O}^1$ in (112). We first move all the operators on segments further away from the density matrix $\alpha \geq 2$ to the right owing to the fact that they appear under the contour ordering symbol.

In fact they are spectators for analyzing the contribution from the first segments; we will therefore concatenate them into a abstract symbol $\mathcal{X}_{\alpha\geq 2}$. One can then insert the identity $1 = \sum_{\sigma \in S_{n_1}} \Theta_{\sigma(1)\sigma(2)\cdots\sigma(n_1)}$ and rearrange the operators to respect the time ordering suggested for each term of this sum. Implementing this we find

$$
\begin{aligned}
&\left[\chi^{z_1}\cdots\chi^{z_{n_1}}\right]\langle\mathcal{T}_{\mathcal{C}}\left\{\left[\mathbb{O}^1_{z_1}(1)\ldots\mathbb{O}^1_{z_{m_1}}(m_1)\mathbb{O}^1_{z_{m_1+1}}(m_1+1)\ldots\mathbb{O}^1_{z_{n_1}}(n_1)\right]\mathcal{X}_{\alpha\geq 2}\right\}\rangle \\
&=\sum_{\sigma\in S_{n_1}}\Theta_{\sigma(1)\sigma(2)\cdots\sigma(n_1)}\left[\chi^{z_1}\cdots\chi^{z_{n_1}}\right]\langle\mathcal{T}_{\mathcal{C}}\left\{\left[\mathbb{O}^1_{z_1}(1)\ldots\mathbb{O}^1_{z_{m_1}}(m_1)\mathbb{O}^1_{z_{m_1+1}}(m_1+1)\ldots\mathbb{O}^1_{z_{n_1}}(n_1)\right]\mathcal{X}_{\alpha\geq 2}\right\}\rangle \\
&=\sum_{\sigma\in S_{n_1}}\Theta_{\sigma(1)\sigma(2)\cdots\sigma(n_1)}\left[\chi^{z_{\sigma(1)}}\cdots\chi^{z_{\sigma(n_1)}}\right]\langle\mathcal{T}_{\mathcal{C}}\left\{\left[\mathbb{O}^1_{z_{\sigma(1)}}(\sigma(1))\ldots\mathbb{O}^1_{z_{\sigma(n_1)}}(\sigma(n_1))\right]\mathcal{X}_{\alpha\geq 2}\right\}\rangle.
\end{aligned}
$$
$$(114)$$

We can now simplify this expression as follows. Pick a particular temporal ordering say $t_{\sigma(1)} > t_{\sigma(2)} > \cdots > t_{\sigma(n_1)}$. The earliest time is $t_{\sigma(n_1)}$ by this choice. Picking out the corresponding operator we now examine whether this term originates from an average or a difference operator. The former will lead to an anti-commutator and the latter to a commutator. Let us carry out this exercise explicitly by isolating the term of interest. This is

$$
\text{Term} = \Theta_{\sigma(1)\sigma(2)\cdots\sigma(n_1)}\langle\mathcal{T}_{\mathcal{C}}\left\{\left[\mathcal{Y}_1\chi^{z_{\sigma(n_1)}}\mathbb{O}^1_{\sigma(n_1)}\right]\mathcal{X}_{\alpha\geq 2}\right\}\rangle, \tag{115}
$$

where $\mathcal{Y}_1$ denotes the other operators we have not singled out. There are two cases of interest:

- For $1 < \sigma(n_1) < m_1$ the operator $\chi^{z_{\sigma(n_1)}}\mathbb{O}^1_{\sigma(n_1)}$ is the average operator on the 1$^{\text{st}}$ segment. Ordering the 1R field to be the latest insertion and 1L to be the earliest insertion we learn that we should read this term as

$$
\begin{aligned}
\text{Term} &= \frac{1}{2}\Theta_{\sigma(1)\sigma(2)\cdots\sigma(n_1)}\left(\langle\mathcal{T}_{\mathcal{C}}\left[\mathcal{Y}_1\mathbb{O}^1_{\text{R}}(\sigma(n_1))\mathcal{X}_{\alpha\geq 2}\right]\rangle + \langle\mathcal{T}_{\mathcal{C}}\left[\mathbb{O}^1_{\text{L}}(\sigma(n_1))\mathcal{Y}_1\mathcal{X}_{\alpha\geq 2}\right]\rangle\right) \\
&= \frac{1}{2}\Theta_{\sigma(1)\sigma(2)\cdots\sigma(n_1)}\langle\mathcal{T}_{\mathcal{C}}(\mathcal{Y}_1\mathcal{X}_{\alpha\geq 2})\mathbb{O}^1_{\text{L}}(\sigma(n_1)) + \mathbb{O}^1_{\text{R}}(\sigma(n_1))\,\mathcal{T}_{\mathcal{C}}(\mathcal{Y}_1\mathcal{X}_{\alpha\geq 2})\rangle \\
&= \Theta_{\sigma(1)\sigma(2)\cdots\sigma(n_1)}\langle\{\mathcal{T}_{\mathcal{C}}(\mathcal{Y}_1\mathcal{X}_{\alpha\geq 2}),\widehat{\mathbb{O}}(\sigma(n_1))\}_{\pm}\rangle.
\end{aligned}
$$
$$(116)$$

  In the last line we have expressed the result as the anticommutator of the single copy operator with the remainder of the fields.

- A similar exercise can be carried through for $m_1 < \sigma(n_1) < n_1$ whence the operator $\chi^{z_{\sigma(n_1)}}\mathbb{O}^1_{\sigma(n_1)}$ is the difference operator on the 1$^{\text{st}}$ segment. The only difference is the relative sign leading to the end result

$$
\text{Term} = \Theta_{\sigma(1)\sigma(2)\cdots\sigma(n_1)}\langle\left[\mathcal{T}_{\mathcal{C}}(\mathcal{Y}_1\mathcal{X}_{\alpha\geq 2}),\widehat{\mathbb{O}}(\sigma(n_1))\right]_{\pm}\rangle. \tag{117}
$$

The astute reader will realize that the argument given above is simply the standard derivation of the Keldysh rules for the Schwinger-Keldysh 1-OTO contour

# B  Recurrence relations for degeneracy factor

In §5 we derived the numbers $h^{(q)}_{n,k}$ describing the degeneracy of the representation of a proper $q$-OTO $n$-point function using a $k$-OTO contour. While various definitions of these rather non-trivial numbers were given there (see, e.g., (52)–(55)), we found some additional recursion relations, which we collect here.

Some recursion relations satisfied by $h_{n,k}^{(q)}$ are:

$$(k-q)h_{n,k}^{(q)} = 2(n+1-q)h_{n,k-1}^{(q)} + (q+k-2)h_{n,k-2}^{(q)},$$

$$\sum_{k=q}^{\infty}(k+2-q)h_{n,k+2}^{(q)}t^k = 2(n+1-q)\sum_{k=q}^{\infty}h_{n,k+1}^{(q)}t^k + \sum_{k=q}^{\infty}(k+q)h_{n,k}^{(q)}t^k. \tag{118}$$

Similarly, one can verify the following differential recurrence relations involving the generating functional $\mathcal{H}_{q,n}(t)$ of (53):

$$\frac{(t\partial_t-q)(\mathcal{H}_{q,n}(t)-t^q h_{n,q}^{(q)}-t^{q+1}h_{n,q+1}^{(q)})}{t^2} = (t\partial_t+q)\mathcal{H}_{q,n}(t) + \frac{2(n+1-q)(\mathcal{H}_{q,n}(t)-t^q h_{n,q}^{(q)})}{t},$$

$$(t(1-t^2)\partial_t-q(1+t^2))\mathcal{H}_{q,n}(t)-t^{q+1}h_{n,q+1}^{(q)} = 2(n+1-q)t(\mathcal{H}_{q,n}(t)-t^q h_{n,q}^{(q)}). \tag{119}$$

Other recurrence relations which hold for $k > q, n \geq 2q$ are:

$$h_{n,k}^{(q)} = 2\sum_{j=q}^{k-1}h_{n-1,j}^{(q)} + h_{n-1,k}^{(q)}, \qquad \text{or} \qquad h_{n,k}^{(q)} + h_{n-1,k}^{(q)} = 2\sum_{j=q}^{k}h_{n-1,j}^{(q)},$$

$$h_{n,k}^{(q)} = h_{n,k-1}^{(q)} + h_{n-1,k}^{(q)} + h_{n-1,k-1}^{(q)},$$

$$h_{n,k}^{(q)} + 4\sum_{j=q-1}^{k-1}(-)^{k-j}(k-j)h_{n,j}^{(q-1)} = 0, \tag{120}$$

$$h_{n,k}^{(q)} + h_{n,k-1}^{(q)} = 4\sum_{j=q}^{k}(-)^{k-j}h_{n,j-1}^{(q-1)}.$$

Finally, we can complement Table 2 by giving some explicit values of $h_{n,k}^{(q)}$ for special values of the parameters. For $k \leq q+2$ we find:

$$h_{n,k<q}^{(q)} = 0, \qquad h_{n,k=q}^{(q)} = 2^{2q-1}, \qquad h_{n,k=q+1}^{(q)} = (n+1-q)2^{2q},$$

$$h_{n,k=q+2}^{(q)} = [2q^2-(4n+3)q+2(n+1)^2]2^{2q-1}. \tag{121}$$

Similarly, we have the following expressions for some relevant values of $n$:

$$h_{n=2q-1,k}^{(q)} = 2^{2q-1}\binom{n+k-q}{n} = 2^{2q-1}\frac{(k+q-1)!}{(k-q)!n!},$$

$$h_{n=2q,k}^{(q)} = \frac{k}{q}h_{n=2q-1,k}^{(q)} = \frac{2k}{\binom{n}{1}}h_{n=2q-1,k}^{(q)} = (2k)2^{2q-1}\frac{(k+q-1)!}{(k-q)!n!},$$

$$h_{n=2q+1,k}^{(q)} = \frac{2k^2+q}{q(2q+1)}h_{n=2q-1,k}^{(q)} = \frac{2k^2+q}{\binom{n}{2}}h_{n=2q-1,k}^{(q)} = 2(2k^2+q)2^{2q-1}\frac{(k+q-1)!}{(k-q)!n!}, \tag{122}$$

$$h_{n=2q+2,k}^{(q)} = \frac{2k^3+(3q+1)k}{q(q+1)(2q+1)}h_{n=2q-1,k}^{(q)} = 4k(2k^2+3q+1)2^{2q-1}\frac{(k+q-1)!}{(k-q)!n!}.$$

## C  Mathematical details

### C.1  Proof of Lemma

In this appendix we provide the proof of the following group theory result:

*Lemma:* The regular representation of the group is induced by regular representation of a subgroup.

*Proof:* To prove this statement we will begin with the following fact: say we are given a group $G$, a subgroup $H$, and a set of coset representatives $C_H = \{r_i\}$. Given a representation $\rho(H)$ of $H$, the character of the corresponding induced representation is given by

$$\chi^{\rho(H)\uparrow G}(g) = \sum_{r_i \in C_H} \delta_H\left(r_i^{-1} g \, r_i\right) \chi^{\rho(H)}(r_i^{-1} g \, r_i), \tag{123}$$

where $\delta_H(\alpha) = 1$ if $\alpha \in H$ and is zero otherwise. We now use the character in regular representation $\chi^{\mathcal{R}(H)}(h) = |H| \, \delta_{h,e}$ where $|H|$ is the number of elements in $H$ to get

$$
\begin{aligned}
\chi^{\mathcal{R}(H)\uparrow G}(g) &= \sum_{r_i \in C_H} \delta_H\left(r_i^{-1} g \, r_i\right) \chi^{\mathcal{R}(H)}(r_i^{-1} g \, r_i) = \sum_{r_i \in C_H} \delta_H\left(r_i^{-1} g \, r_i\right) \delta_{r_i^{-1} g \, r_i, e} \, |H| \\
&= \sum_{r_i \in C_H} \delta_{g,e} \, |H| = |G| \, \delta_{g,e} = \chi^{\mathcal{R}(G)}(g) \, .
\end{aligned}
\tag{124}
$$

Thus, the induced representation coming from a regular representation of a subgroup $H$ is indeed the regular representation of $G$.

## C.2 Proof of Theorem 1

We provide a proof of the following result:

*Theorem 1:* Proper $n$ sJacobi identities lie in the regular representation $\mathcal{R}(S_n)$.

*Proof:* We use induction to demonstrate that improper sJacobis lie in $(2^{n-2} - 2)$ copies of $\mathcal{R}(S_n)$.

For $n = 3$ there are no improper Jacobis (since the minimum number of operators required to form an sJacobi is 3) and so all sJacobis are proper and they lie in a $(2^{n-2} - 1) = 1$ copy of $\mathcal{R}(S_3)$. So, our claim is true for $n = 3$.

Next assume proper $k$ sJacobis lie in the regular representation $\mathcal{R}(S_k)$ for all $3 \le k < n$. We then need to show that proper $n$ sJacobis lie in the regular representation $\mathcal{R}(S_n)$. Given a proper $k$ sJacobi in $\mathcal{R}(S_k)$, we nest it within $(n-k)$ number of commutators/anti-commutators in $2^{n-k}$ ways. This gives $2^{n-k}$ copies of representation $\mathcal{R}(S_k) \times \mathcal{R}(S_{n-k})$ of the subgroup $S_k \times S_{n-k}$. We now invoke the lemma asserting that regular representation of a subgroup induces regular representation of the group to argue that the improper sJacobis coming from proper k sJacobis lie in the representation $2^{n-k}\mathcal{R}(S_n)$. Using $(2^{n-2} - 2) = \sum_{k=3}^{n-1} 2^{n-k}$, the set of all improper sJacobis then lie in $(2^{n-2} - 2)$ copies of $\mathcal{R}(S_n)$. This, as we have stated before, is equivalent to the assertion that proper $n$ sJacobis lie in the regular representation $\mathcal{R}(S_n)$. QED.

## C.3 Proof of Theorem 2

In §5.2 we have seen that the number of ways of generating an $n$-point correlator whose proper OTO number is less than or equal to $q$ from an $(n-1)$-point correlator whose proper OTO number is less than or equal to $q$ leads to the following recursion relation for $g_{n,q}$:

$$\sum_{j=1}^{q} g_{n,j} = n \sum_{j=1}^{q-1} g_{n-1,j} + (2q) g_{n-1,q} \, . \tag{125}$$

A useful version of this recursion is obtained by subtracting the recursion relation for $(q-1)$ from that of $q$:

$$g_{n+1,q+1} = 2(q+1)\, g_{n,q+1} + (n-2q+1)\, g_{n,q}\,. \tag{126}$$

We will now show how this recursion relation can be used to compute $g_{n,q}$. We will begin by setting $q = 0$ in the above which gives

$$g_{n+1,1} = 2g_{n,1}\,. \tag{127}$$

This along with $g_{1,1} = 1$ (viz., there is only one 1-OTO one-point function), then gives $g_{n,1} = 2^{n-1}$. Next, we set $q = 1$ in the recursion relation to get

$$g_{n+1,2} = (n-1)\, g_{n,1} + 4\, g_{n,2} = (n-1)\, 2^{n-1} + 4\, g_{n,2}\,. \tag{128}$$

This can be solved with the initial condition $g_{2,2} = 0$ (viz., there are no 2-OTO one-point function) to get $g_{n,2} = 2^{n-2}(2^{n-1} - n)$.

We can now proceed by rewriting the above recursion relation as a differential-difference equation for the generating function:

$$\mathcal{G}_n(\mu) \equiv \sum_{q=1}^{\lfloor \frac{n+1}{2} \rfloor} \mu^q g_{n,q}\,. \tag{129}$$

We obtain by direct manipulation

$$\mathcal{G}_n(\mu) = \left[ 2\mu(1-\mu)\frac{d}{d\mu} + n\mu \right] \mathcal{G}_{n-1}(\mu)\,. \tag{130}$$

This equation, along with the initial condition $\mathcal{G}_1(\mu) = \mu$ for 1-point functions, is solved by

$$\mathcal{G}_n(\mu) = \left(2\sqrt{1-\mu}\right)^{n+1} \mathrm{Li}_{-n}\!\left(\frac{2}{1+\sqrt{1-\mu}} - 1\right), \tag{131}$$

where $\mathrm{Li}_{-n}$ is the polylogarithm function of negative integer index defined via

$$\mathrm{Li}_{-n}(x) \equiv \sum_{j=1}^{\infty} j^n x^j = \left(x\frac{d}{dx}\right)^n \frac{x}{1-x}\,. \tag{132}$$

The first few $\mathcal{G}_n(\mu)$ are

$$\begin{aligned}
&\mathcal{G}_1(\mu) = \mu\,, \qquad \mathcal{G}_2(\mu) = 2\mu\,, \qquad \mathcal{G}_3(\mu) = 2\mu(\mu+2)\,, \\
&\mathcal{G}_4(\mu) = 8\mu(2\mu+1)\,, \qquad \mathcal{G}_5(\mu) = 8\mu(2\mu^2 + 11\mu + 2)\,.
\end{aligned} \tag{133}$$

Note that these polynomials are sometimes called the 'peak polynomials' in the combinatorics literature and it occurs as the integer sequence A008303 of the Online Encyclopedia of Integer Sequences (OEIS).

Once we have the functions $\mathcal{G}_n(\mu)$ we can recover the numbers $g_{n,q}$ of interest, as these are given by picking up the appropriate coefficients, viz.,

$$\begin{aligned}
g_{n,q} &= \text{Coefficient of } \mu^q \text{ in } \left(2\sqrt{1-\mu}\right)^{n+1} \mathrm{Li}_{-n}\!\left(\frac{2}{1+\sqrt{1-\mu}} - 1\right) \\
&= \text{Coefficient of } \frac{z^n}{n!}\mu^q \text{ in } \mu\left\{\left(1 - \frac{\tan(z\sqrt{\mu-1})}{\sqrt{\mu-1}}\right)^{-1} - 1\right\}.
\end{aligned} \tag{134}$$

This proves Theorem 2.

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
