# Peer review of "Classification of out-of-time-order correlators"

_SciPost Physics, doi:SciPost Phys. 6, 001 (2019)_

## Round 3 · Referee Report · Anonymous (Referee 1) · 2018-11-26

Strengths

1) Nice and clear formalism developed to calculate general out of time ordered correlation functions.

2) Thorough analysis that will likely be of use to the community in the future.

Weaknesses

1) Main body of the paper hard to read at the moment. Too many details in the main text, that can otherwise be moved to the appendices.

2) Beyond the formalism developed here (which is useful in its own right; see above), no fundamentally "new" results/insights at the moment.

Report

I have gone through the manuscript by Haehl et al. with a great deal of interest. In this work the authors have formally set up a framework for computing general (higher) OTO correlation functions using a timefolded path integral. I believe that the present manuscript will be of some use to those working on the subject in the future. I will recommend publication of this manuscript in SciPost.

However I do have some questions and comments for the authors that they might want to take into consideration before the paper is formally published.

To the best of my knowledge, most of the field-theoretic computations of the OTO correlation functions work with the "regularized" version of the correlators as opposed to the usual "unregularized" version. The latter object is physically more relevant. If the late time behavior (especially the exponential growth and Lyapunov exponent) are the same for both sets of correlators, then this regularization is harmless. Is this something that the authors can shed some light on using their formalism?

I bring up this question primarily since there has been some recent claim (https://arxiv.org/abs/1807.09799) that the two sets of correlators can lead to different Lyapunov exponents for a specific problem. Perhaps the authors can use their formalism to comment on what might lead to such differences in a more general setting even at late times?

Requested changes

1) The manuscript, in my opinion, is quite hard to read through at the moment. Even though the formalism and steps involved in the formal manipulations are relatively straightforward, I feel that by including an excessive amount of the technical details in the main text, the authors have made the manuscript harder to read. If the authors could move some of these details to the appendices, I feel that it will be much easier to read the main body of the paper.

On a related note, I think it would have been nicer for the authors to give some examples while setting up their formalism. At the moment, the few examples that they discuss appears in section 7, which is practically at the end of the paper. From the point of view of the young readers and newcomers to the field, I think a restructuring of the paper would be nicer (but is not mandatory).

2) In section 7.2, when the authors discuss the example of the scalar field theory, I think it would be appropriate for them to cite Phys. Rev. D 96, 065005 (2017).

3) A minor comment: I found it quite non-standard to use the acronym "W.l.o.g." for without loss of generality. I don't think it is that standardized and it only appears three times in the text. Perhaps the authors can refrain from using this acronym.

---

## Round 4 · Author Response

We thank the referee for their comments. We have made some changes as requested. these are detailed in the "List of Changes" below.

---

## Round 4 · List of Changes

The following changes have been made in the new version of the manuscript to incorporate the suggestions of the referee. The numbering refers to the referee's comments at the end of their report.

Re 1) We have moved some of the very mathematical details from three different parts of the paper to a new appendix (C). We also put boxes around some of the important results in the main text to highlight the essential points. We left the derivation of Eq. (5.20) in the main text since it is mostly about building intuition for the contours rather than complicated mathematics (we did, however, put a suggestion to skip to the main result, on page 33 before the derivation starts).

We did not move the example section (section 7) to the beginning of the paper. We believe that sections 1 and 2 are quite pedagogical and provide an extensive introduction to the subject. Furthermore, the way section 7 is written is largely motivated by the results of the paper, so we feel it is more appropriate to have it at the end rather than at the beginning.

Re 2) We have now included a citation to this paper as suggested.

Re 3) We have now spelled out ``w.l.o.g.'' everywhere (however, please see https://en.wikipedia.org/wiki/Without_loss_of_generality where the usage is noted as standard).

---

## Editorial Decision

published